# Migratory and anti-fibrotic programmes define the regenerative potential of human cardiac progenitors

Christine M. Poch [1,16], Kylie S. Foo[2,3,16], Maria Teresa De Angelis[1,4,16], Karin Jennbacken [5,16], Gianluca Santamaria[1,4,16], Andrea Bähr[1,16], Qing-Dong Wang [5], Franziska Reiter[1], Nadja Hornaschewitz[1], Dorota Zawada[1,4], Tarik Bozoglu[1], Ilaria My[1], Anna Meier [1,4], Tatjana Dorn[1,4], Simon Hege[1], Miia L. Lehtinen[3], Yat Long Tsoi [2], Daniel Hovdal [5], Johan Hyllner[5,6], Sascha Schwarz[7], Stefanie Sudhop[7], Victoria Jurisch[1], Marcella Sini[8], Mick D. Fellows[8], Matthew Cummings[9], Jonathan Clarke[10], Ricardo Baptista[10], Elif Eroglu [2], Eckhard Wolf [11], Nikolai Klymiuk[1,12], Kun Lu[13], Roland Tomasi[13], Andreas Dendorfer [12,13], Marco Gaspari[14], Elvira Parrotta[14], Giovanni Cuda [14], Markus Krane[12,15], Daniel Sinnecker [1,12], Petra Hoppmann[1], Christian Kupatt [1,12 ✉], Regina Fritsche-Danielson [5 ✉], Alessandra Moretti [1,4,12 ✉], Kenneth R. Chien [2,3 ✉] and Karl-Ludwig Laugwitz [1,12 ✉]

Heart regeneration is an unmet clinical need, hampered by limited renewal of adult cardiomyocytes and fibrotic scarring. Pluripotent stem cell-based strategies are emerging, but unravelling cellular dynamics of host–graft crosstalk remains elusive. Here, by combining lineage tracing and single-cell transcriptomics in injured non-human primate heart biomimics, we uncover the coordinated action modes of human progenitor-mediated muscle repair. Chemoattraction via CXCL12/CXCR4 directs cellular migration to injury sites. Activated fibroblast repulsion targets fibrosis by SLIT2/ROBO1 guidance in organizing cytoskeletal dynamics. Ultimately, differentiation and electromechanical integration lead to functional restoration of damaged heart muscle. In vivo transplantation into acutely and chronically injured porcine hearts illustrated CXCR4-dependent homing, de novo formation of heart muscle, scar-volume reduction and prevention of heart failure progression. Concurrent endothelial differentiation contributed to graft neovascularization. Our study demonstrates that inherent developmental programmes within cardiac progenitors are sequentially activated in disease, enabling the cells to sense and counteract acute and chronic injury.

Whereas mammals undergo endogenous cardiac regeneration during development and shortly after birth[1,2], the regenerative capacity of the human heart in adulthood is markedly low[3]. The inability to replace lost myocardium is accompanied by extensive tissue remodelling and fibrosis[4], leaving patients with cardiac disease vulnerable to heart failure. Although several drugs and mechanical devices can moderately improve cardiac function, they do not replace lost cardiomyocytes (CMs) or abolish fibrotic scar formation[5,6]. Biotherapies have emerged as innovative strategies for heart repair[7–10]. Induction of endogenous CM proliferation[11–14], in vivo direct reprogramming of non-CMs

to a cardiac fate[15] and exogenous transplantation of human pluripotent stem cell (hPSC)-derived CMs[16–18] or cardiac progenitors[19] have been recently explored as potential approaches to generate de novo myocardium.

Studies in lower vertebrates, where robust cardiac regeneration occurs throughout life, have demonstrated that endogenous heart repair is a highly coordinated process involving inter-lineage communication, cellular de-/re-differentiation, migration and extracellular matrix (ECM) remodelling without fibrotic scarring[20–23]. Similar programmes are the foundation of organ morphogenesis and are inherent of embryonic cardiac progenitors. During heart

[1]Medical Department I, Cardiology, Angiology, Pneumology, Klinikum rechts der Isar, Technical University of Munich, Munich, Germany. [2]Department of Cell and Molecular Biology, Karolinska Institutet, Stockholm, Sweden. [3]Department of Medicine, Karolinska Institutet, Huddinge, Sweden. [4]Institute of Regenerative Medicine in Cardiology, Technical University of Munich, Munich, Germany. [5]Research and Early Development, Cardiovascular, Renal and Metabolism (CVRM), BioPharmaceuticals R&D, AstraZeneca, Gothenburg, Sweden. [6]Division of Biotechnology, IFM, Linköping University, Linköping, Sweden. [7]Center for Applied Tissue Engineering and Regenerative Medicine (CANTER), Munich University of Applied Sciences, Munich, Germany. [8]Clinical Pharmacology and Safety Sciences, BioPharmaceuticals R&D, AstraZeneca, Cambridge, UK. [9]Western Michigan School of Medicine, Kalamazoo, MI, USA. [10]Procella Therapeutics, Stockholm, Sweden. [11]Chair for Molecular Animal Breeding and Biotechnology, Gene Center and Department of Veterinary Sciences, LMU Munich, Munich, Germany. [12]DZHK (German Centre of Cardiovascular Research), Munich Heart Alliance, Munich, Germany. [13]Walter-Brendel-Centre of Experimental Medicine, University Hospital, LMU Munich, Munich, Germany. [14]Department of Experimental and Clinical Medicine, University of Magna Grecia, Catanzaro, Italy. [15]Department of Cardiovascular Surgery, INSURE, German Heart Center Munich, Technical University of Munich, Munich, Germany. [16]These authors contributed equally: Christine M. Poch, Kylie S. Foo, Maria Teresa De Angelis, Karin Jennbacken, Gianluca Santamaria, Andrea Bähr. ✉e-mail: Christian.Kupatt@tum.de; Regina.Fritsche-Danielson@astrazenca.com; amoretti@mytum.de; kenneth.chien@ki.se; KL.Laugwitz@mri.tum.de

development, defined embryonic precursors, including first heart field (FHF) and second heart field (SHF), give rise to distinct cardiac compartments and cell types[24,25]. While FHF cells differentiate early into CMs of the primitive heart tube, ISL1+ SHF has broader lineage potential and its differentiation is preceded by an extensive proliferation and directed migration into the forming myocardium[26–28]. We recently reported the generation of an enriched pool of hPSC-derived ISL1+ ventricular progenitors (HVPs), which can expand and differentiate into functional ventricular CMs in vitro and in vivo[29].

In this Article, we sought to determine whether HVPs could effectively promote heart regeneration by orchestrating sequential programmes of cardiac development, ultimately leading to de novo myocardium formation and positively influencing fibrotic scar remodelling.

## Results

**HVPs functionally repopulate a tissue model of chronic heart failure.** To molecularly dissect HVP-mediated cardiac repair at the single-cell level, we utilize an ex vivo non-human primate (NHP) adult heart tissue model imitating key steps of heart failure. NHP left ventricle (LV) slices were cultured in biomimetic chambers[30], allowing structural and functional preservation for 14 days (Fig. 1a,b and Extended Data Fig. 1a). Thereafter, progressive loss of contractile force coincided with increased CM apoptosis (Fig. 1b,c and Extended Data Fig. 1a,b). *NKX2.5*^eGFP/wt human embryonic stem cells (hESC) were coaxed towards *ISL1*+/*NKX2.5*+ heart progenitors using our protocol enriching for HVPs[29], with small numbers of multipotent ISL1+ precursors[31] (Fig. 1a and Extended Data Fig. 1c). After magnetic-activated cell sorting (MACS)-based depletion of undifferentiated hESCs, cells were seeded onto NHP-LV slices by bioprinting (Extended Data Fig. 1c,d). Expression of enhanced green fluorescent protein (eGFP) enabled live tracing of HVPs and their derivative CMs (Extended Data Fig. 1e). Labelling with 5-ethynyl-2-deoxyuridine (EdU) and activated caspase-3 (ClCasp3) indicated that eGFP+ cells were highly proliferative until day 14 (D14), but stopped by D21 when NHP-CMs underwent substantial apoptosis (Fig. 1c and Extended Data Fig. 1b,f). This corresponded to extensive differentiation towards CMs and ISL1 downregulation (Extended Data Fig. 1g). Remarkably, heart slices gradually regained contraction in the third week of co-culture (Extended Data Fig. 1e), reaching 2 mN force, and maintained to D50 (Fig. 1b and Extended Data Fig. 1e). Atrial and ventricular markers (MLC2a/MLC2v) revealed that ~81% of eGFP+ cells acquired ventricular identity by D50 (Fig. 1d). By then, most eGFP+/MLC2v+ CMs were rod shaped with well-aligned myofibrils, structural characteristics of maturation (Fig. 1d). A small proportion of cells expressing endothelial marker CD31 were detected (Fig. 1e), probably from multipotent precursors within the HVP pool.

To establish a molecular roadmap for HVP specification and maturation, we profiled cells on D0 and eGFP+ cells from D3 and D21 ex vivo co-culture by single-cell RNA sequencing (scRNA-seq). We integrated data with our published scRNA-seq from D−3 of in vitro differentiation[32]. Unsupervised clustering identified seven stage-dependent subpopulations (Fig. 2a). On D−3, corresponding to cardiac lineage commitment[33], cells expressed high levels of early cardiac mesodermal genes (*EOMES*, *MESP1* and *LGR5*). On D0, cells distributed into four distinct clusters: transcriptomes of early (*KRT18* and *ID1*), intermediate (*KRT8* and *PRDX1*) and proliferating (*TOP2A* and *CCNB1-2*) progenitor states including cardiac mesenchymal cells (*PLCE1* and *PPA1*). Transcripts related to ECM organization (*DCN*, *TIMP1*, *LUM*, *FN1* and *COL3A1*) and ventricular structure/maturation (*MYL3*, *TTN*, *TNNC1*, *ACTC1* and *PLN*) defined late eGFP+ cells and ventricular CMs on D3 and D21 (Fig. 2a,b, Extended Data Fig. 2a and Source Data Fig. 2). Once aligned in a pseudotime trajectory[34], D3 cells bifurcated into two

lineages: endothelial-committed progenitors and HVPs with their CMs (Fig. 2c and Extended Data Fig. 2b).

Gene Ontology (GO) enrichment analysis of differentially expressed genes (DEGs) in cells from D0 to D21 revealed progressive activation of terms related to cardiac ventricular morphogenesis or maturation, while pathways relevant to cardiac progenitor state, such as ECM organization, cell cycle and canonical BMP signalling, were gradually suppressed (Fig. 2d). On D3, a significant enrichment of pathways important for progenitor proliferation and cardiac growth was detected, including canonical Wnt, ERK1/2 and TOR signalling (Fig. 2d). Interestingly, genes upregulated in HVPs at the early time of co-culture also associated with cell migration, cell projection organization, cytokine production and response to TGFβ (Fig. 2d), suggesting a specific sensing-reacting response of HVPs to the tissue environment. Notably, enriched vasculature development confirmed the potential of some early precursors to differentiate into vessels. To define the maturation of HVP-derived CMs, we integrated our data with published scRNA-seq of in vivo human adult ventricular muscle[35] in pseudotime (Fig. 2e and Extended Data Fig. 2c). D21 eGFP+ cells partially allocated together with adult ventricular CMs at the end of the differentiation trajectory and expressed high levels of structural, functional and metabolic genes characteristic of the adult state (Fig. 2e and Extended Data Fig. 2c). Quantitative PCR with reverse transcription (qRT–PCR) confirmed progressive myofibril maturation (sarcomeric isoform switching) and electrophysiological/Ca2+-handling maturation of eGFP+-CMs from D14 to D21 (Extended Data Fig. 2d).

Collectively, our single-cell transcriptomic analyses facilitated the construction of a differentiation route through which early mesodermal cardiac progenitors generate mature CMs in response to signalling cues of a dying myocardium.

**HVPs migrate and remuscularize acutely damaged myocardium.** Next, we designed an acute injury model in NHP-LV slices to simulate tissue death and elucidate HVP properties in response to injury (Fig. 3). We used radiofrequency ablation (RFA), clinically employed to terminate arrhythmogenic foci, to consistently destroy a defined area of cellular compartment, leaving the ECM scaffold intact (Extended Data Fig. 3a). Gradually, progressive invasion of activated cardiac fibroblasts (CFs) expressing the discoidin domain receptor 2 (DDR2) and increased collagen type I deposition were visible in the RFA-injured area, with tissue scarring by D21 (Extended Data Fig. 3b). We seeded equal amounts of *NKX2.5*^eGFP/wt HVPs or CMs onto bioprinted pluronic frames on one side of the NHP-LV slices, generated RFA injury on the opposite and evaluated the cellular response to the damage by eGFP live imaging (Fig. 3a). Fluorescence-activated cell sorting (FACS) analysis of the cells before seeding indicated their purity (Extended Data Fig. 3d). Contrary to CMs, HVPs departed from their deposition site and directionally migrated towards the injured region, colonizing it within 4 days (Fig. 3a and Extended Data Fig. 3e). By D15, HVPs differentiated into CMs and the RFA area appeared remuscularized, with new eGFP+-CMs properly organized on D21 (Fig. 3b,c). Proliferation rate of eGFP+ cells at the RFA injury declined progressively from D7 to D21 (Fig. 3d), confirming CM maturation. Significant reduction of scar volume was measured in HVP-treated heart slices, and tissue contractile function improved (Fig. 3e,f and Extended Data Fig. 3f). Real-time intracellular Ca2+ analysis demonstrated that, unlike CM-treated heart slices, Ca2+ waves propagated through the RFA injury when HVPs had been applied; here, HVP-derived CMs displayed intracellular Ca2+ concentration oscillations similar to and synchronized with the adjacent native NHP myocardium (Fig. 3g), indicative of electromechanical integration.

To dissect the mechanisms underlying directed HVP migration towards RFA and the subsequent positive remodelling during the scarring process, we evaluated the cellular composition of the tissue

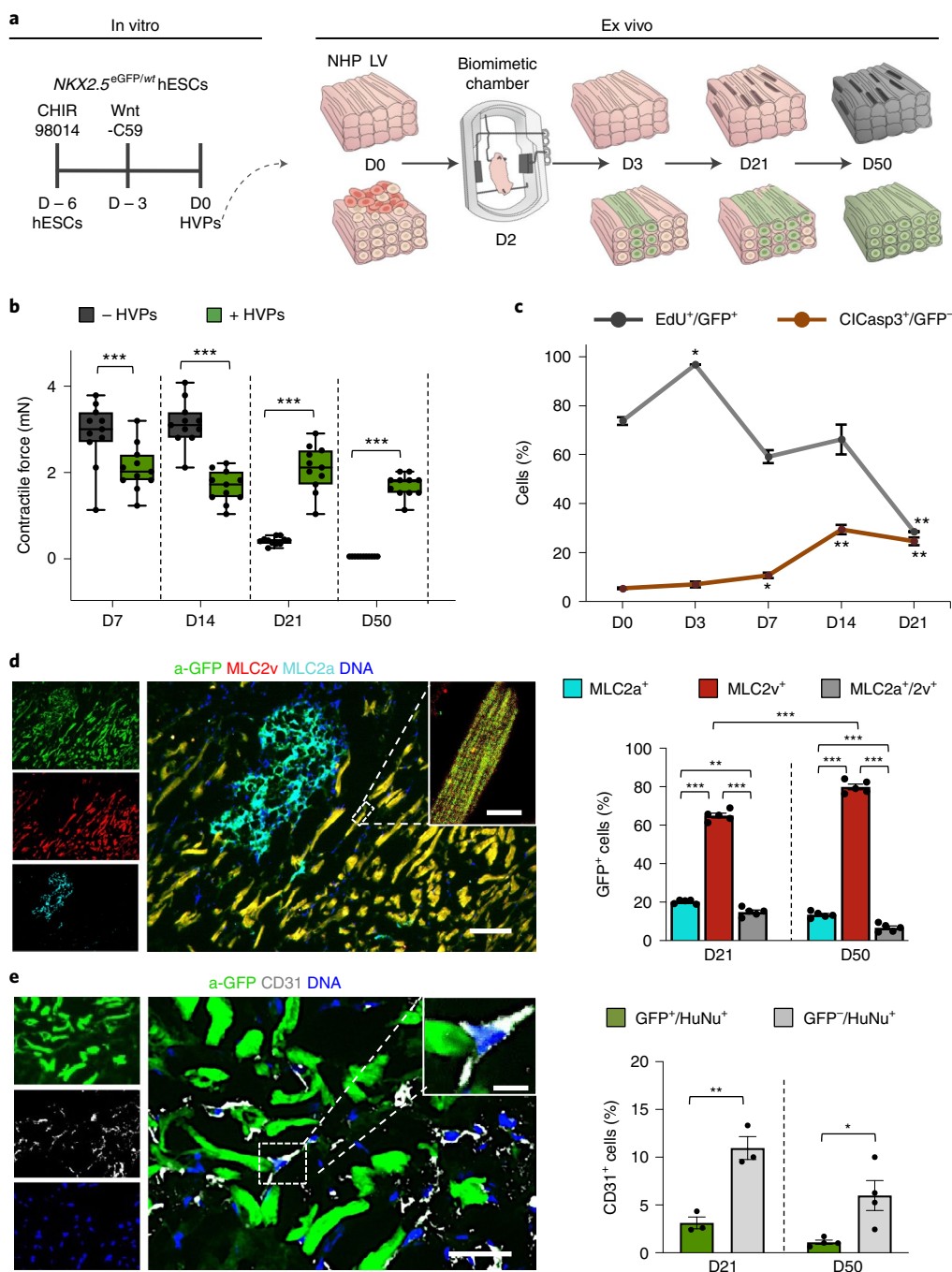

**Fig. 1 | HVPs expand, repopulate and functionally mature in an ex vivo 3D NHP heart model. a**, Schematic of the experimental setup for in vitro differentiation of HVPs from *NKX2-5*$^{eGFP/wt}$ hESCs (left) and their ex vivo co-culture with native NHP-LV slices in biomimetic chambers (right). **b**, Contractile force of ex vivo cultured NHP heart slices with and without HVPs on indicated days of co-culture. Box plot shows all data points as well as minimum, maximum, median and quartiles; $n = 11$ biological replicates per group; ***$P < 0.001$ (two-way ANOVA). **c**, Percentage of EdU$^+$/eGFP$^+$ and ClCasp3$^+$/eGFP$^-$ cells during co-culture. Data are mean ± s.e.m.; $n = 3$ biological replicates per timepoint for EdU analysis; $n = 4$ biological replicates per timepoint for ClCasp3 analysis; *$P < 0.05$, **$P < 0.005$ versus D0 (one-way ANOVA). **d,e**, Left: representative immunofluorescence images of D50 chimeric human–NHP heart constructs using an antibody against GFP (a-GFP) together with antibodies for MLC2a and MLC2v (**d**) or CD31 (**e**). Scale bars, 100 μm (**d**), 50 μm (**e**) and 10 μm (insets). Right: percentage of eGFP$^+$ cells expressing MLC2v, MLC2a or both (**d**) and human cells expressing CD31 (**e**) on D21 and D50. HuNu, human nuclear antigen. Data are mean ± s.e.m. and individual data points; $n = 5$ biological replicates per timepoint in **d**, $n = 3$ biological replicates per timepoint in **e**; *$P < 0.05$, **$P < 0.005$, ***$P < 0.001$ (two-way ANOVA for **d** and *t*-test for **e**). For **b–e**, exact $P$ values and numerical data are provided in Source Data Fig. 1.

around and at the injury site. One day after RFA, activated DDR2$^+$ NHP CFs heavily populated the border zone and reached the damaged area before eGFP$^+$-HVPs; both cells co-existed in the injured and surrounding regions after 1 week (Fig. 3h). Subsequently, the RFA site was predominantly colonized by eGFP$^+$ cells and the border zone by NHP DDR2$^+$ CFs (Fig. 3h). These observations suggested

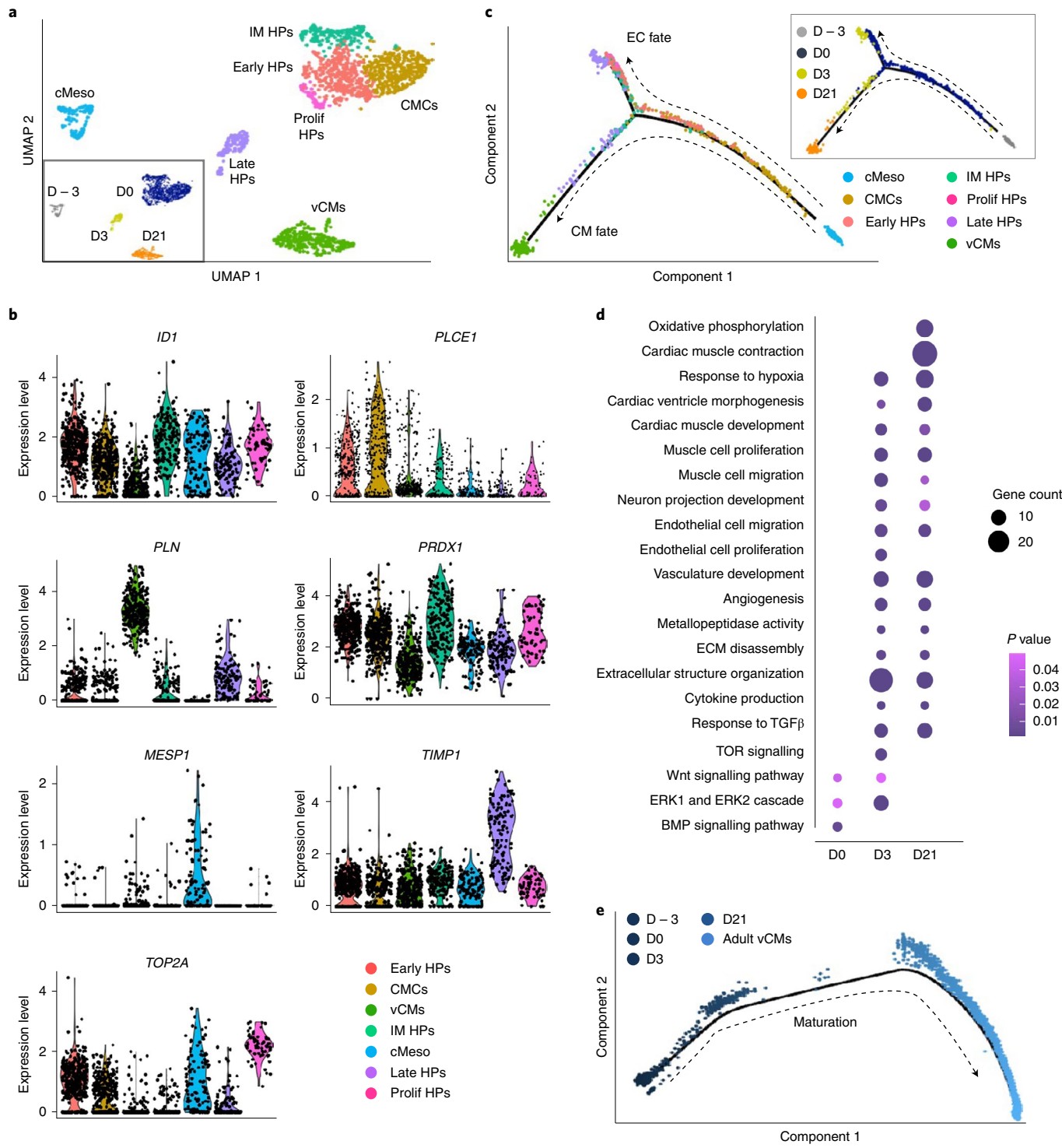

**Fig. 2 | scRNA-seq reveals dynamic transcriptional changes of HPs in the ex vivo 3D NHP heart model. a**, UMAP clustering of single cells captured on D − 3 and D0 of in vitro differentiation together with D3 and D21 of ex vivo co-culture. cMeso, cardiac mesoderm; CMCs, cardiac mesenchymal cells; early HPs, early heart progenitors; IM HPs, intermediate heart progenitors; late HPs, late heart progenitors; prolif HPs, proliferating heart progenitors; vCMs, ventricular CMs. **b**, Violin plots of cluster-specific marker genes; $P < 0.05$. **c**, Developmental trajectory analysis of captured cells coloured by population identity and time of collection (inset). EC, endothelial cell. **d**, Representative GO terms upregulated during ex vivo co-culture. **e**, Pseudotime trajectory of captured cells combined with adult vCMs from Wang et al.[35]. Colour gradient (from dark to light) according to maturation. For **a** and **c–e**, single cells have been dissociated from three biological replicates. Numerical data are provided in Source Data Fig. 2.

that cell–cell communication through chemokines or physical interaction between host CFs and human progenitors might instruct HVP migration, HVP differentiation and scar remodelling.

**CXCL12/CXCR4 signalling mediates HVP chemotaxis to injury sites.** Developmentally, ISL1+ HVPs are highly migratory during heart tube extension[24]. To elucidate the mode of HVP migration, we

performed a trans-well migration assay, where HVPs were placed on a permeable membrane and RFA-injured or uninjured NHP-LV slices at the bottom (Extended Data Fig. 3g). RFA significantly boosted migration. Interestingly, while multiple, homogeneously distributed RFAs prompted HVPs to migrate evenly, a directional migration towards the injured area was observed with a single RFA. No migration after RFA in decellularized NHP-LV slices confirmed that HVP migration is dependent on a chemoattractant gradient specifically arising from NHP cells at the damaged area (Extended Data Fig. 3g,h).

To molecularly examine the directed HVP chemotaxis and response, we profiled eGFP⁺ cells migrating (24 h, 485 cells) and arriving at the RFA injury (48 h, 269 cells) together with eGFP⁻ tissue-resident host cells (315 cells) by scRNA-seq (Fig. 4a). Seven clusters (0-6) were recovered, grouped into three populations (Fig. 4a, Extended Data Fig. 4a and Source Data Fig. 4). Clusters 1 and 4 belonged to the NHP group and mapped to CFs and monocytes/macrophages. Human cells formed the other two groups. One contained four clusters, which were classified as: early HVPs (rich in metabolic genes such as *MBOAT1*, *UQCRQ* and *MT-ND1,2,4,5,6*, but lacking CM transcripts; cluster 0), activated HVPs (*LAMA5*, *FLRT2* and *TNC*; cluster 2), proliferating HVPs (*TOP2A*, *CDC20* and *CCNB2*; cluster 5), and early ventricular CMs (*MYH6*, *MYL3* and *TNNC1*; cluster 6). The second group encompassed a homogeneous population of HVPs (cluster 3) characterized by high expression of genes involved in chemotaxis (*NRP1*, *CCL2-19-21*, *CXCL2-6-8-12*, *ITGB1*, *WASF1*, *RPS4X* and *INPPL1*), a unique gene signature not captured before. GO analysis of DEGs between cluster 3 and the other HVP clusters identified enrichment related to cell motility, chemotaxis, actin filament organization, axon-guidance cues and ECM organization (Extended Data Fig. 4b), supporting the migratory feature of this population. We also characterized the intercellular communication signals between HVPs and NHP cardiac cells by performing an in silico single-cell receptor–ligand pairing screen. We found over-representation pairing of CXCL12 as ligand with several membrane receptors: CXCR4, SDC4, ITGB1 and ACKR3 (Fig. 4b). While CXCL12, SDC4 and ITGB1 were expressed in HVPs and NHP fibroblasts, CXCR4 and ACKR3 receptors were highly enriched in the HVPs (Extended Data Fig. 4c). Trans-well migration assays under gain- and loss-of-function conditions demonstrated that HVPs exhibited enhanced migratory behaviour under CXCL12 as chemoattractant, which was reduced by blocking antibodies of CXCR4 or SDC4 and pharmacological inhibition of CXCR4 via ADM3100 (Fig. 4c). Notably, binding of CXCL12 to SDC4 facilitates its presentation to CXCR4 (ref. [36]). ADM3100 treatment was sufficient to inhibit HVP migration towards the RFA-injured area in NHP-LV slices (Extended Data Fig. 4d). Collectively, our data support the hypothesis that HVPs expressing CXCR4 sense CXCL12 secreted by CFs at the injury site as a chemoattracting signal to repopulate the damaged myocardium. Similarly, chemokine-controlled deployment of SHF cells has been identified as intra-organ crosstalk between progenitors and FHF CMs during mouse cardiogenesis[37], suggesting that migration programmes that are functional during development are re-activated in HVPs during regeneration.

**Dynamical cellular states underlie HVP regenerative potential.** To capture transition cell types and analyse the stepwise process of HVP-mediated cardiac repair, we integrated scRNA-seq data from HVPs (D0, 24 h, 48 h after RFA injury, and HVP-derived CMs on D21 co-culture; 2,114 cells) and generated a diffusion map of tissue-damage-induced cardiac differentiation (Fig. 4d,e). Heat mapping of gene expression with cells ordered in the trajectory revealed a temporal sequence of events and identified cells at intermediate stages of injury sensing and injury response (Extended Data Fig. 5a). Dot plotting illustrated gene signature shifts among different stages (Fig. 4f). In the first 24 h after injury, HVPs 'sense' the tissue damage and activate gene programmes for ECM remodelling (*COL6A1*, *ADAMTS9* and *FLRT2*), secretion and response to cytokine (*SPP1*, *STX8*, *TGFBI* and *IL6ST*), as well as initiation of migration (*PLAT*). Subsequently (48 h), they upregulate genes typical of migratory cells, including transcripts for chemoattraction (*PLXNA2*, *CMTM3* and *CXCL12*), cell motility (*SNAI1*, *SNAI2*, *FAT1* and *TIMP1*), cytoskeleton organization (*ARPC2*), axon guidance (*SLIT2*, *NFIB* and *UNC5B*) and cell projection (*RGS2*, *THY1* and *ITGA1*). During the migratory state, gene signatures of secretion (*COPB2*, *VPS35* and *SPTBN1*) and cardiac muscle differentiation (*VCAM1*, *MHY6*, *PALLD* and *TMOD1*) become increasingly important as a counteracting response to injury (Fig. 4f). Mass spectrometry analysis of supernatants from NHP-LV slices 48 h after RFA injury revealed a significant upregulation of secreted proteins in the presence of HVPs (Extended Data Fig. 5b). The majority are involved in ECM organization (HSPG2, SPARC and FN1) and fibrotic/inflammation response (FSTL1, PRDX1 and SPTAN1), reinforcing the concept of HVP-influenced scar remodelling.

**SLIT2/ROBO1 mediates HVP-guided fibroblast repulsion.** CFs are essential in cardiac development and repair[2,38]. To investigate the temporal and spatial crosstalk between CFs and HVPs in our ex vivo cardiac injury model, we isolated CFs from NHP hearts, stably expressed dsRed by lentiviral transduction and performed live imaging of co-culture with *NKX2.5*^eGFP/wt^ HVPs. RFA injury was applied on one site of the dsRed⁺-CF monolayer, while seeding of *NKX2.5*^eGFP/wt^ HVPs on the other (Fig. 5a). Like the native tissue, dsRed⁺-CFs were the first to invade the injured area, followed by eGFP⁺-HVPs within 5 days (Fig. 5a). Remarkably, while HVPs were directly chemoattracted to the injury, CFs appeared dynamically repelled at the contact sites with migrating HVPs (Fig. 5b and

**Fig. 3 | HVPs show directed migration towards acute cardiac RFA injury and remuscularize the scar. a**, Left: schematic of experimental design for selective seeding of *NKX2-5*^eGFP/wt^ hESC-derived HVPs or CMs onto bioprinted frame on NHP heart slices and RFA injury on the opposite tissue site. Right: live imaging of eGFP signal on indicated days. Scale bars, 200 μm. **b**, Representative immunostaining of a-GFP and cTNT in NHP constructs on D15 and D21 after RFA. Magnifications of framed areas are shown in adjacent panels. Scale bars, 200 μm (D15), 100 μm (D21) and 10 μm (magnifications). **c**, Statistical analysis of GFP⁺ HVPs expressing cTNT on D15 and D21. Data are mean ± s.e.m. and individual data points; *n* = 6 biological replicates per timepoint; \*\*\**P* < 0.001 (*t*-test). **d**, Left: immunofluorescence images of proliferating (PH3⁺) cells on D7 and D21. Right: statistical analysis of PH3⁺/GFP⁺ cells on D7, D15 and D21. Data are mean ± s.e.m. and individual data points; *n* = 3 biological replicates per timepoint; \*\**P* < 0.005 (one-way ANOVA). Scale bar, 100 μm. **e**, Statistical analysis of relative reduction of scar volume with HVPs compared with CMs on D21. Data are shown as mean ± s.e.m. and individual data points; *n* = 3 biological replicates per group; \*\**P* < 0.005 (*t*-test). **f**, Left: representative recordings of contractile force before and after RFA, separated by a blanking period of 2 days for re-adjustment of preload. Right: corresponding statistical analysis. Data are shown as mean ± s.e.m.; *n* = 3 biological replicates per condition; \**P* < 0.05 versus D7 of the same group (two-way ANOVA). **g**, Representative images of Fluo-4-loaded NHP-HVP and NHP-CM constructs (left) and corresponding Ca²⁺ transients at indicated regions of interest (ROI) (right). Scale bar, 100 μm. Red box indicates stimulation point (1 Hz). **h**, Left: representative immunostaining of a-GFP and DDR2 in NHP constructs at indicated days after RFA. Scale bars, 200 μm. Right: percentage of a-GFP⁺ and DDR2⁺ cells at RFA injury or border zone. Data are mean ± s.e.m.; *n* = 3 biological replicates per timepoint; \**P* < 0.05, \*\**P* < 0.005 versus D1 of corresponding group (two-way ANOVA). For **c**–**f** and **h**, exact *P* values and numerical data are provided in Source Data Fig. 3.

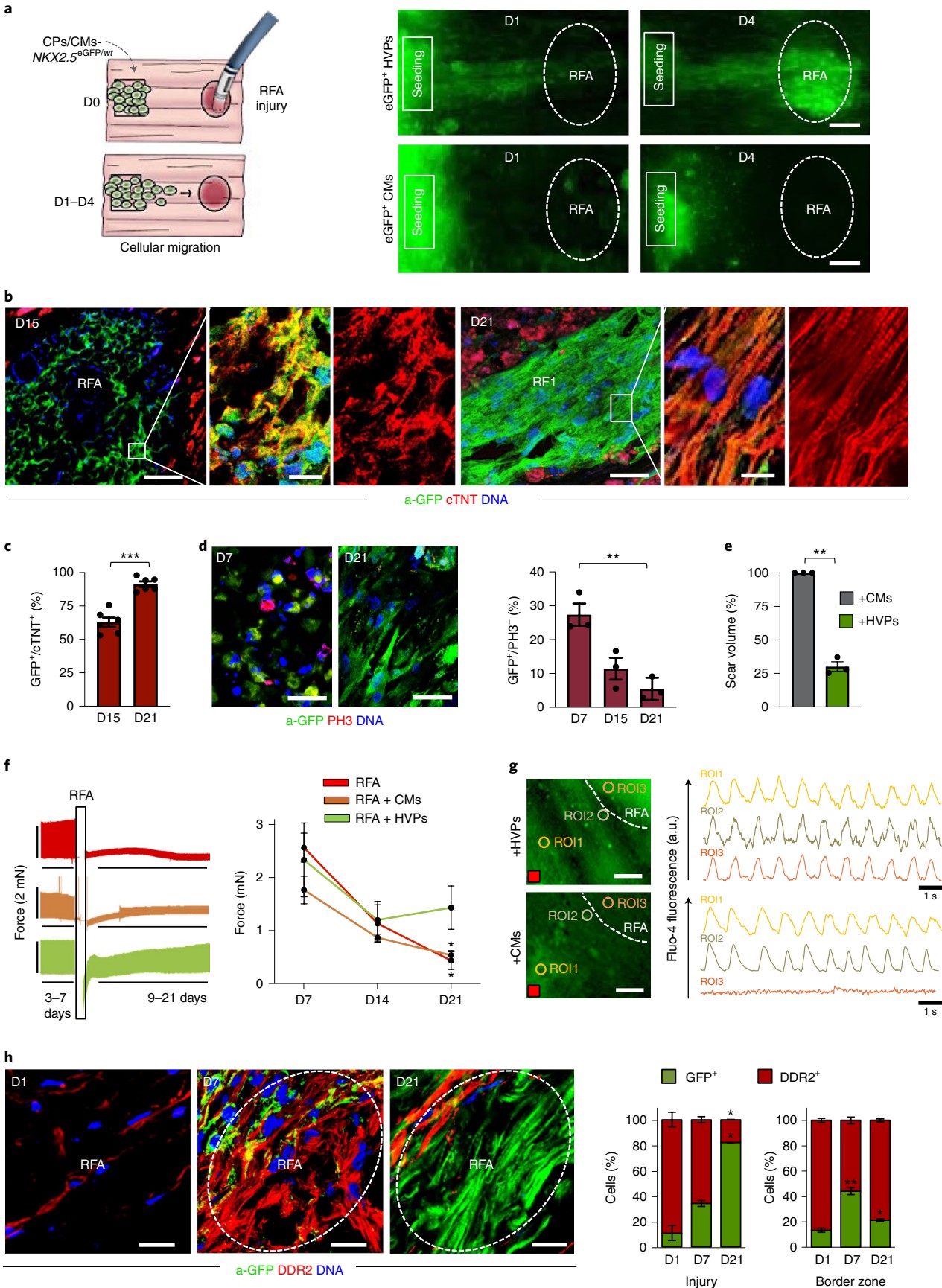

Supplementary Video 1). Live-cell tracking of over 100 cells for 3 days demonstrated that most CFs, after interacting with HVPs, indeed deviate from the HVP-migratory path and were repelled from the injured area when the HVPs started to densely populate it on D8 (Fig. 5b). Immunocytochemistry of filamentous (F)-actin revealed a specific retraction of cell protrusions precisely at cellular contact sites with the HVPs (Extended Data Fig. 6a), suggesting that the latter possibly control actin dynamics of CFs at the interaction sites. Given the upregulation of axon-guidance genes in the migratory HVP state, including SLIT2, we postulated that SLIT2/ROBO1, a known repulsive guidance cue for axons[39], might control HVP-mediated CF repulsion by regulating cytoskeletal organization and cell motion. Co-immunofluorescence demonstrated expression of SLIT2 ligand and ROBO1 receptor in migrating HVPs on D3, while the signal was absent in the surrounding CFs (Fig. 5c). On D8, however, co-localization of SLIT2 and ROBO1 was observed mainly at the repulsed CFs membrane, with enriched SLIT2 signal at the contact sites with HVPs (Fig. 5c). qRT–PCR confirmed SLIT2 production by HVPs and ROBO1 expression in both cell types at the stage of CF repulsion (Extended Data Fig. 6b). Loss-of-function experiments using an antibody blocking ROBO1 substantiated that, under ROBO1 inhibition, HVPs failed to induce actin polymerization and lamellipodia formation in the interacting CFs, leading to reduced CF motility and lack of repulsion (Fig. 5d,e and Extended Data Fig. 6c). No effects were observed in distant CFs (Extended Data Fig. 6c). Conversely, treatment with recombinant human SLIT2 enhanced F-actin content and membrane protrusions in CFs communicating with HVPs (Extended Data Fig. 6d), resulting in enhanced repulsion (Fig. 5e). FACS analysis indicated that most ROBO1+ CFs expressed periostin, a TGFβ superfamily-responsive protein defining a specialized reparative subpopulation of CFs required for healing and scar formation after injury[40] (Fig. 5f).

**HVPs migrate and regenerate injured porcine myocardium in vivo.** To investigate HVPs' ability to migrate and remuscularize injured myocardium in vivo, we performed transplantation in pigs ubiquitously expressing LEA29Y, a human CTLA4-Ig derivative blunting systemic T-cell response[41]. Two epicardial RFA injuries were induced afar in the anterior heart wall and $6 \times 10^7$ *NKX2.5*eGFP/wt HVPs were injected ~1 cm apart from one damaged site, while the other served as control (Fig. 6a and Supplementary Video 2). Assessment of RFA-induced tissue damage demonstrated consistent size of myocardial injury (Fig. 6b,c). Animals were treated daily with methylprednisolone and killed on D3 ($n = 1$), D5 ($n = 4$) and D14 ($n = 2$) post-transplantation. None showed any signs of tumour formation (Extended Data Fig. 7a). D3 and D5 immunohistology documented a directed, CXCR4-guided migration of eGFP+-HVPs towards the RFA-injured area (Fig. 6d,e). On D5, eGFP+ cells reached the RFA site in clusters, and repopulated $6.3 \pm 0.6\%$ of the scar (Fig. 6d,g). By D14, they constituted $21.0 \pm 2.9\%$ of the injured area, reducing control scar volume by half

(Fig. 6f,g). Remarkably, their highest concentration was at the epicardial layers with the largest damage (Fig. 6g). Most of eGFP+ cells engrafted in the injured tissue were elongated cardiac troponin T (cTNT)+ CMs with aligned myofibrils (Fig. 6h). Gap-junction protein connexin-43 was detected at the eGFP+-CMs' intercalated discs and at graft and host CMs' contact zone (Fig. 6h). CD31 immunostaining documented enhanced neo-angiogenesis at the RFA site after HVP transplantation, with ~6% of CD31+ cells of human origin (Fig. 6i and Extended Data Fig. 7b). No acute graft rejection was detected on D14 post-transplantation, as assessed by CD68 immunodetection. Interestingly, we even observed a reduction of CD68+ cells in HVP-treated RFA areas (Extended Data Fig. 7c), suggesting that HVPs might mitigate post-injury inflammation.

**HVPs remuscularize chronic scars and preserve cardiac function in vivo.** With a translational aim, we investigated HVPs' ability to engraft host myocardium in a porcine model of chronic ischaemic injury. Myocardial infarction (MI) was created by occluding the left anterior descending (LAD) coronary artery for 90 min, followed by reperfusion (Fig. 7a). Twenty-one days later, ~$1 \times 10^9$ HVPs (from WA09 hESCs) or vehicle were injected into the border zone and necrotic tissue of the MI region (Fig. 7a). Immunosuppressant started 6 days before cell delivery (Methods). Seventeen pigs underwent cardiac magnetic resonance imaging (cMRI) to study LV function and infarct volume 7 days before and 12 weeks after transplantation. No signs of teratoma or human DNA were detected in heart or other organs (lung, liver, kidney, spleen, brain, thyroid, adrenal glands, pituitary, prostate and lymph nodes) over the 3-month follow-up (Extended Data Fig. 8a,b).

Histological examination at 12 weeks indicated large cTNT+ human grafts in the fibrotic scar within the MI area and near the normal host tissue (Fig. 7b, Extended Data Fig. 8c and Supplementary Video 3). Graft size ranged from 3.0% to 9.4% of the scar area (mean $4.2 \pm 1.3\%$). Expression of MLC2v confirmed that most human CMs in the graft had ventricular identity and well-organized sarcomeres (Fig. 7c). Strong signals of N-cadherin, which anchor myofibrils with connexin-43, were observed at the intercalated discs of the human CMs within the transplants and at the graft–host tissue interconnection, suggesting functional maturation and graft integration (Fig. 7d). CD31 immunohistochemistry indicated enhanced neo-angiogenesis at the MI site following treatment (Fig. 7e).

At 3 months, cMRI (Fig. 8a) documented a significant reduction in infarct volume in the HVP group ($7.0 \pm 1.3\%$ versus $2.5 \pm 1.6\%$) (Fig. 8b). After induction of ischaemia before treatment, both groups exhibited equally depressed LV functions, with left-ventricular ejection fraction (LVEF) averaging 38% (vehicle $39.4 \pm 1.3\%$, HVP $37.3 \pm 2.8\%$). Over 12 weeks, LVEF further deteriorated significantly by ~10% in controls ($29.4 \pm 3.9\%$) and only by half (~5%) in HVP-treated animals ($31.9 \pm 3.0\%$), though differences between the groups did not reach statistical significance (Fig. 8c). However, the global longitudinal strain (GLS), a sensitive measure of LV function,

**Fig. 4 | HVPs are chemoattracted to sites of cardiac injury via CXCL12/CXCR4 signalling and undergo dynamic functional states in the process of injury repair. a**, Left: representative images of HVPs seeded on an injured NHP heart slice at the timepoints used for cell collection (24 h and 48 h) (top) and UMAP plot of all captured cells (bottom). Right: relative UMAP clustering of captured cells. Mφ, macrophages. **b**, Circos plot for ligand–receptor pairing showing top ten interactions identified in scRNA-seq of NHP-HVP constructs at 24 and 48 h after RFA injury and HVP application. Fraction of expressing cells and link direction (chemokine to receptor) are indicated. **c**, Percentage of chemoattracted HVPs in trans-well migration assays in absence and presence of low dose (LD) or high dose (HD) of CXCL12 (left), after addition of the indicated receptor blockers (middle) or after application of the pharmacologic CXCR4 blockage AMD3100 in LD or HD (right). Data are indicated as mean ± s.e.m. with individual data points; $n = 3$ biological replicates per condition; *$P < 0.05$, **$P < 0.005$, ***$P < 0.001$ versus CXCL12 HD (one-way ANOVA). **d**, Human scRNA-seq 24 h and 48 h datasets are integrated with D0 and D21 CM dataset and projected onto UMAP plots, coloured by cluster assignment and annotated post hoc. Both the aligned (left) and split (right) views are shown. HVPs (na), non-activated; HVPs (s), sensing; HVPs (m/c), migrating and counteracting. **e**, PCA plot of different cell clusters, with the principal curve indicating the pathway of injury response. **f**, Dot plot showing gene signature shifts among different dynamic cellular states. The shadings denote average expression and the size of dots the fractional expression. For **d–f**, single cells have been isolated from three biological replicates. Exact $P$ values and numerical data are provided in Source Data Fig. 4.

significantly worsened in the vehicle-treated ($-3.1 \pm 1.0$) compared with the HVP group ($-0.2 \pm 0.6$) (Fig. 8d), demonstrating that HVP treatment attenuated the progressive decline of cardiac function in this model.

## Discussion

Human CMs have poor proliferative potential, resulting in virtually non-existent de novo CM renewal after injury. The inability to replace lost contractile units after acute MI is paralleled by

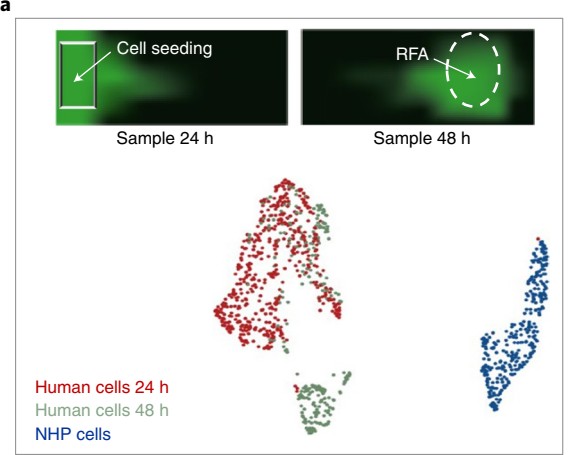
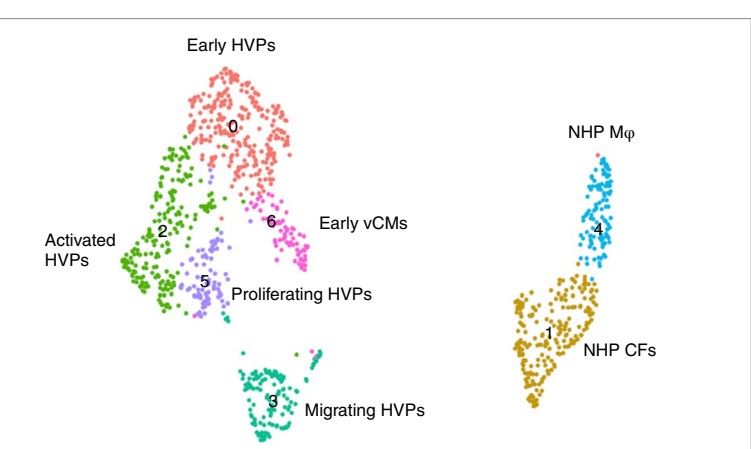

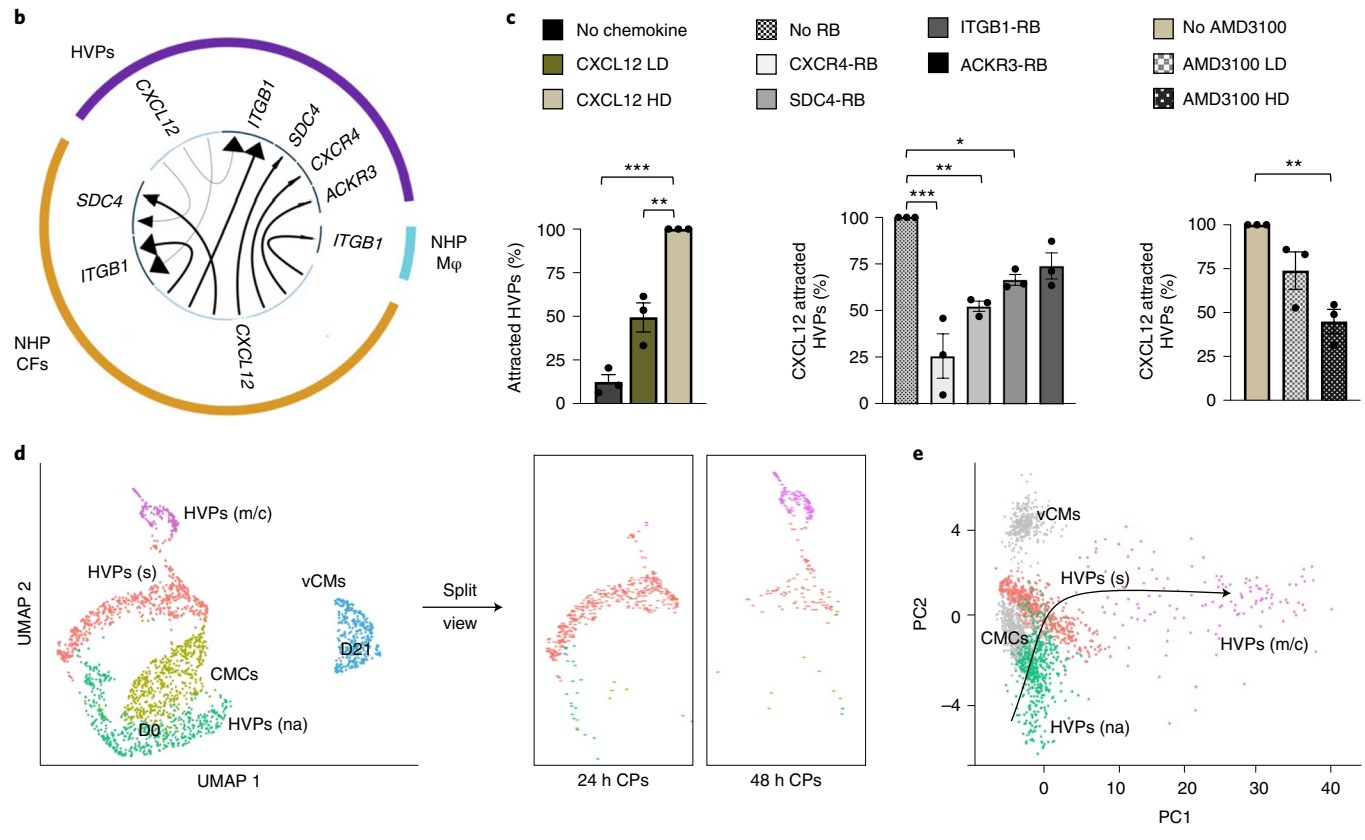

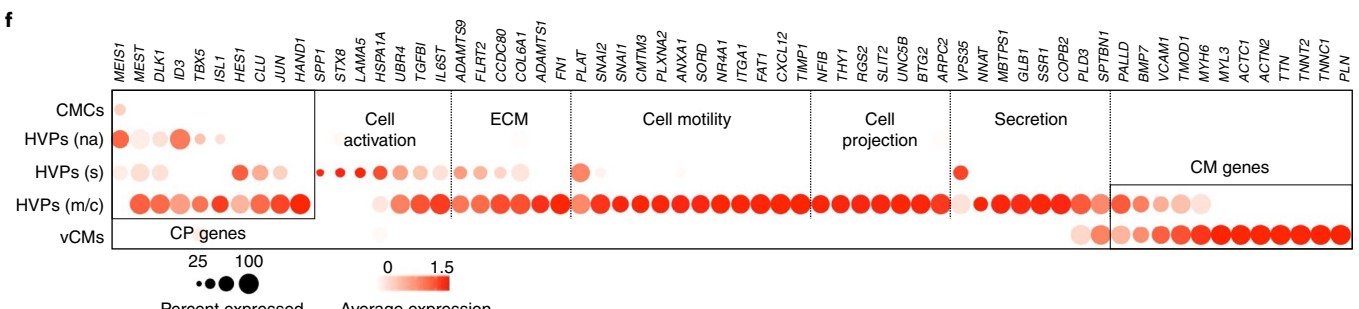

Percent expressed

Average expression

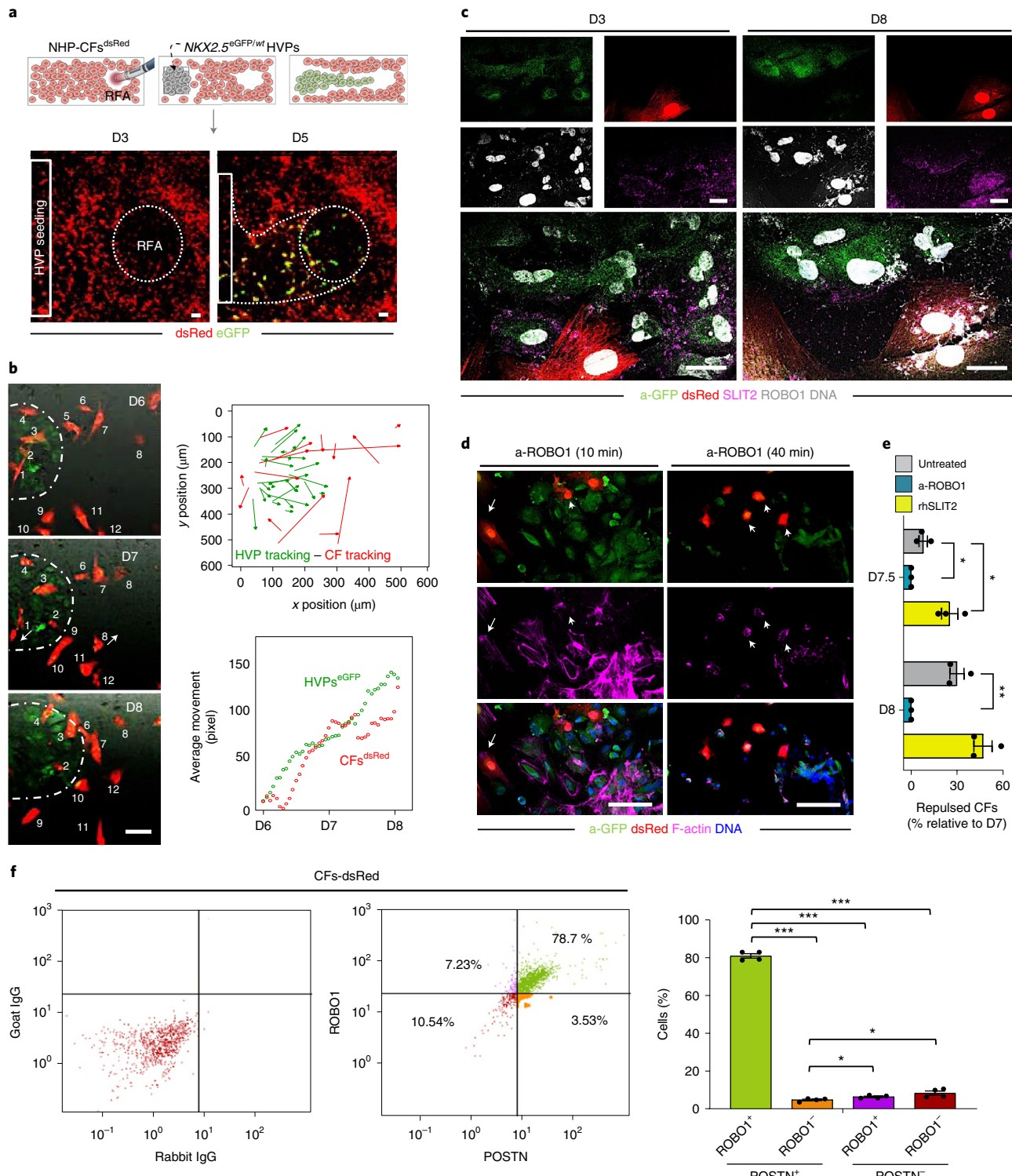

**Fig. 5 | SLIT2/ROBO1 signalling mediates activated CF repulsion and prevents myocardial scarring. a**, Top: schematic of 2D model for RFA injury of NHP CFs expressing dsRed followed by *NKX2-5*eGFP/wt HVP seeding and monitoring of co-culture. Bottom: sequential live imaging of dsRed+ and eGFP+ cells during migration. Scale bars, 200 μm. **b**, Left: representative time-lapse images of dsRed+ and eGFP+ cells at the RFA injury site during CF repulsion on indicated days. Dotted line delineates HVP migration front. Scale bar, 100 μm. The numbers indicate individual cells followed and tracked during the time lapse imaging. Right: cell tracking over time (top) and average movement (bottom) analysis of HVPs and CFs. **c**, Representative immunostaining for eGFP, SLIT2 and ROBO1 on D3 and D8. Scale bars, 25 μm. **d**, F-actin and eGFP immunofluorescence an D8 after ROBO1 antibody exposure for 10 and 40 min. Change of CF shape (arrow head) and F-actin localized on protrusion side of CFs (arrow). Scale bars, 75 μm. **e**, Percentage of repulsed CFs at the injured site analysed on D7.5 and D8 in standard condition (untreated) or after ROBO1 antibody and rhSLIT2 treatment on D7. Data are normalized to D7 and presented as mean ± s.e.m. and individual data points; *n* = 3 biological replicates per condition; *P < 0.05, **P < 0.005 versus untreated (*t*-test). **f**, Flow cytometry analysis for ROBO1 and POSTN in CFsdsRed after 8 days of co-culture with HVPs. Data are shown as mean ± s.e.m. and individual data points; *n* = 4 biological replicates per condition; *P < 0.05, ***P < 0.001 (*t*-test). For **e** and **f**, exact *P* values and numerical data are provided in Source Data Fig. 5.

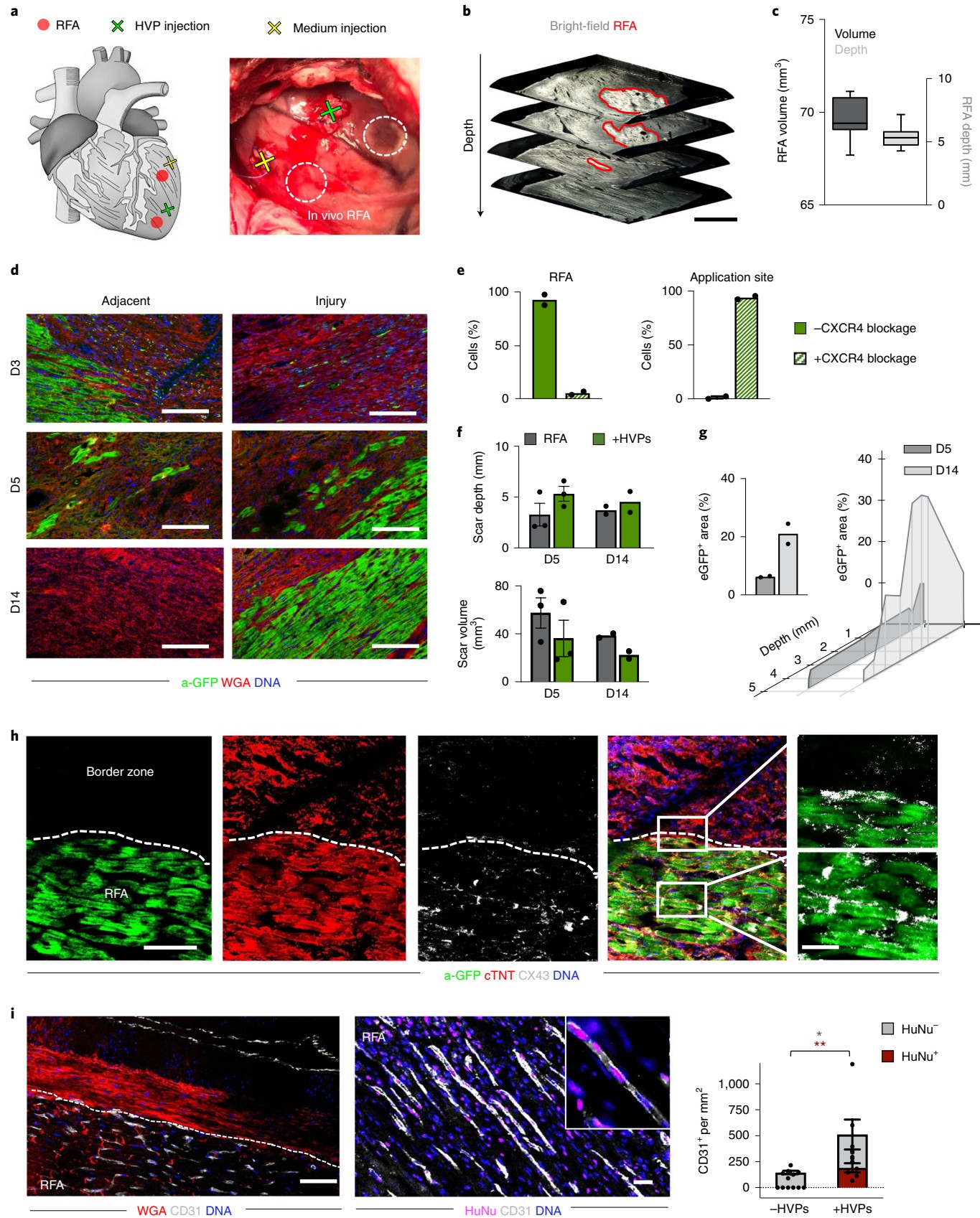

scar formation and fibrosis in the injury zone[2,21]. To unleash the full regenerative potential of cardiac cell therapy, it is essential to identify the cues that guide the recruitment of transplanted cells to target areas, modulate electrical integration and govern the cellular/molecular host–graft crosstalk. Our ex vivo model of HVPs and NHP heart tissue provides an unprecedented system to

**Fig. 6 | HVPs regenerate RFA-injured porcine myocardium in vivo. a**, Schematic of in vivo experimental design with two left ventricular RFA injuries and adjacent injection of HVPs or cell-free medium. **b**, Representative 3D reconstruction of non-transmural RFA injury. Scale bar, 2 mm. **c**, Statistical analysis of scar volume and depth of RFA injuries in freshly explanted wild-type pig hearts indicating standardized injury size. Box plot shows minimum, maximum, median and quartiles; $n = 3$ biological replicates. **d**, Representative fluorescence images of injury and adjacent sites after wheat germ agglutinin (WGA) and a-GFP co-staining on days D3, D5 and D14. Scale bars, 100 μm. **e**, Analysis of cells at application site and RFA in the presence or absence of pharmacological CXCR4 blockage (AMD3100) on day 5. Data are mean with individual data points; $n = 2$ biological replicates per condition. **f**, Analysis of in vivo scar depth and volume on D5 and D14 with or without HVP treatment. Data are mean ± s.e.m. with individual data points; $n = 3$ biological replicates per group on day 5, $n = 2$ biological replicates per group on day 14. **g**, Percentage of GFP+ area within the RFA injury (left) and according to depth of the cutting plane (right). Data are mean with individual data points; $n = 2$ biological replicates per group. **h**, Representative immunofluorescence images of RFA and border zone on D14 for anti-GFP, cTNT and CX43. Magnifications on the right correspond to the boxed area in the merged image. Scale bars, 50 μm and 10 μm (magnifications). **i**, Representative fluorescence images of HVP-treated RFA injury site after immunostaining for CD31 in combination with WGA (left) or with anti-human nuclei (HuNu, right). Scale bars, 50 μm (left), 25 μm (right). Bar graph shows the average number of CD31+ cells per mm² cells from host (HuNu⁻) and human HVPs (HuNu⁺) in HVP-treated and untreated RFAs. Data are presented as mean ± s.e.m. with individual data points; $n = 6$ biological replicates per group; $*P < 0.05$, $**P < 0.005$ (two-way ANOVA). For **c**, **e–g** and **i**, exact $P$ values and numerical data are provided in Source Data Fig. 6.

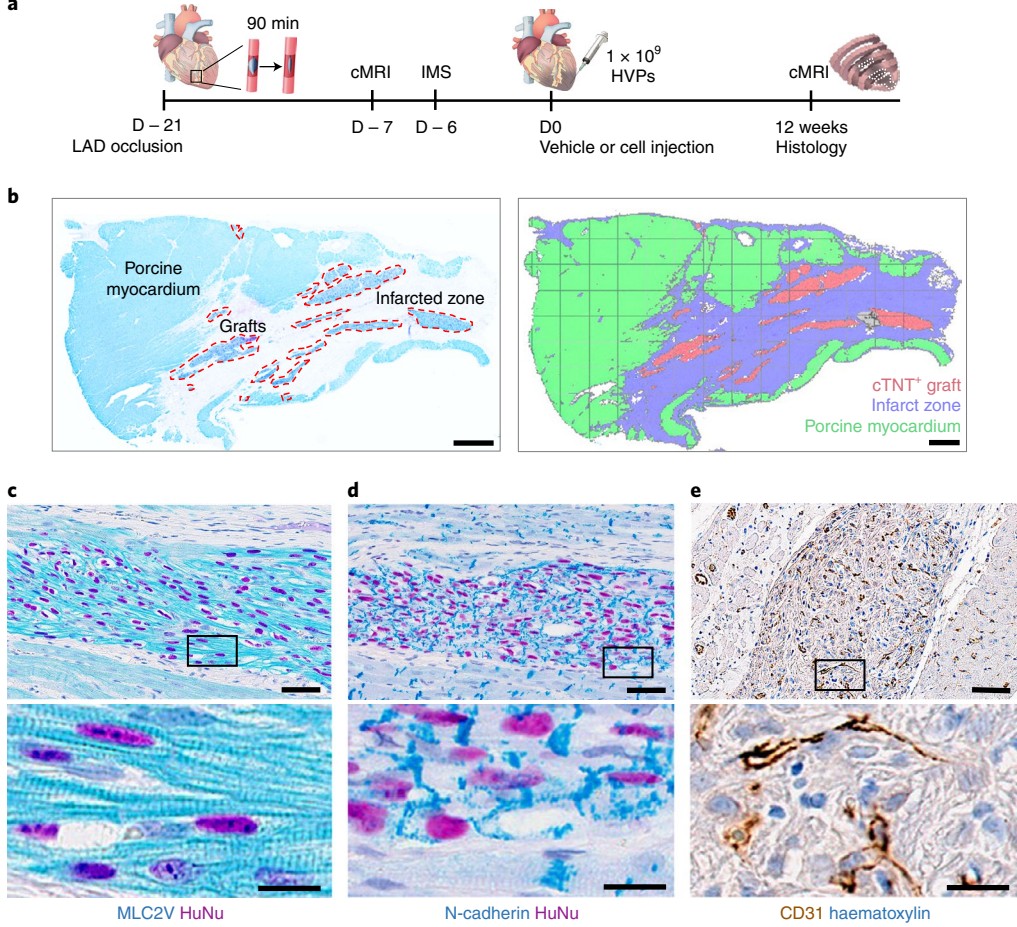

**Fig. 7 | HVPs remuscularize chronic scars in a porcine model of chronic ischaemia in vivo. a**, Schematic of in vivo experimental design of acute MI by balloon occlusion of the LAD coronary artery (ischaemia) and reperfusion after 90 min. Triple- immunosuppressive regimen (IMS) with cyclosporine (D − 6 to D84), methylprednisolone (D − 1 to D84) and abatacept (D − 1 to D84). Analysis of baseline infarct volume by cMRI on day −6 followed by epicardial cell injection (15 injection sites, total 1 × 10⁹ HVPs) into myocardial injury. Follow-up period of 12 weeks with cMRI scans at 12 weeks before termination and histological work-up. **b**, Overview of infarct zone and human grafts with labelling of porcine myocardium (HuNu⁻ cTNT⁺), infarct zone and cTNT⁺ graft (HuNu⁺ cTNT⁺). Scale bar, 2 mm. **c–e**, Immunohistochemistry of graft for cardiac ventricular muscle marker (MLC2v) (**c**), electrical coupling (N-cadherin) (**d**), and vessel formation (CD31) (**e**) at 12 weeks. Scale bar, 50 μm. Lower panels show magnifications of boxed areas. Scale bar, 15 μm.

refine molecular pathways implicated in cardiac regeneration at a single-cell resolution, thus offering an innovative approach. We demonstrated that HVPs harbour the unique potential to sense and counteract injury by re-activating sequential developmental programmes for directed migration, fibroblast repulsion and ultimate muscle differentiation within an injured heart (Fig. 8e). Future studies should investigate whether ex vivo human heart slices could predict outcome of cell-based regeneration in patients with

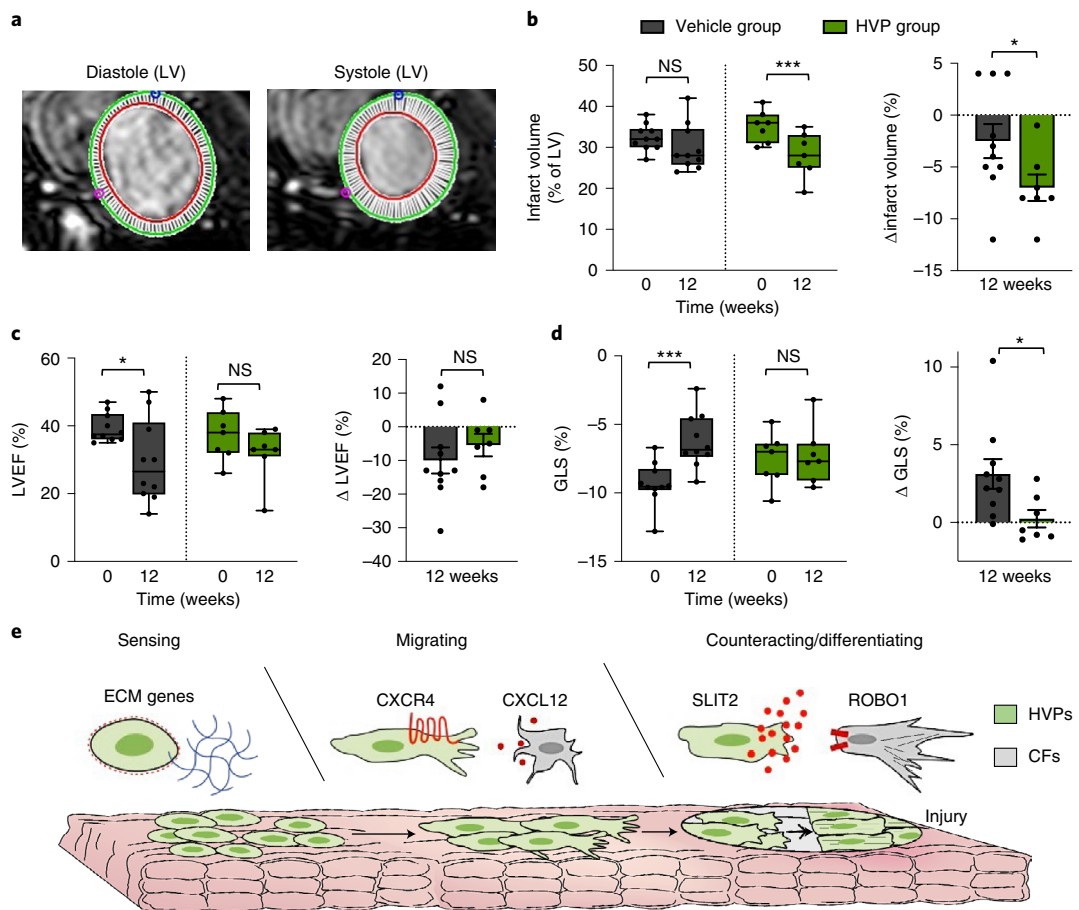

**Fig. 8 | HVPs preserve cardiac function in vivo. a**, Representative LV cMRI images of diastole and systole used for calculation of infarct volume, LVEF and GLS. **b**–**d**, Statistical analysis of infarct volume (**b**), LVEF (**c**) and GLS (**d**). Infarct volume, LVEF and GLS (%) are shown as minimum-to-maximum range with mean and individual data points; NS, not significant; *$P < 0.05$, ***$P \leq 0.001$ (two-way ANOVA). Delta values (Δ) are shown as mean ± s.e.m. and individual data points; *$P < 0.05$ (*t*-test); $n = 10$ pigs in the vehicle group and $n = 7$ pigs in the HVP-treated group. For **b**–**d**, exact *P* values and numerical data are provided in Source Data Fig. 8. **e**, Schematic summary of HVPs undergoing dynamic cellular states during cardiac tissue repair. HVPs sense tissue damage by activating programmes of ECM remodelling and migration and are chemoattracted to sites of cardiac injury via CXCR4/CXCL12 signalling. Counteraction to injury occurs via CF repulsion in a SLIT2/ROBO1-dependent manner and subsequent CM differentiation to remuscularize scar tissue.

different aetiologies of heart failure (for example, ischaemic, genetic and inflammatory).

Cell homing constitutes an indispensable step in repair processes of many organs[42,43]. We uncovered that the CXCL12/CXCR4 pathway mediates the inherent homing potential of HVPs, an ability lost once they entered the fully differentiated myocytic lineage. CXCL12 is implicated in migration of haematopoietic progenitors[44]. We demonstrate that HVPs utilize similar molecular pathways to facilitate homing and repopulation like the haematopoietic system.

Upon injury, activated CFs produce ECM components for tissue reconstructions while sending signals to neighbouring CMs and others for initiating reparative processes[45]. Our scRNA-seq unravelled that the SLIT2/ROBO1 axis mediates the HVPs' ability to repel CFs, thus reducing scarring. It will be of particular interest to evaluate whether such signalling pathway acts similarly in vivo and whether its pharmacological manipulation could circumvent cell application.

The study demonstrates rapid engraftment of HVPs with extensive de novo myocardium generation in porcine models of acute injury and chronic ischaemic heart failure. In both settings, scar volume was reduced and, in the latter, deterioration of cardiac function was prevented. This repairment cannot occur without sufficient blood supply. Increased neovascularization was documented in HVP grafts. Additional analysis is needed to demonstrate whether such

response is sufficient to restore physiological blood flow, since robust arterial input is crucial for permanent functional improvement. Moreover, before HVP transplantation can be translated clinically, one should determine whether HVP-based therapies could achieve higher remuscularization compared with CM transplantation with reduced ventricular arrhythmia and the use of hypo-immunogenic PSC lines can circumvent long-term rejection. Recently, the ESCORT trial performed first transplants of hPSC-derived cardiac progenitor patches in patients with ischaemic cardiomyopathy and reported no adverse effects including tumour formation[19]. Concordantly, after cardiac HVP injection, we did not detect tumourigenesis over the 3-month follow-up, demonstrating that MACS depletion of undifferentiated hESCs is safe for clinical translation.

In conclusion, our data indicate that HVPs harbour the unique capability to target both loss of myocardium and fibrotic scarring of the injured heart, supporting their therapeutic potential. Developing innovative therapeutic strategies rooted in fundamental biology of cardiac development could pave the way for successful cell-based cures of heart disease.

## Online content

Any methods, additional references, Nature Research reporting summaries, source data, extended data, supplementary information, acknowledgements, peer review information; details of

author contributions and competing interests; and statements of data and code availability are available at https://doi.org/10.1038/s41556-022-00899-8.

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

## Methods

**ESC maintenance, cardiac differentiation and HVP MACS-based purification.** Embryonic stem cell lines ES03 NKX2.5[eGFP/wt] and H9 NKX2.5[eGFP/wt] (ref. [46]) were generously gifted from Dr David Elliott (MCRI Australia), and maintained on Matrigel-coated plates (BD) in E8 (Gibco).

HVP differentiation was performed according to our previously published protocol[29]. ESCs were seeded on 12-well plates ($1 \times 10^6$ per well) in E8 with 5 µM Y-27632 (Torcis) for 24 h. On D0, RPMI/B27 minus insulin (Gibco) with 1 µM CHIR 98014 (Selleckchem) were added. Medium was changed to RPMI/B27 minus insulin ('CCM') after 24 h. On D3, 2 µM Wnt-C59 (Selleckchem) was applied and replaced by CCM on D5. On D6, HVPs were collected for MACS. After dissociation, cells were stained with Anti-TRA-1-60 MicroBeads (Miltenyi) before negative sorting with autoMACS Pro Separator (Miltenyi). Some cells were stained with TRA-1-60 antibody with Alexa488 (STEMCELL Technologies) and analysed by FACS using BD FACSCantoII. Batches with <5% TRA-1-60[+] cells were used for transplantation. For mature CMs, HVPs continued to be cultured in 12-well plates in RPMI/B27 containing insulin. Cells were cryopreserved in CryoStor (Sigma).

**Large-scale generation of HVPs for chronic ischaemia studies.** The method above was adapted to WA09 line (WiCell Research Institute) and scaled up using mTesR (STEMCELL Technologies).

**Ex vivo NHP heart slice culture.** For ex vivo heart slice cultivation, explanted NHP−LV tissue was placed in 2,3-butanedione 2-monoxime (Sigma-Aldrich) at 4 °C and shipped from German primate centre, Göttingen (reference 33.19-42502-04-16/2264), Karolinska Institutet, Sweden (reference N 277/14) or Walter Brendel Institute, Germany (reference ROB-55.2−2532.Vet_02-14-184). Within 24 h, tissues were sectioned on vibratome (Leica Biosystems) to 1 cm × 2 cm × 300 µm. Slices were anchored in biomimetic cultivation chambers (BMCC) with tissue adhesive (histoacryl; B. Braun) according to fibre direction and subjected to physiological preload of 1 mN and stimulation at 1 Hz, as previously described[30]. The BMCCs were anchored on a rocker in an incubator. Contractile force was continuously measured, and data were imported and analysed by LabChart Reader software (V8.1.14, AD Instruments).

**Generation of chimeric human–NHP heart constructs.** NHP-LV heart slices within BMCCs were underpinned with a hand-trimmed filter (0.40 µm). For homogeneous seeding, $2 \times 10^6$ HVPs were seeded onto the tissue within a pluronic F-127 (Sigma-Aldrich) frame using a bioprinting device (CANTER Bioprinter V4), equipped with a 0.58 mm standard Luer-Lock nozzle (Vieweg Dosiertechnik). For selective seeding, $0.5 \times 10^6$ HVPs or D25 CMs were used. Twelve hours after seeding, they were cultured in CCM with 5 µM ROCKi without rocking or pacing. After 12–24 h, rocking (60 rpm, 15° tilt angle) was resumed, and from D2, continuous pacing (1 Hz) and rocking.

**RFA injury.** NHP-LV heart slices seeded with HVPs or CMs were injured on the opposite tissue border 3 days after seeding by applying 20 W for 15 s using a THERMOCOOL SF uni-directional catheter, tip electrode 3.5 mm (Biosense Webster) and Stockert 70 radiofrequency generator (Biosense Webster). During the RFA procedure, physiologic preload was reduced to 0.5 mN and re-adjusted to 1 mN after 2 days. For in vivo experiments, epicardial RFA with 25 W for 7 s was performed to produce a standardized, non-transmural injury.

**Ca²⁺ imaging of RFA-injured heart slices.** NHP-LV heart slices seeded with HVPs/CMs after RFA injury were loaded with 3 µM Fluo-4-AM in CCM (without phenol red) with 0.75% Kolliphor EL (Sigma-Aldrich) incubation, washed and incubated again to allow de-esterification of the dye in Tyrode's solution supplemented with Ca²⁺. Fluo-4 intensity was imaged by DMI6000 B (Leica) with a Zyla V sCMOS camera (Andor Technology). Point stimulation electrodes were connected to an HSE stimulus generator (Hugo Sachs Elektronik) providing depolarizing pulses (40 V, 3 ms) at 1 Hz on the tissue border opposite the RFA injury. ImageJ ROI Manager was used for quantifications. Analysis was performed in RStudio (V1.2.5001).

**Cell isolation for scRNA-seq.** Samples were dissociated using 20 U ml⁻¹ papain[47] (Worthington Biochemical) for 20 min. Using FACSAria III (Becton Dickinson), cells were sorted according to eGFP+ and eGFP− cells onto 384-well plates containing Smart-Seq2 cell lysis buffer. The quality of the complementary DNA was confirmed using Agilent Bioanalyzer (Agilent) and RNA-seq libraries were prepared using in-house compatible Tn5 and Nextera index primers (Illumina).

**scRNA-seq and gene expression analysis.** scRNA-seq was performed at the Karolinska Institutet sequencing facility using the Genome Analyzer HiSeq2500 (Illumina) for single-end sequencing of 56 bp. The Genome Analyzer Analysis Pipeline (Illumina) was used to process the sequencing files of raw reads in the FASTQ format. The cDNA insert was aligned to the hg19/Mmul_1 reference genomes using Tophat2, combined with Bowtie2. Only confidently mapped and non-PCR duplicates were used to generate the gene-barcode cell matrix. Quality check, including the identification of highly variable genes, dimensionality reduction, standard unsupervised clustering algorithms and the DEG analysis were performed using the standard Seurat R pipeline[48].

**Cell clustering, UMAP visualization and marker-gene identification.** The gene-barcode matrix was scaled, normalized and log transformed. The dimensionality of the data was reduced by principal component analysis (PCA) (20 components) first and then with uniform manifold approximation and projection (UMAP) (resolution 0.3). Then, all cells from each cluster were sampled and DEGs across different clusters were identified with the FindAllMarkers and FindMarkers functions of Seurat R. Clusters were assigned to known cell types on the basis of cluster-specific markers (Source Data Figs. 2 and 4).

**Integrated analysis of single-cell datasets.** To integrate and validate the robustness of our analysis, two published single cell profiles were utilized[32,35]. To reduce batch-effect differences, Seurat alignment re-scaling and re-normalizing for the integrated dataset was used. For all new integrated datasets, we identified variable genes generating a new dimensional reduction that was used for further analysis. Pseudotemporal ordering was done using Monocle 2 (ref. [49]). An integrated gene-expression matrix was constructed as described above. With the function differentialGeneTest, we analysed DEGs across different development conditions. At maximum, the top 3,000 genes with the lowest $q$ value were used to construct the pseudotime trajectory.

**Determination of biological processes and molecular function on the basis of enrichment analysis.** Statistical analysis and visualization of gene sets were performed using the clusterProfiler R package[50]. Gene set enrichment analysis databases were used to determine the enrichment of biological processes, cellular components and molecular function on the basis of the genes that were significantly upregulated. Process-specific signatures were defined by the top genes as ranked by the significance and expression scores.

**NHP CF isolation and lentiviral transduction.** Explanted NHP LV was minced and incubated with 550 U ml⁻¹ collagenase II (Worthington) for dissociation. Isolated CFs were cultured in CF medium (DMEM−F12, 10% FBS, 2 mM L-glutamine and 0.5% penicillin–streptomycin).

For lentiviral transduction, a dsRed-expressing lentivirus was produced using a pRRLsin-18-PGK-d transfer plasmid combined with the packaging plasmid (pCMVdR8.74) and the envelope plasmid (pMD2.VSV.G) in HEK293T cells. The CFs were incubated with PGK-dsRed lentivirus and 8 µg ml⁻¹ of polybrene hexadimethrine bromide (Sigma-Aldrich), and the transduction efficiency was evaluated by dsRed expression after 96 h.

**Co-culture of HVPs and CFs for cell interaction studies after RFA injury.** NHP CF[dsRed] ($1 \times 10^4$ per well) were seeded in four-well chamber slides (Thermo Fisher) coated with fibronectin (Sigma-Aldrich). After 3 days, RFA injury (20 W, 7 s) was introduced on one end of the chamber slide followed by seeding of $5 \times 10^5$ HVPs on the opposite side. Cellular migration and interaction were studied by time-lapse microscopy. For the analysis of CF repulsion, anti-ROBO1 (5 µg ml⁻¹) and rhSLIT2 (2 µg ml⁻¹) treatments (Supplementary Table 1) were performed on D7 and D8 after RFA injury and HVP seeding. Videos were analysed with ImageJ for cell movement of HVPs and CFs by TrackMate plug-in[51].

**Identification of migratory signalling by trans-well assay.** For trans-well migration studies, the CytoSelectTM Cell Migration and Invasion Assay (Cell Biolabs) was used, and samples were processed according to the manufacturer's instructions. For this, $0.5 \times 10^6$ HVPs were suspended in serum-free medium and plated on the upper compartment of the trans-well migration assay (8 µm pore size). The chemoattractant factor CXCL12 was added (low dose, 20 ng ml⁻¹; high dose, 80 ng ml⁻¹) in the lower compartment. Agents that inhibit cell migration were added directly to the cell suspension in the upper compartment (CXCR4-RB (12 µg ml⁻¹), SDC4-RB (1:500), ITGB1-RB (8 µg ml⁻¹), ACKR-RB (10 µg ml⁻¹), AMD-3100 (low dose, 50 ng ml⁻¹; high dose, 100 ng ml⁻¹) (Supplementary Table 1). For quantification of migratory cells, medium was aspirated and all non-migratory cells within the insert were removed. Inserts were stained with Cell Stain Solution, and migratory cells were imaged with a minimum of three fields per insert.

**Secretome analysis.** Protease inhibitors (cOmplete Mini EDTA free, Thermo Fisher) were added to collected medium and supernatant concentration was achieved by ultrafiltration using protein concentrators with molecular weight cut-off of 10 kDa (Thermo Fisher). Proteins (50 µg) were diluted in 1% SDS, 50 mM dithiothreitol and 100 mM Tris buffer (pH 8.0), denatured and digested by filter-aided (Merck Millipore) sample preparation for proteome analysis[52]. Overnight digestion was performed and peptides were recovered. An aliquot corresponding to 10% of the digest volume was purified by strong cation exchange StageTips[53]. The eluate was diluted and injected for nanoscale liquid chromatography with tandem mass spectrometry (LC–MS/MS) analysis. Tryptic peptides were analysed by nanoscale LC–MS/MS by a top-12 data-dependent analysis method run on a Q-Exactive 'classic' instrument (Thermo Fisher)[54] with a gradient length of 85 min. Raw data were loaded in MaxQuant (V1.6.2.6a)

for database search and label-free quantification by the MaxLFQ algorithm (Cox Mann, MCP 2014). For data processing, default parameters in MaxQuant were adopted, except for the following: LFQ min. ratio count: 2; fast LFQ: ON; quantification on unique peptides; match between runs: activated between technical replicates, not between different samples (achieved by assigning a separate sample group to each biological sample and allowing match between runs for each group only). For protein identification, MS/MS data were queried using the Andromeda search engine implemented in MaxQuant against the *Homo sapiens* reference proteome (accessed in December 2019, 74,788 sequences) and the *Macaca fascicularis* reference proteome (accessed in January 2020, 46,259 sequences).

**RNA isolation, RT–PCR and qRT–PCR.** Total RNA of CF/HVP co-culture and conditioned CFs exposed to co-culture medium was extracted using the Absolutely RNA Miniprep Kit (Agilent Technologies) according to the manufacturer's instructions and 1 μg was reverse transcribed using the High-Capacity cDNA Reverse Transcription Kit (Thermo Fisher). qRT–PCR was performed using 25 ng cDNA per reaction and Power SYBR Green PCR Master Mix (Thermo Fisher). Gene expression levels were assessed in three biological samples and normalized to *GAPDH* expression using the Ct method (ΔCt). The following primers were used: GAPDH_Fw: TCCTCTGACTTCAACAGCGA; GAPDH_Rv: GGGTCTTACTCCTTGGAGGC; ROBO1_Fw: GGGGGAGAGAGAGTGGAGAC; ROBO1_Rv: AGGCTCTCCTACTGCAACCA; SLIT2_Fw: TAGTGCTGGCGA TCCTGAA; SLIT2_Rv: GCTCCTCTTTCAATGGTGCT.

**Flow cytometry.** For CF characterization, co-culture of HVPs and CF-dsRed on D8 was dissociated, fixed with 4% PFA and permeabilized. Primary antibodies against desired epitopes (Supplementary Table 1) were incubated at the indicated dilutions, followed by secondary antibodies (1:300). For proliferation quantification, samples were incubated with 10 μM EdU, dissociated with 480 U ml⁻¹ collagenase type II, fixed with 4% PFA and processed using the Click-iT EdU594 Flow Cytometry Assay Kit (Thermo Fisher) according to manufacturer's instructions. For baseline expression of Isl-1, cTNT and Tra-1-60, human ES cells were differentiated into ventricular progenitor cells, in vitro. On D6 and D25, cells were dissociated with accutase before staining for desired epitopes according to manufacturer's instructions. Data were acquired with a Gallios flow cytometer (Beckman Coulter) and evaluated with Kaluza software V1.2 (Beckman Coulter) or with FACS Canto III and FloJo software V8 (BD Biosciences). Gating strategies are provided in Supplementary Information.

**Porcine studies with RFA injury.** LEA29Y pigs were sedated by intramuscular injection of ketamine (Ursotamin), azaperone (Stresnil) and atropinsulfate (B. Braun) and in full anaesthesia with mechanical ventilation by continuous intravenous application of propofol (Narcofol) and fentanyl (Fentadon). Following left lateral thoracotomy, pericardium was opened, with the anterior wall of LV exposed. After induction of RFA, 6 × 10⁷ HVPs (<5% TRA-1-60⁺) were thawed in CCM with ROCKi and cell pellet was transplanted approximately 1 cm from the injury, using an insulin syringe (BD Micro Fine 30 G). U stitches with pericardial patches were placed and closed using a tourniquet (Supplementary Video 2).

All pigs had a central venous catheter (Careflow) over the course of the study. Additional immunosuppression, 5 mg kg⁻¹ methylprednisolone was applied intravenously on D1 and 2.5 mg kg⁻¹ on D2 (Urbason). For migration blockage, animals received 1 mg kg⁻¹ KG AMD3100 twice a day from D1 to D4. Pigs were killed under anaesthesia with systemic injection of potassium chloride (B. Braun).

Explanted porcine hearts on D3, D5 and D14 after RFA injury and HVP injection were examined for tumour formation with corresponding HVP or control injection sites and fixed with 4% PFA, followed by cryopreservation with ice-cold methanol or acetone and sectioning at 12 μm in optimal cutting temperature compound. All experiments were performed with permission from the local regulatory authority. The Regierung von Oberbayern, Sachgebiet 54, 80534 München approved the studies (AZ 02-18-134). Applications were reviewed by the ethics committee according to §15 TSchG German Animal Welfare Law.

**Porcine experiments for chronic ischaemia studies.** Male Yucatan minipigs (28–40 kg) were from Sinclair Research Center, USA. Following anaesthesia, a small incision was made over the carotid artery and jugular vein, a small opening was made in the artery and a sheath was introduced. Heparin (250–350 IU kg⁻¹) was administered to maintain an activated clotting time twice the baseline activated clotting time level. A guide catheter was advanced into the ostium of the LAD. A balloon catheter was introduced by advancing it through the guide catheter to the LAD and placed below the first diagonal branch of the LAD and inflated to occlude the artery for 90 min. It was then deflated to re-perfuse the ischaemic area. All handling and treatments were performed according to the appropriate animal welfare standards and in compliance with regulatory guidelines (IACUC 1974-061) in Charles River, Mattawan (MI, USA).

Echocardiography was performed under anaesthesia 2 weeks after MI. Animals with LVEF ≤ 45% were randomized into vehicle or HVP (1 × 10⁹) groups. For immunosuppression, a combination of cyclosporin A, methylprednisolone and abatacept was used[18]. From D − 6 to D84, cyclosporin A (20–30 mg kg⁻¹) was given orally twice a day, except for D1. On the morning of D1, cyclosporin A dose was

10 mg kg⁻¹ intravenous infusion. From D − 1, methylprednisolone (128 mg d⁻¹) was given until D3, and subsequently tapered to 80 mg d⁻¹ (D4 to D6); 48 mg d⁻¹ (D7 to D9) and 24 mg d⁻¹ from D10 for the remainder of the study. Abatacept (12.5 mg kg⁻¹) was given as 30 min intravenous infusion every second week from D − 1. Blood samples for monitoring cyclosporin A levels were collected, ranging from 100 μg l⁻¹ to 1,000 μg l⁻¹, except immediately after the cyclosporin A infusion (Extended Data Fig. 8d).

**Transplantation of HVPs into porcine hearts after MI.** Three weeks after MI, an incision was made over the femoral artery and vein under anaesthesia and an opening in the artery and a sheath was introduced to monitor blood pressure. A midline incision was made over the sternum, and the skin and underlying musculature was retracted. The pericardium was incised, and sutures were passed through and attached to the chest wall to form a sling. Glass beads were used to indicate injection locations. Vehicle or HVPs (n = 10 vehicle and n = 7 HVP) were injected 300 μl per site, using a 30 G needle, in the LV myocardium: eight injections in the border zone and seven injections in the infarcted area. After final injection, the heart was returned to the pericardial sling and sutures were removed.

A board-certified pathologist investigated the grafts for presence of teratoma. All HVP-derived grafts were negative for teratoma at 3 months. Lung, liver, heart, kidney, spleen, brain, thyroid, adrenal glands, pituitary, prostate and lymph nodes were collected. Human haemoglobin B was not detected by Quantitect Multiplex RT–PCR with probe HS00758889_s1 HBB FAM and no signs of HVP-derived cells outside the heart were present.

**Cardiac magnetic resonance imaging.** cMRI was performed under anaesthesia, using a 1.5 T MRI scanner (Philips Intera platform R12 software), 1 week before and 12 weeks after surgery. Short- and long-axis images with a 1 cm interval between slices were obtained. Data were analysed with Circle Cardiovascular Imaging cvi42 (V5.12). Three-dimensional (3D) volumes were calculated as the sum of (area × (slice thickness + distance between slices)) for all short-axis slices. Ejection fraction was calculated as 100 × (end diastolic volume − end systolic volume)/end diastolic volume. Late gadolinium enhancement MRI was used to quantify infarct size as percentage volume (%). GLS was also analysed. Animals were killed after 12 weeks. Hearts were removed, perfused with lactated Ringer's solution. LV containing all the injection sites was fixed in formalin for 48 ± 12 h, then transferred to 70% ethanol and paraffin embedded.

**Immunohistochemistry and quantification of the scar after MI.** Paraffin-embedded hearts were sectioned into 4 μm slices. Immunohistochemistry methods and protocols were set up on an automated Ventana Discovery Ultra autostainer (V12.5.4; Roche). Immunohistochemistry for detection of desired epitopes (Supplementary Table 1) was carried out according to the manufacturer's recommendation and all reagents except antibodies were Ventana products (Roche). Antigen retrieval was done before primary antibody was added, followed by antibody block and secondary anti-rabbit reagent or anti-mouse horseradish peroxidase, purple teal or DAB chromogenic detection in single or double staining. The slides were digitized using a Aperio XT whole slide scanner and Aperio ImageScope software (V12.3.3.5048). Quantification was performed using Visiopharm software (V2020.08.03.9090).

**Immunofluorescence analysis.** Cells on chamber slides were fixed with 4% PFA. Co-culture 3D constructs were fixed in 4% PFA, cryopreserved with ice-cold methanol and sectioned at 12 μm in optimal cutting temperature compound (Sakura Finetek, JP). Primary antibodies against desired epitopes (Supplementary Table 1) were incubated overnight at 4 °C at the indicated dilutions, followed by secondary antibodies (Supplementary Table 1) and Hoechst 33258 (5 μg ml⁻¹). Images were acquired using a DMI6000-AF6000, thunder or SP8 confocal laser-scanning Leica microscope. Images were assigned with pseudocolours and processed with ImageJ (V1.53a). Quantification of scar volumes, eGFP⁺ area and cell-type proportions was performed with ImageJ cell counter and volume calculator.

**Statistics and reproducibility.** Statistical analyses were performed using GraphPad Prism (V9) or R program (V3.5.1). Data are presented as mean ± s.e.m. unless otherwise indicated. Two groups were compared using Welch's *t*-test. Three groups or repeated measures were analysed using one-way or multiple-way analysis of variance (ANOVA). Summarized views on data that underlie the statistical tests, as well as the exact *P* values are available in Source data. The statistical details for each experiment are also provided in the figure legends. Most experiments were performed independently at least three times. For the chronic injury model, power calculation was performed with historic in-house data and data distribution determined by QQ plot analysis. For all other experiments, no statistical methods were used to pre-determine sample size. Data distribution was assumed to be normal and individual data points are presented in all graphs. In the porcine model, animals were evaluated on the basis of ejection fraction after MI, and subsequently randomized to vehicle- or HVP-treatment groups. cMRI was analysed blinded by two independent cardiologists. Ex vivo experiments were randomly assigned to experimental and control groups. Data collection was not performed blinded owing to the experimental conditions.

**Reporting Summary.** Further information on research design is available in the Nature Research Reporting Summary linked to this article.

## Data availability

All sequencing data that support the findings of this study can be found at Gene Expression Omnibus under the accession number GSE153282. The mass spectrometry data have been deposited to the ProteomeXchange Consortium via the PRIDE database with the dataset identifier PXD019521. The GTF and FASTA files used for bioinformatics analysis (GRCh37 and Mmul_1, Ensembl Releases 75 and 105, respectively) can be downloaded from Illumina iGenomes (https://emea.support.illumina.com/sequencing/sequencing_software/igenome.html). All other data supporting the findings of this study are available from the corresponding authors on request. Source data are provided with this paper.

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

## Acknowledgements

We thank R. Hinkel (Deutsches Primatenzentrum Göttingen), M. Spångberg, B. Eriksson, A. Fagreaus and P. Ekeland (Karolinska Institute), as well as J.-M. Abicht and M. Längin (Walter Brendel Centre of Experimental Medicine, LMU) for providing NHP hearts, D. Elliott for his generosity of the two NKX2.5-eGFP cell lines, A. Goedel, R. Yang and X. Li for FACS and bioinformatics assistance, J. Sohlmér and E. Rohner for HVP production, Gubra for performing the light sheet imaging of pig cardiac tissue, and S. Dietzel (Core Facility Bioimaging, Biomedical Center, LMU) for two-photon live microscopy. We acknowledge B. Campbell and C. Scherb for their technical support in cell culture, P. Henderson, G. Linhardt, F. Nunes, P. Rodrigues, M. Persson, R. Hicks, L.-M. Gan, C. Graneli, J. Eriksson, T. Incitti, P. Saxena, M. Jansson, T. Marlow and Y. Wang for the assistance and valuable discussions on in vivo experiments. We thank S. Luger (moments-of-aha.com) for all graphic illustrations, S. Panula, D. Topcic, N. Rahkonen, M. Kurek, J. Xu, A. Harisankar, C. Gavin, E. Kerr and B. Bossen (Procella) for their valuable discussions on the adaptation and scale-up production of HVP cells. This work was supported by grants from: European Research Council (ERC) under the European Union's Horizon 2020 research and innovation programme to K.R.C. (grant agreement number 743225), to A. Moretti (grant agreement number 788381) and to C.K. (grant agreement number 101021043); the German Research Foundation, Transregio Research Unit 152 (to A. Moretti and K.-L.L.) and 267 (to A. Moretti, K.-L.L. and C.K.); Swedish Research Council Distinguish Professor Grant (to K.R.C.); German Centre for Cardiovascular Research (DZHK). AstraZeneca would like to dedicate this publication to the memory of Dr Frank Seeliger who was the board certified pathologist responsible for the histological and pathological evaluation of the chronic pig study.

## Author contributions

K.-L.L., A. Moretti, R.F.-D. and K.R.C. together set up the collaboration and conceived the overall experimental plan. C.M.P. and M.T.D.A. performed functional experiments and histochemistry, analysed data and generated figures. K.S.F. and M.L.L. produced HVPs and CMs, performed MACS sorting and analysed data. C.M.P., M.T.D.A. and K.S.F contributed to the conception and design of experiments. G.S. performed bioinformatics analyses. F.R. and P.H. established RFA and measured scar parameters. T.D. and D.Z. conducted FACS analysis. T.D., I.M. and A. Meier. produced heart slices and performed cellular seeding. Y.L.T. contributed to bioinformatics. S. Schwarz and S. Sudhopp developed cellular bioprinting. R.T., S.H., K.L. and A.D. introduced and adapted biomimetic slice culture. D.S. conducted calcium imaging and analysis and analysed time lapse. E.P., M.G. and G.C. executed mass spectrometry. A.B., N.H., M.K. and C.K. performed in vivo pig experiments. M.K. injected HVPs in vivo. V.J. performed CD68 immunodetection and analysis. N.K. and E.W. generated and provided transgenic LEA29Y pigs. N.K., C.K. and A.B. supervised in vivo studies and provided conceptual advice. K.J. and Q.-D.W. contributed to the overall conception and design of the chronic efficacy study in pigs and interpretation of data. M.F. designed safety endpoints and performed safety evaluation. D.H. contributed to dose setting and monitored and adjusted the immunosuppression over the study period. M.C. supervised the cMRI scan and performed data analysis. M.S. performed immunohistochemistry, analysed histology data and generated images. J.H., J.C and R.B. contributed to development of the large-scale culture system of HVP cells and supervised cell production for the chronic pig study. E.E. edited the text. A. Moretti, K.R.C., R.F.-D. and K.-L.L. conceived and supervised this study and provided financial support. A. Moretti, K.R.C. and K.-L.L. wrote the manuscript. All authors commented on and edited the manuscript.

## Competing interests

J.C. and R.B. are employees of Procella Therapeutics. K.J., Q.-D.W., R.F.-D., J.H., D.H., M.F. and M.S. are employees of AstraZeneca. K.S.F. and K.R.C. are co-inventors on a patent (Patent no. 10508263) based on the HVP technology and its applications. The HVP intellectual property is assigned to Procella Therapeutics. A.D. holds a patent (Patent no. USN 15/781,454) on the technology of biomimetic cultivation and is co-founder and shareholder of InVitroSys GmbH. The other authors declare no competing interests.

## Additional information

**Extended data** is available for this paper at https://doi.org/10.1038/s41556-022-00899-8.

**Correspondence and requests for materials** should be addressed to Christian Kupatt, Regina Fritsche-Danielson, Alessandra Moretti, Kenneth R. Chien or Karl-Ludwig Laugwitz.

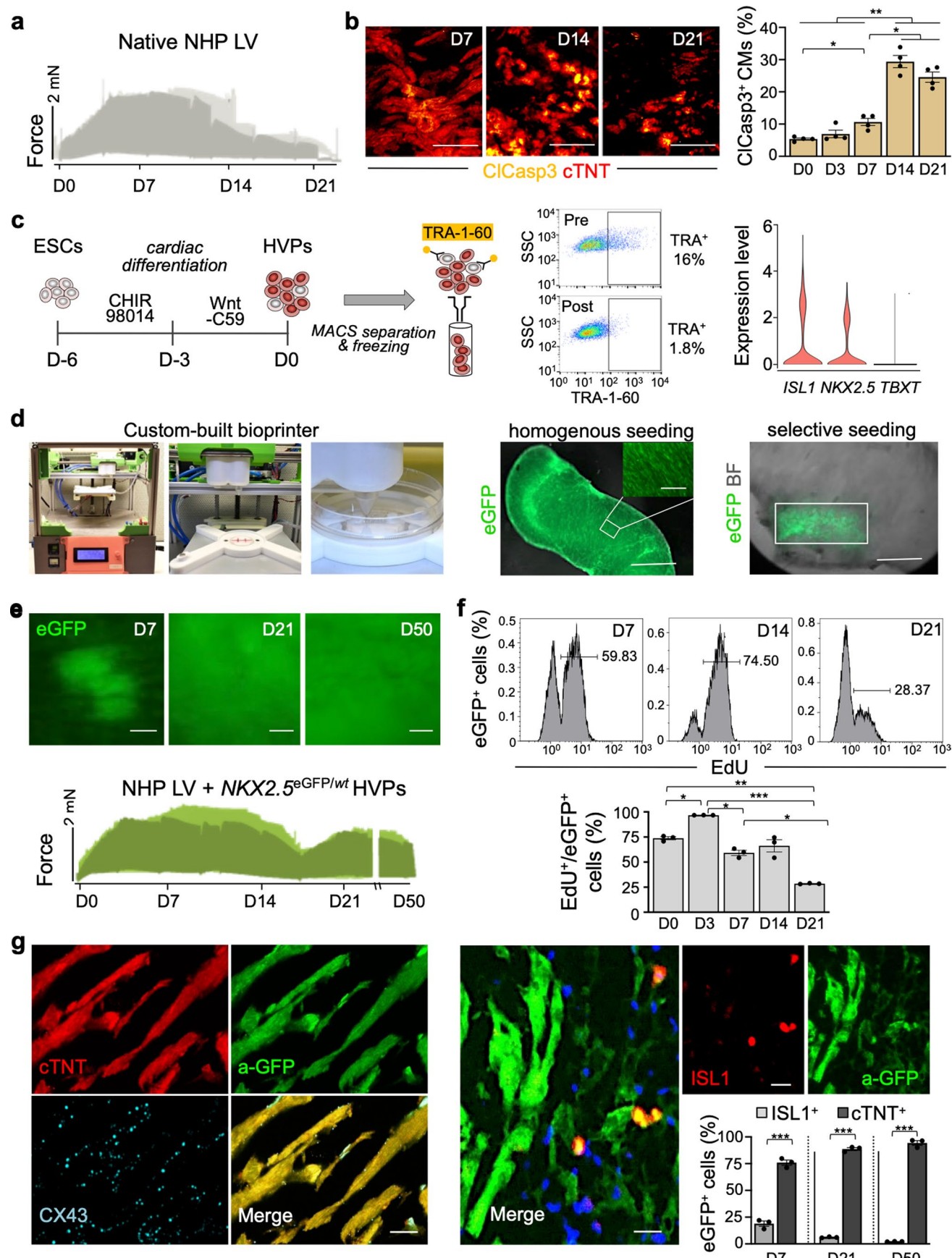

**Extended Data Fig. 1 | See next page for caption.**

**Extended Data Fig. 1 | Generation and analysis of an *ex-vivo* 3D chimeric human-NHP heart model. a**, Representative, overlapped traces of contractile force of native NHP heart slices cultured exvivo for 21 days in biomimetic chambers. **b**, Representative immunofluorescence images for activated cleaved caspase 3 (ClCasp3) and cardiac troponin T (cTNT) in ex vivo cultured NHP heart slices (left) and correspondent quantification (right) at the indicated days. Scale bars 50 μm. Data are shown as mean ± SEM and individual data points, n = 4 biological replicates per time point, *p < 0.05, **p < 0.01 (one-way ANOVA). **c**, Schematic of ESC differentiation into HVPs by Wnt pathway modulation followed by MACS depletion of Tra-1-60+ cells and cryopreservation until seeding. Single cell RNAseq confirmed expression of *ISL1* and *NKX2.5* and loss of brachyury T (*TBXT*) on D0. **d**, Left, custom-built bioprinting device with pneumatic printhead. Right, exemplary images of homogeneous or selective seeding of eGFP+ HVPs onto NHP heart slices by bioprinting. Scale bars 250 μm, inlet 75 μm. **e**, Live eGFP imaging of NHP heart slices after *NKX2-5*[eGFP/wt] HVP seeding at the indicated days of co-culture (top) and representative contractile force traces (bottom). Scale bars 200 μm. **f**, Flow cytometry analysis for EdU in eGFP+ cells isolated at the indicated days of co-culture. Data are shown as mean ± SEM and individual data points, n = 3 biological replicates per time point, *p < 0.05, **p < 0.005, ***p < 0.001 (one-way ANOVA). **g**, Immunostaining of eGFP in combination with cTNT and Connexin-43 (CX43) (left) or ISL1 and a-GFP (right) on D50 of co-culture. Scale bars 25 μm. Bar graph shows the percentage of eGFP+ cells expressing ISL1 and cTNT on the indicated days of co-culture as mean ± SEM and individual data points, n = 3 biological replicates per time point, ***p < 0.001 (two-way ANOVA). For **b, f** and **g**, exact p-values and numerical data are provided in Source Data Extended Data Fig. 1.

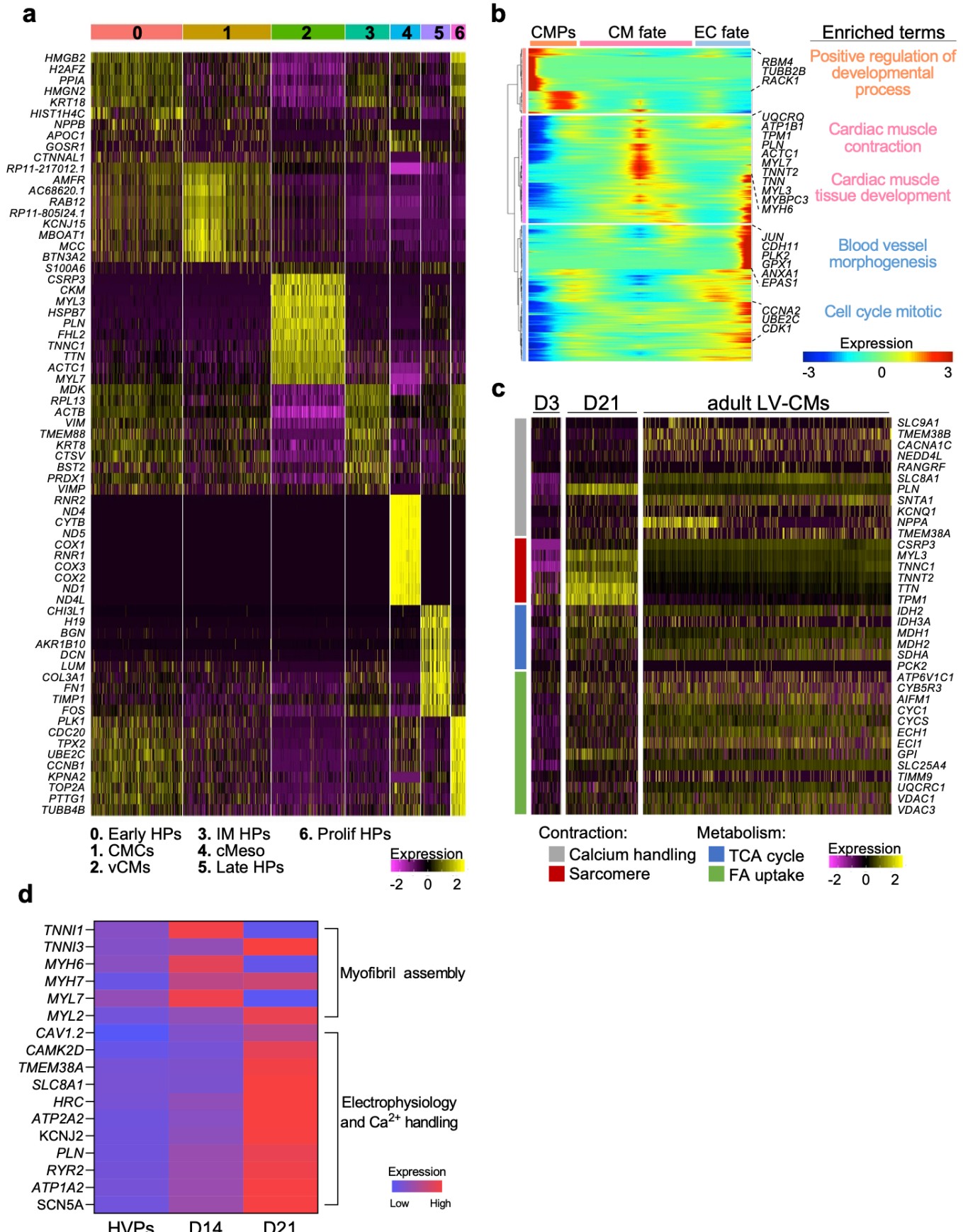

**Extended Data Fig. 2 | See next page for caption.**

**Extended Data Fig. 2 | scRNAseq analysis of human *NKX2.5*^*eGFP/wt* HPs in a chronic injury model of NHP heart slices. a**, Heatmap showing expression of the top 10 genes in each cluster defined as 0- early heart progenitors (Early HPs), 1- cardiac mesenchymal cells (CMCs), 2- ventricular cardiomyocytes (vCMs), 3- intermediate heart progenitors (IM HPs), 4- cardiac mesodermal cells (cMeso), 5- late heart progenitors (Late HPs), 6- proliferating heart progenitors (Prolif HPs). **b**, Heatmap of different blocks of DEGs along the pseudotime trajectory and representative genes in each cluster. Cardiac mesodermal precursors (CMPs, D-3), endothelial cell (EC) fate (D0 and D3) and CM fate (D21). Selected top biological process and canonical pathway terms related to corresponding DEGs. **c**, Heatmap showing the expression of genes related to contraction (gray and red) and metabolism (blue and green) in eGFP+ cells on D3 and D21 of ex-vivo co-culture compared to adult human LV-CMs (Wang *et al.*, 2020). Expression levels are presented as a colour code. **d**, Progressive analysis of myofibril assembly, calcium handling and electrophysiology signatures by quantitative RT-PCR, n = 3 biological replicates. Expression levels are presented colour coded. For **a-c**, single cells have been isolated from 3 biological replicates. Data from all replicates are included. Exact p-values and numerical data are provided in Source Data Extended Data Fig. 2.

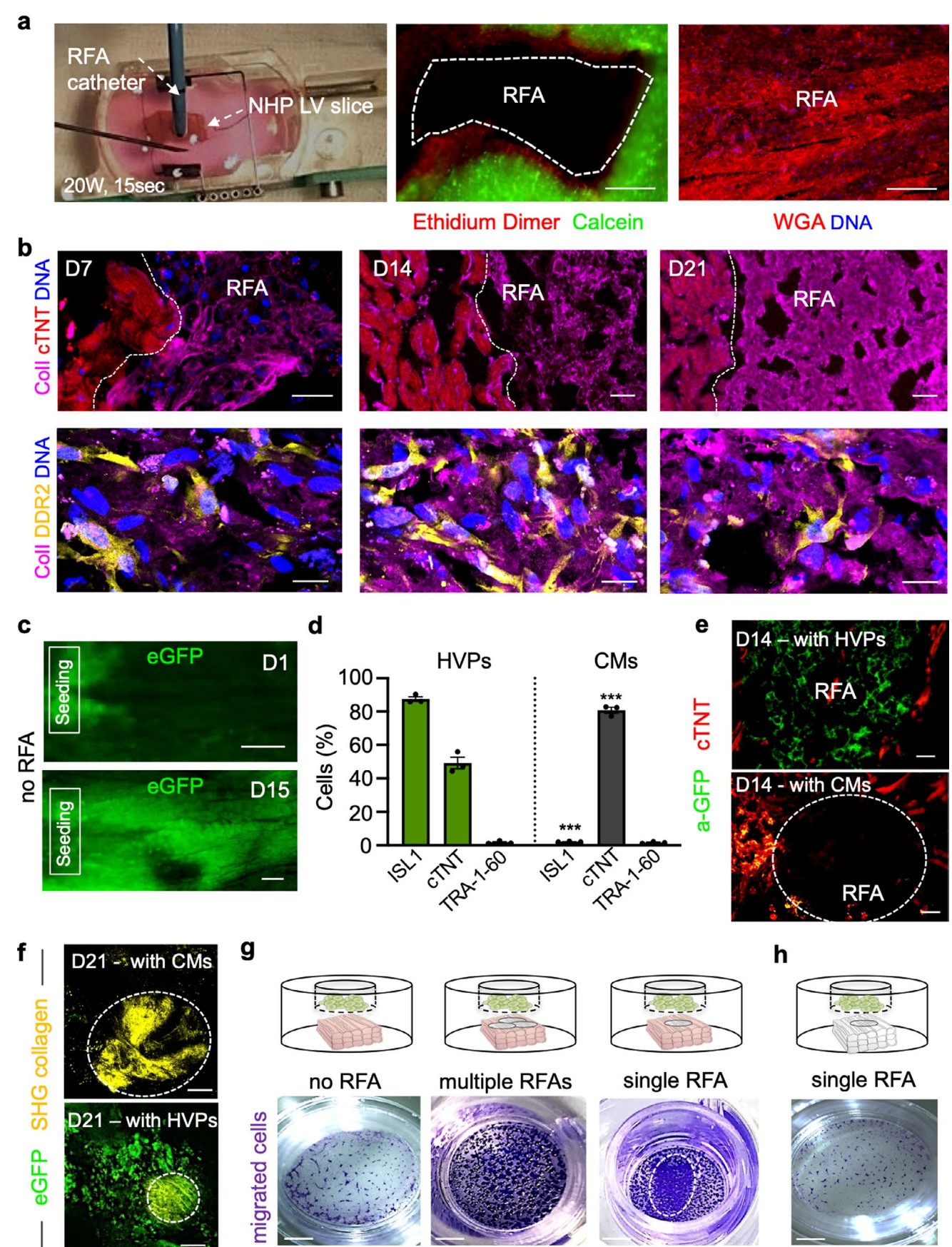

**Extended Data Fig. 3 | See next page for caption.**

**Extended Data Fig. 3 | Generation and analysis of an acute ex-vivo NHP heart injury model. a**, Standardized non-transmural myocardial injury in NHP heart slices by defined RFA. Live and dead cells are stained by calcein and ethidium dimer, respectively (middle) Scale bar 200 μm. Intact ECM scaffold after RFA injury shown by WGA immunostaining. Scale bar 200 μm. Stainings were performed immediately after RFA. **b**, Representative fluorescence images of RFA-injured slices after immunostaining for Collagen type I (ColI) combined with cTNT (top) or DDR2 (bottom) on indicated days. Lower panels show images of the RFA area. Scale bars 30 μm (top) and 25 μm (bottom). **c**, Sequential live imaging of *NKX2-5*$^{eGFP/wt}$ HVPs migrating from the seeding frame into the tissue showing homogenous repopulation of the slice by D15 in the absence of RFA injury. Scale bars 200 μm. **d**, FACS analysis of HVP and CM batches before seeding indicating cellular purity (ISL-1, cTNT and TRA-1-60). Data are mean ± SEM and individual data points, n = 3 biological replicates per group, ***p < 0.001 *vs* HVPs (two-way ANOVA). **e**, Representative immunostainings of eGFP and cTNT in RFA-injured area on D14 after selective seeding of *NKX2-5*$^{eGFP/wt}$ HVPs (left) or CMs (right). Scale bars 50 μm. **f**, Two-photon live microscopy of RFA-injured slices for eGFP and second-harmonic-imaging (SHG) visualization of collagen and scar size on D21. Circles demarcate areas with collagen deposition. Scale bars 100 μm. **g**, Trans-well migration assays with D0 *NKX2-5*$^{eGFP/wt}$ HVPs in the upper and NHP heart slice in the lower compartment, respectively. Images show trans-well migrated HVPs on polycarbonate membrane in the absence (left) or presence of multiple (middle) or single (right) RFA injury. Dashed line marks the site of HVP accumulation. Scale bars 2 mm. **h**, Trans-well migration assays with D0 *NKX2-5*$^{eGFP/wt}$ HVPs in the upper and decellularized NHP heart slice in the lower compartment with a single RFA injury. Scale bar 2 mm. For **d**, exact p-values and numerical data are provided in Source Data Extended Data Fig. 3.

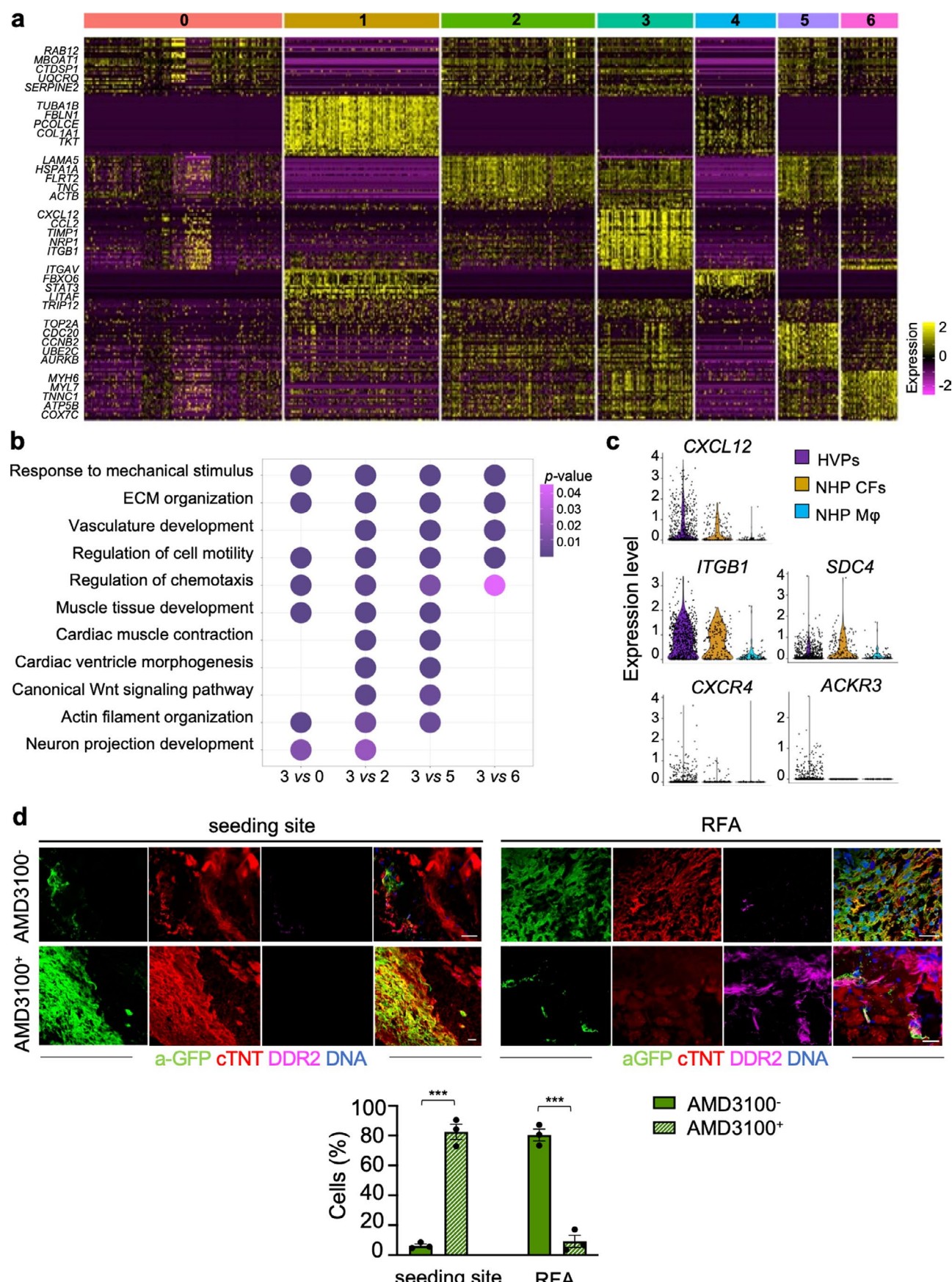

**Extended Data Fig. 4 | See next page for caption.**

**Extended Data Fig. 4 | scRNAseq analysis of human *NKX2-5*[eGFP/wt] HVPs and NHP cardiac cells after acute RFA heart injury. a**, Heatmap of top 50 genes in each cluster with representative genes indicated. 0. Early HVPs; 1. NHP CFs; 2. Activated HVPs; 3. Migrating HVPs; 4. NHP Mφ; 5. Proliferating HVPs; 6. Early vCMs. HVPs, human ventricular progenitors; NHP, non-human primate; CFs, cardiac fibroblasts; Mφ, macrophages; vCMs, ventricular cardiomyocytes. **b**, Representative GO terms upregulated in cluster 3 (migrating HVPs) compared to the other human clusters (0, 2, 5, 6). **c**, Violine plots of *CXCL12* and its binding targets identified by italk analysis of scRNA sequencing of migrating HVPs, NHP CFs and NHP Mφ. **d**, Representative immunostainings of a-GFP, cardiac troponin T (cTNT) and DDR2 on D15 after injury at seeding site and RFA in the absence of CXCR4 blockage (AMD3100[-]) or in the presence of CXCR4 blockage (AMD3100[+]). Scale bars 25 μm. Statistical data are shown as mean ± SEM and individual data points, n = 3 biological replicates per condition, *** p < 0.001 (two-way ANOVA). For **a-c** single cells have been isolated from 3 biological replicates. Data from all replicates are included. Exact p-values and numerical data are provided in Source Data Extended Data Fig. 4.

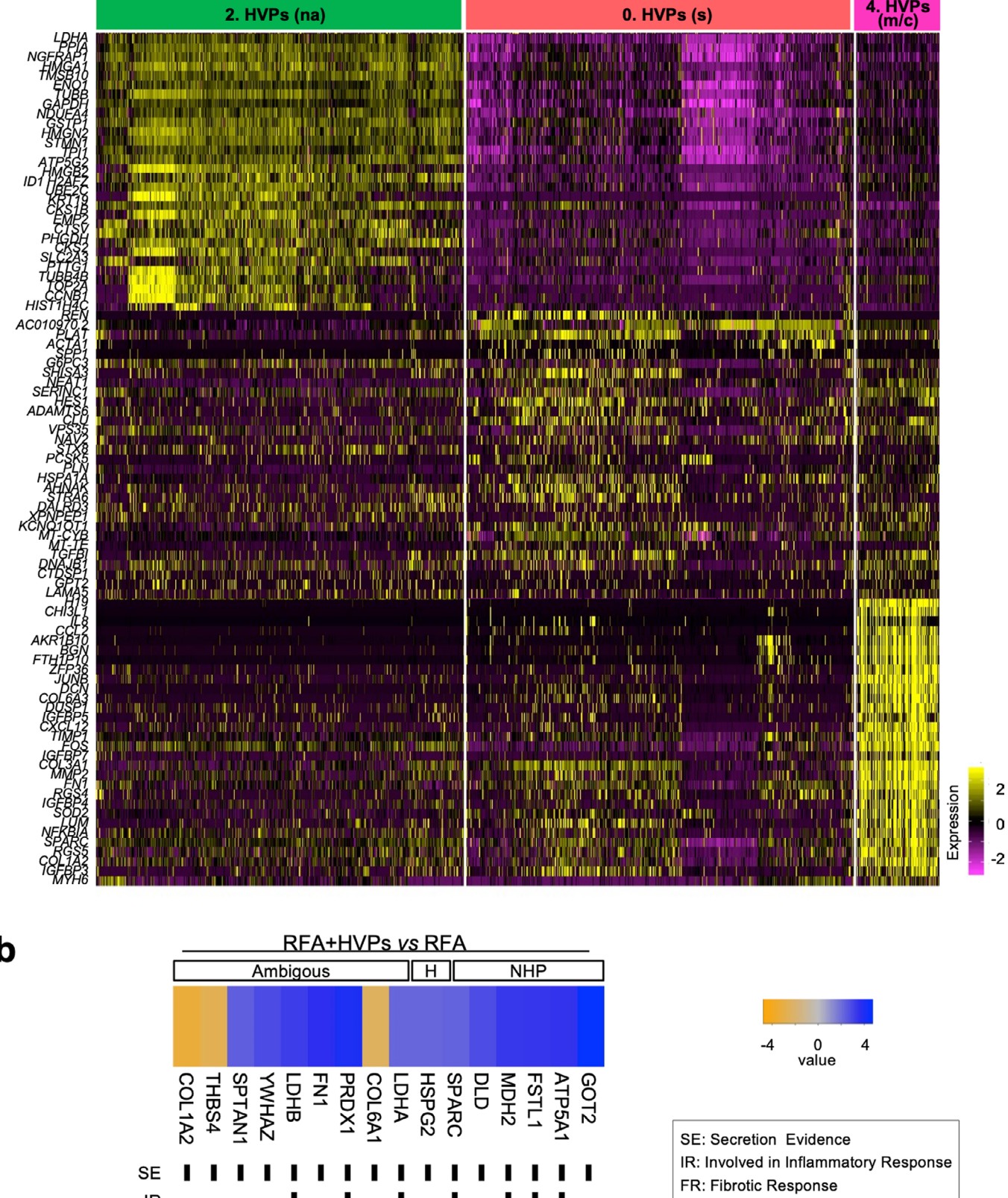

**a**

**b**

RFA+HVPs *vs* RFA

Ambigous | H | NHP

SE: Secretion Evidence
IR: Involved in Inflammatory Response
FR: Fibrotic Response

Pro | Anti

**Extended Data Fig. 5 | See next page for caption.**

**Extended Data Fig. 5 | Gene signatures of dynamical cardiac progenitor states and proteomic analysis of secretome during acute injury response.** Heatmap of top 30 genes depicting the expression of DEGs in non-activated HVPs (cluster 2), sensing HVPs (cluster 0), and migrating/counteracting HVPs (cluster 4). **b**, Proteomic analysis of supernatant of injured NHP heart slices with and without application of HVPs at 48 h after RFA. NHP, H, and ambiguous, was assigned to proteins for which the majority of identified peptides belonged to protein sequences of Macaca Fascicularis, Homo Sapiens, or both species, respectively. n = 3 biological replicates per group, p-value ≤0.05. Data from all replicates are included. For **a** and **b**, numerical data are provided in Source Data Extended Data Fig. 5.

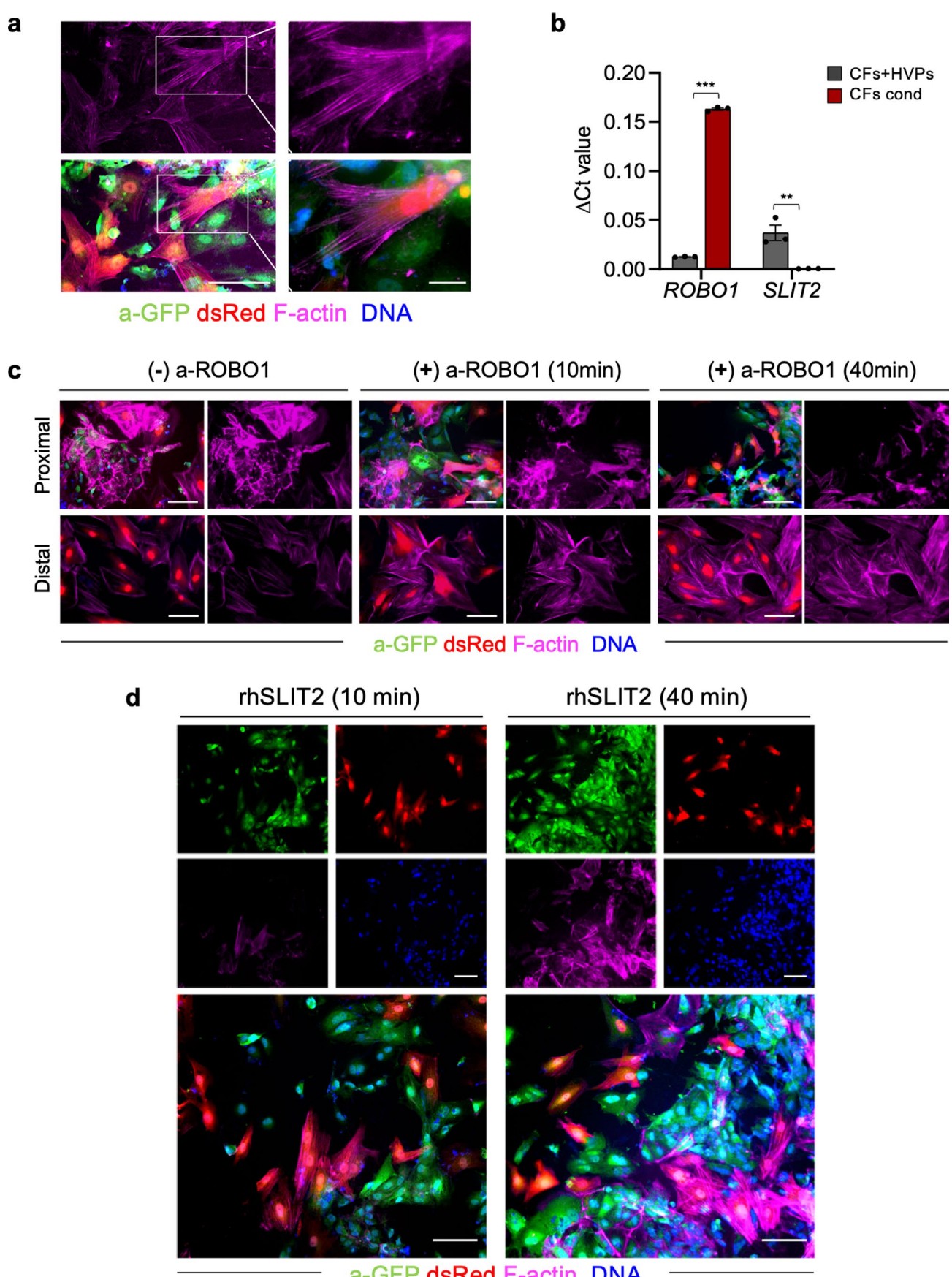

**Extended Data Fig. 6 | See next page for caption.**

**Extended Data Fig. 6 | Analysis of CF repulsion signalling during acute injury response in 2D monolayer. a**, Representative images of eGFP[+] HVPs and dsRed[+] CFs after F-actin staining during the repulsion phase in the injury area on D8. **b**, Quantitative RT-PCR analysis of *ROBO1* and *SLIT2* expression in injured CFs cultured with HVPs (CFs + HVPs) or alone in conditioned medium from HVP−CF co-culture (CFs cond) on D8. Data are mean ± SEM and individual data points, n = 3 biological replicates per condition, **$p < 0.01$ ***$p < 0.001$ (t-test). **c**, Representative F-actin immunostaining on D8 in standard condition and after ROBO1 antibody exposure for 10 and 40 minutes showing CFs in contact with HVPs (proximal) and CFs in the remote area from the injury site (distal). **d**, Immunodetection of eGFP in conjunction with Phalloidin (F-actin) stain in HVP−CF co-culture on D8 after recombinant human SLIT2 (rhSLIT2) exposure for 10 and 40 minutes. Nuclei were counterstained with Hoechst and CFs are labelled with dsRed (**a, c, d**). Scale bars 75 μm (**a, c, d**). For **b**, exact p-values and numerical data are provided in Source Data Extended Data Fig. 6.

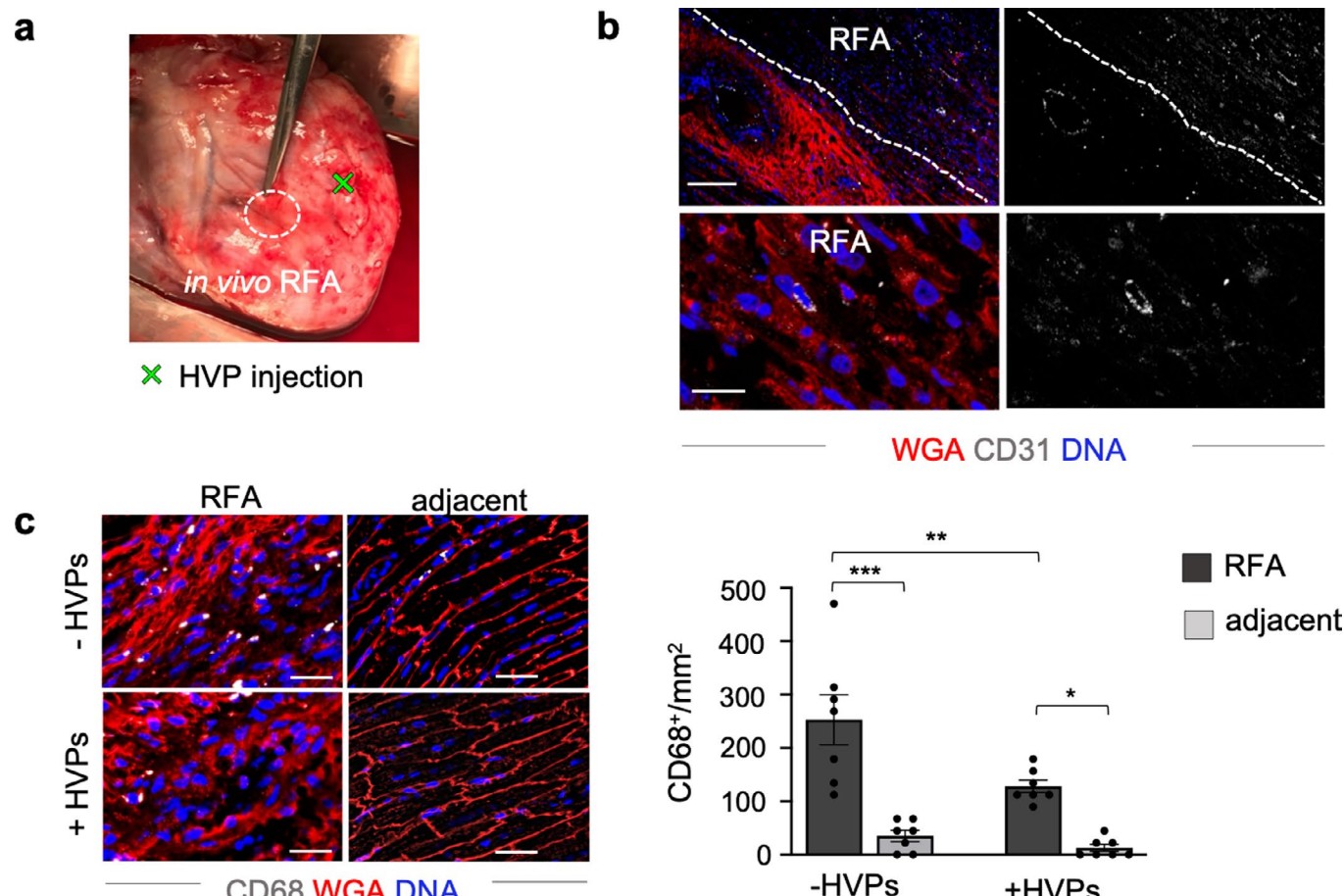

**Extended Data Fig. 7 | Immune-cell analyses of LEA29Y pig hearts after RFA injury and HVP injection in-vivo. a**, Image of a freshly explanted LEA29Y pig heart 14 days after in-vivo RFA and adjacent HVP injection showing no macroscopic signs of teratoma formation. **b**, Representative fluorescence images of control RFA and adjacent area (top) or magnified zoom of control RFA (bottom) after CD31 immunodetection and WGA labelling. Scale bars 100 μm (top) and 10 μm (bottom). **c**, Representative immunofluorescence stainings of CD68 (left) and correspondent statistical analysis (right) of RFA and adjacent areas in the presence or absence of HVPs. Scale bars 25 μm. Data are shown as mean ± SEM and individual data points, n = 7 biological replicates per group, *p < 0.05, **p < 0.01, ***p < 0.001 (one-way ANOVA). Exact p-values and numerical data are provided in Source Data Extended Data Fig. 7.

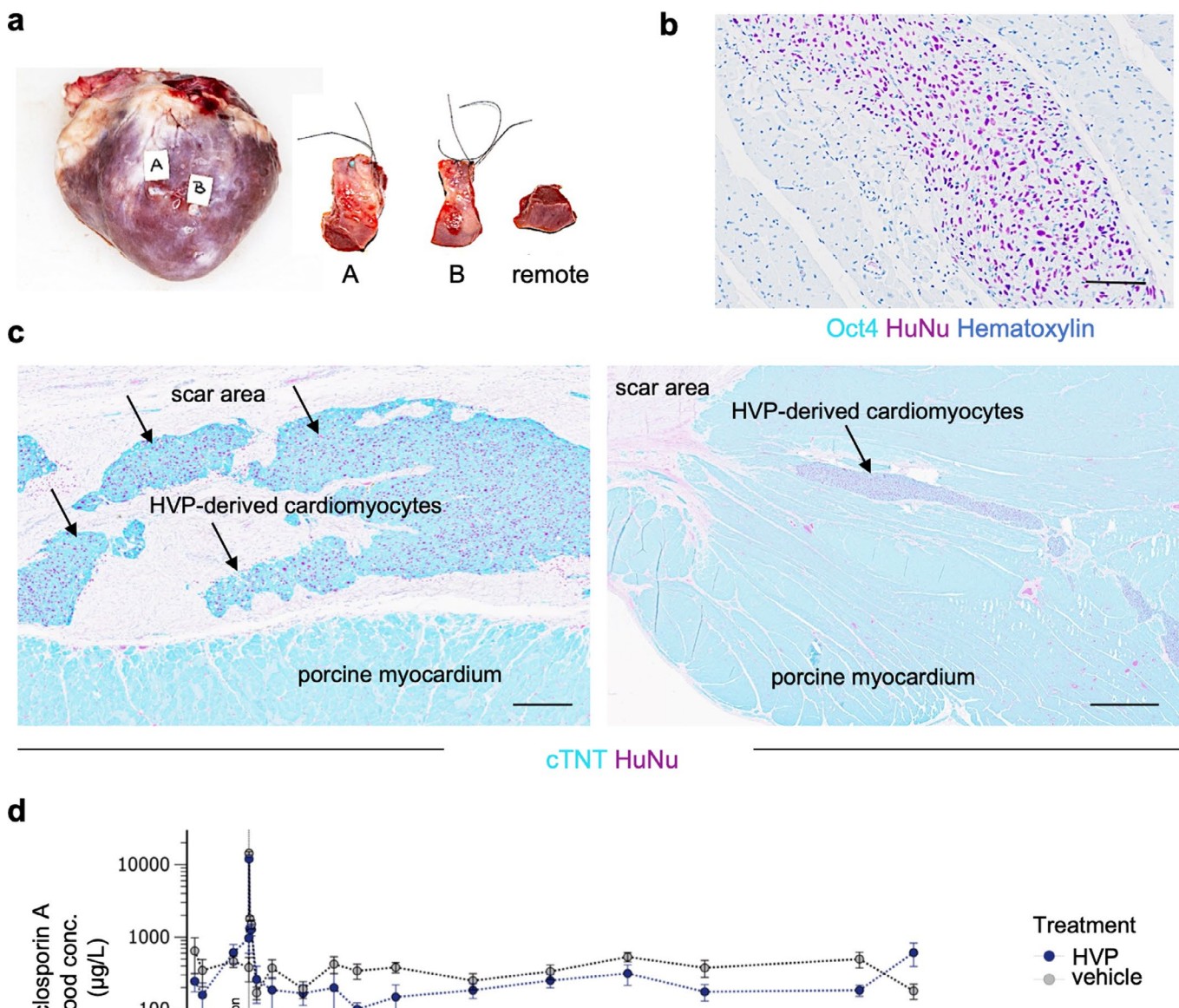

**Extended Data Fig. 8 | Macro- and microscopic analyses of pig hearts after chronic myocardial ischemia and HVP-injection. a**, Image of freshly explanted pig heart 84 days after in-vivo HVP-injection indicating injection sites (A, B), showing no macroscopic signs of teratoma. **b**, Negative immunostaining for OCT4 confirms absence of undifferentiated pluripotent stem cells in the human graft after 12 weeks. Scale bar 50 μm.
**c**, Representative immunohistochemistry images of scar (left) and border area (right) after HVP-injection with marked human nuclei (HuNu) and cTNT. Scale bars 250 μm (left) and 1mm (right). **d**, Quantification of Cyclosporin A concentrations (immunosuppression) in whole blood of HVP- and vehicle-treated animals performed regularly throughout follow up period. Data are shown as mean ± SEM. n = 10 pigs in vehicle treated group and n = 7 pigs in HVP-treated group. Numerical data are provided in Source Data Extended Data Fig. 8.

Christian Kupatt
Regina Fritsche-Danielson
Alessandra Moretti
Ken Chien

# Reporting Summary

Nature Research wishes to improve the reproducibility of the work that we publish. This form provides structure for consistency and transparency in reporting. For further information on Nature Research policies, see our Editorial Policies and the Editorial Policy Checklist.

## Statistics

For all statistical analyses, confirm that the following items are present in the figure legend, table legend, main text, or Methods section.

| n/a | Confirmed | |
|---|---|---|
| ☐ | ☒ | The exact sample size (*n*) for each experimental group/condition, given as a discrete number and unit of measurement |
| ☐ | ☒ | A statement on whether measurements were taken from distinct samples or whether the same sample was measured repeatedly |
| ☐ | ☒ | The statistical test(s) used AND whether they are one- or two-sided *Only common tests should be described solely by name; describe more complex techniques in the Methods section.* |
| ☒ | ☐ | A description of all covariates tested |
| ☐ | ☒ | A description of any assumptions or corrections, such as tests of normality and adjustment for multiple comparisons |
| ☐ | ☒ | A full description of the statistical parameters including central tendency (e.g. means) or other basic estimates (e.g. regression coefficient) AND variation (e.g. standard deviation) or associated estimates of uncertainty (e.g. confidence intervals) |
| ☐ | ☒ | For null hypothesis testing, the test statistic (e.g. *F*, *t*, *r*) with confidence intervals, effect sizes, degrees of freedom and *P* value noted *Give P values as exact values whenever suitable.* |
| ☒ | ☐ | For Bayesian analysis, information on the choice of priors and Markov chain Monte Carlo settings |
| ☒ | ☐ | For hierarchical and complex designs, identification of the appropriate level for tests and full reporting of outcomes |
| ☐ | ☒ | Estimates of effect sizes (e.g. Cohen's *d*, Pearson's *r*), indicating how they were calculated |

*Our web collection on statistics for biologists contains articles on many of the points above.*

## Software and code

Policy information about availability of computer code

| Data collection | Leica application suite, Ventana Discovery Ultra autostainer V12.5.4, 1.5 Tesla Philips Intera platform R12 software |
|---|---|
| Data analysis | Kaluza V1.2, FloJo V8, , FACSAria III, Prism V9, ImageJ V1.53a, MaxQuant V1.6.2.6a, Aperio Imagescope V12.3.3.5048, MyoDish software V1, MyoDish Data File Converter V1.0, LabChart Reader V8.1.14, Rstudio Version 1.1.453, Visiopharm Software V2020.08.03.9090, Circle Cardiovascular Imaging cvi42 (V5.12) <br> RNA-seq analysis was performed in Tophat V2.1.1, Bowtie V2.4.5, R (V3.5.1), using Seurat R pipeline (Satija, R., Farrell, J. A., Gennert, D., Schier, A. F. & Regev, A. Spatial reconstruction of single-cell gene expression data. Nat. Biotechnol. 33, 495-502, doi:10.1038/nbt.3192 (2015), Pseudotemporal ordering was done using Monocle 2 (Qiu, X. et al. Single-cell mRNA quantification and differential analysis with Census. Nat Methods 14, 309-315, doi:10.1038/nmeth.4150 (2017). <br> Statistical analysis and visualization of gene sets were performed using the clusterProfiler R package (Yu, G., Wang, L. G., Han, Y. & He, Q. Y. clusterProfiler: an R package for comparing biological themes among gene clusters. OMICS 16, 284-287, doi:10.1089/omi.2011.0118 (2012) |

For manuscripts utilizing custom algorithms or software that are central to the research but not yet described in published literature, software must be made available to editors and reviewers. We strongly encourage code deposition in a community repository (e.g. GitHub). See the Nature Research guidelines for submitting code & software for further information.

## Data

Policy information about availability of data

All manuscripts must include a data availability statement. This statement should provide the following information, where applicable:

- Accession codes, unique identifiers, or web links for publicly available datasets
- A list of figures that have associated raw data
- A description of any restrictions on data availability

All sequencing data that support the findings of this study can be found at Gene Expression Omnibus under the accession number GSE153282.
The mass spectrometry data have been deposited to the ProteomeXchange Consortium via the PRIDE database with the dataset identifier PXD019521.
The GTF and FASTA files used for Bioinformatics analysis (GRCh37 and Mmul_1, Ensembl Releases 75 and 105, respectively) can be downloaded from Illumina iGenomes ( https://emea.support.illumina.com/sequencing/sequencing_software/igenome.html).
All other data supporting the findings of this study are available from the corresponding author on reasonable request. Source data for all figures are provided with this paper.

# Field-specific reporting

Please select the one below that is the best fit for your research. If you are not sure, read the appropriate sections before making your selection.

☒ Life sciences          ☐ Behavioural & social sciences          ☐ Ecological, evolutionary & environmental sciences

For a reference copy of the document with all sections, see nature.com/documents/nr-reporting-summary-flat.pdf

# Life sciences study design

All studies must disclose on these points even when the disclosure is negative.

| | |
|---|---|
| Sample size | The exact sample size for each experiment is provided in the figures, legends and the text.<br>For the chronic injury model, power calculation was performed with historic in-house data and data distribution determined by QQ plot analysis. For all other experiments, no statistical methods were used to pre-determine sample size. Data distribution was assumed to be normal and individual data points are presented in all graphs. |
| Data exclusions | No data were excluded. |
| Replication | All attempts for replication were successful. Unless indicated, at least 3 independant experiments were performed. |
| Randomization | In the porcine model, animals were evaluated based on ejection fraction after MI, and subsequently randomized to vehicle- or HVP-treatment groups. Ex vivo experiments were randomly assigned to experimental and control groups. |
| Blinding | cMRI was analysed blinded by two independent cardiologists.<br>In ex vivo experiments, data collection was not performed blinded due to the experimental conditions. |

# Reporting for specific materials, systems and methods

We require information from authors about some types of materials, experimental systems and methods used in many studies. Here, indicate whether each material, system or method listed is relevant to your study. If you are not sure if a list item applies to your research, read the appropriate section before selecting a response.

### Materials & experimental systems

| n/a | Involved in the study |
|---|---|
| ☐ | ☒ Antibodies |
| ☐ | ☒ Eukaryotic cell lines |
| ☒ | ☐ Palaeontology and archaeology |
| ☐ | ☒ Animals and other organisms |
| ☒ | ☐ Human research participants |
| ☒ | ☐ Clinical data |
| ☒ | ☐ Dual use research of concern |

### Methods

| n/a | Involved in the study |
|---|---|
| ☒ | ☐ ChIP-seq |
| ☐ | ☒ Flow cytometry |
| ☒ | ☐ MRI-based neuroimaging |

## Antibodies

| | |
|---|---|
| Antibodies used | Primary antibodies<br>Anti-a-actinin, rabbit polyclonal  Abcam ab137346 1:300 (IHC)<br>Anti-ACKR3, rabbit polyclonal LSBio LS-A1893 10 µg/ml (migration assay)<br>Anti-cardiac Troponin I, recombinant Abcam ab52862 1:500 (IHC) |

Anti-Cardiac Troponin T, mouse monoclonal Thermo Fisher Scientific MA5-12960, Cl. 13-11 1:500 (IF)
Anti-cardiac Troponin T, rabbit polyclonal Sigma-Aldrich HPA015774 1:300 (IHC)
Anti-Cardiac Troponin T, rabbit polyclonal Abcam ab45932 1:400 (IF)
Anti-CD31 (PECAM-1), sheep polyclonal R&D systems AF806 1:100 (IF)
Anti-CD31, rabbit polyclonal Novus NB100-2284 1:50 (IHC)
Anti-CD68, mouse monoclonal eBioscience  14-0688-82, Cl. KP1 1:100 (IF)
Anti-Cleaved Caspase 3, rabbit monoclonal, Thermo Fisher Scientific, MA5-32015, Cl. SR01-02, 1:100 (IF)
Anti-Collagen I, mouse monoclonal Thermo Fisher Scientific MA1-26771, Cl.COL-1 1:100 (IF)
Anti-CX43, rabbit polyclonal Sigma-Aldrich C6219 1:100 (IF)
Anti-CXCR4, mouse monoclonal R&D systems MAB172-SP, Cl.44716 12 µg/ml (migration assay)
Anti-DDR2, rabbit polyclonal, Thermo Fisher Scientific, PA5-27752, 1:100 (IF)
Anti-GFP, chicken polyclonal Abcam ab13970  1:500 (IF)
Anti-Human Nuclei, mouse monoclonal  Sigma-Aldrich MAB1281, Cl. 235-1 1:100 (IF)
Anti-Human Nucleoli, mouse monoclonal Abcam ab190710, Cl.NM95 1:100 (IHC)
Anti-Integrin beta 1, mouse monoclonal Abcam ab24693, Cl. P5D2 10 µg/ml (migration assay)
Anti-ISL1, mouse monoclonal DSHB Cl. 39.4D5 1:100 (IF)
Anti-Ki67, mouse monoclonal Agilent Dako M7240, Cl. MIB1 1:100 (IHC)
Anti-MLC2a AF647, mouse monoclonal Synaptic Systems 311011 AT1, Cl. 56F5 1:100 IF
Anti-MLC2v, mouse monoclonal Synaptic Systems 310111, Cl. 330G5 1:100 IF
Anti-MLC2v, rabbit polyclonal Proteintech 10906-1-AP 1:300 (IHC)
Anti-N-cadherin, recombinant Abcam Ab76011 1:100 (IHC)
Anti-OCT4, rabbit polyclonal Cell Signaling 2750 1:50 (IHC)
Anti-Periostin, rabbit polyclonal Abcam 14041 1:100 (flow cytometry)
Anti-ROBO1, goat polyclonal LSBio LS-B3011  1:100 (IF)
Anti-ROBO1, goat polyclonal R&D systems AF1749 1:50 (flow cytometry)
Anti-ROBO1, rabbit polyclonal Thermo Fisher Scientific PA5-99084  5 µg/ml (signaling blockage)
Anti-SDC-4, rabbit polyclonal Abcam ab74139  1:500 (migration assay)
Anti-SDF-1 (CXCL12), rabbit polyclonal Cell Signaling Technology 3740 1:100 IF
Anti-SLIT2 AF647, rat polyclonal R&D systems FAB5444R  1:100 (IF)
Anti-TRA-1-60, mouse monoclonal, Thermo Fisher scientific, 14-8863-82, Cl. TRA-1-60 1:7 (flow cytometry)

Secondary antibodies:
Alexa Fluor 488, goat anti mouse Abcam ab150113 1:250 (IF)
Alexa Fluor 488, goat anti rabbit Abcam ab150077 1:250 (IF)
Alexa Fluora 647, goat anti mouse Abcam ab150115 1:250 (IF)
Alexa Fluora 647, goat anti rabbit Abcam ab150079 1:250 (IF)
Alexa Fluor 594, goat anti mouse Abcam ab150116 1:250 (IF)
Alexa Fluor 594, goat anti rabbit Abcam ab150080 1:250 (IF)
Alexa Fluor 488, donkey anti chicken Jackson Immuno Research  703-545-155 1:100 (IF)
Alexa Fluor647, donkey anti sheep Abcam  ab150179 1:100 (IF)
Alexa Fluor 647, donkey anti goat Abcam ab150131 1:250 (IF)
Alexa Fluor 594, donkey anti goat Abcam ab150132 1:250 (IF)
Alexa Fluor 647, donkey anti rat Abcam ab150156 1:100 (IF)
Hoechst 33258 Staining Dye Solution Abcam ab228550 1:100 (IF)

Validation

Anti-a-actinin, rabbit polyclonal, validated for WB, ICC/IF, IHC-P in Mouse, Rat and Human
Anti-ACKR3, rabbit polyclonal, validated for IHC, IHC-P in Human, Mouse, Rat, Bat, Bovine, Dog, Hamster, Horse, Pig, Rabbit
Anti-cardiac Troponin I, recombinant Abcam, validated for Flow Cyt (Intra), WB, IP, IHC-P in Human
Anti-Cardiac Troponin T, mouse monoclonal Thermo Fisher Scientific, validated for  IF/ICC, IHC (P), and IM applications  in Avian, Canine, Chicken, Fish, Guinea Pig, Human, mouse, Porcine, Rabbit, and Rat
Anti-cardiac Troponin T, rabbit polyclonal Sigma-Aldrich , validated for IF in Human
Anti-Cardiac Troponin T, rabbit polyclonal Abcam, validated for IHC-P, Sandwich ELISA, WB in Human
Anti-CD31 (PECAM-1), sheep polyclonal R&D systems, validated for ICC/IF, KO, Simple Western, WB in Human
Anti-CD31, rabbit polyclonal Novus, validated for WB, ICC/IF, IHC, IHC-Fr, IHC-P in Human, Mouse, Rat, Porcine, Xenopus
Anti-CD68, mouse monoclonal eBioscience, validated for Flow, IHC(F), IHC(P), IP, WB, ICC/IF in Human, Mouse
Anti-Cleaved Caspase 3, rabbit monoclonal, Thermo Fisher Scientific, validated for WB, IHC, IF in Human
Anti-Collagen I, mouse monoclonal Thermo Fisher Scientific MA1-26771, validated for DB, ELISA, IHC, IP, WB, ICC/IF in Bovine, Deer, Human, Mouse, Pig, Rabbit, Rat
Anti-CX43, rabbit polyclonal Sigma-Aldrich, validated for ICC/IF, IHC in Hamster, Bovine, Human, Mouse, Rat
Anti-CXCR4, mouse monoclonal R&D systems MAB172-SP, validated for Flow, IHC, CyTOF, Neutralization in Human
Anti-DDR2, rabbit polyclonal, Thermo Fisher Scientific, PA5-27752, validated for WB, IHC(P), ICC/IF in Human, Mouse
Anti-GFP, chicken polyclonal Abcam ab13970, validated for WB, ICC/IF, species independent
Anti-Human Nuclei, mouse monoclonal  Sigma-Aldrich MAB1281, validated for ICC, IHC, IP in Mouse
Anti-Human Nucleoli, mouse monoclonal Abcam ab190710, validated for WB, IHC-P, Flow Cyt (Intra), ICC in Human
Anti-Integrin beta 1, mouse monoclonal Abcam ab24693, validated for ICC/IF, Flow Cyt in Human.  This antibody inhibits the function of beta 1 integrins and can be used to block cell adhesion (Dittell et al., 1993 and Yokosaki et al., 1994)
Anti-ISL1, mouse monoclonal DSHB validated for IHC, ICC, IF in  larva, Chicken, Ferret, Fish, Frog, Human, Mouse, Rat, Zebrafish
Anti-Ki67, mouse monoclonal Agilent Dako M7240, validated for WB, IHC in Human
Anti-MLC2a AF647, mouse monoclonal Synaptic Systems 311011 AT1, validated for ICC, IHC in Human, Rat, Mouse
Anti-MLC2v, mouse monoclonal Synaptic Systems 310111, validated for WB, IP, ICC, IHC in Human, rat, mouse
Anti-MLC2v, rabbit polyclonal Proteintech 10906-1-AP, validated for WB, IP, IHC, IF in Mouse, Rat, Human
Anti-N-cadherin, recombinant Abcam Ab76011, validated for IHC, WB in Mouse, Rat, Human
Anti-OCT4, rabbit polyclonal Cell Signaling 2750 1:50, validated for WB, IHC, IF, Flow, IP in Human
Anti-Periostin, rabbit polyclonal Abcam 14041, validated for ELISA, ICC/IF, IHC-Fr, IHC-P in Mouse, Rat, Chicken, Human
Anti-ROBO1, goat polyclonal LSBio LS-B3011  1:100, validated for IHC, IHC-P, IF, WB, Peptide-ELISA in Human, Monkey, Mouse, Rat, Bat, Bovine, Dog, Hamster, Horse, Pig, Chicken, Xenopus

Anti-ROBO1, goat polyclonal R&D systems AF1749, validated for WB, IHC in Rat
Anti-ROBO1, rabbit polyclonal Thermo Fisher Scientific PA5-99084  validated for WB, ICC/IF, ELISA in Human
Anti-SDC-4, rabbit polyclonal Abcam ab74139, validated for ELISA, WB, IHC, ICC/IF in Human
Anti-SDF-1 (CXCL12), rabbit polyclonal Cell Signaling Technology 3740, validated for WB in Human, Mouse, Rat
Anti-SLIT2 AF647, rat polyclonal R&D systems FAB5444R validated for FACS in Human, Mouse
Anti-TRA-1-60, mouse monoclonal, Thermo Fisher scientific, 14-8863-82, validated for WB, IHCm ICC/IFm Flow, IP, Misc in Human,
Non-human primate

Secondary antibodies:
All secondary antibodies are validated for ICC/IF according to the manufacturers;
https://www.abcam.com/products?keywords=secondary%20antibodies

# Eukaryotic cell lines

Policy information about cell lines

| | |
|---|---|
| Cell line source(s) | WA09 (H9; human ES cell line) were purchased from WiCell Research Institute, USA.<br>ES03 NKX2.5GFP and H9 NKX2.5GFP cell lines were generously given to us from Dr. David Elliott (MCRI, Australia).<br>HEK293T were purchased from ATCC, USA. |
| Authentication | All human ES cell lines  were validated in the original studies (Elliott et al., Nat Met 2011, Foo et al., Mol Ther 2018) by various means such as RNAseq, staining for pluripotency markers and determining differentiation capacity, as well as by the commercial vendors.<br>HEK293T was only used for the generation of dsRed expressing lentivirus. No additional authentication was performed. |
| Mycoplasma contamination | All cell lines used have been tested negatively for mycoplasma. |
| Commonly misidentified lines<br>(See ICLAC register) | no commonly misidentified cell line is used. |

# Animals and other organisms

Policy information about studies involving animals; ARRIVE guidelines recommended for reporting animal research

| | |
|---|---|
| Laboratory animals | PIGS: For animal experiments sus scrofa, german landrace pigs, and Yucatan minipigs were used (wild-type or transgenic LEA29Y pigs); Female and male pigs were used. Mean age was 5 months, body weight ranged between 40-63kg for sus scrofa, german landrace pigs. Mean age was over 6 months, body weight ranged between 28-40 kg for Yucatan minipigs.<br>NHPs: NHP slices were obtained after termination of control animals that were part of various studies from primate centers in Göttingen, at the Karolinska Institute or the LMU Munich. |
| Wild animals | study did not involve wild animals |
| Field-collected samples | study did not involve samples collected from the field |
| Ethics oversight | PIGS: All animal experiments with german landrace pigs were performed with permission of the local regulatory authority, Regierung von Oberbayern (ROB), Munich, (approval number: AZ 02-18-134). Applications were reviewed by the ethics committee (Sachgebiet 54) according to §15 TSchG German Animal Welfare Law.<br>All Yucatan minipig experiments were approved by IACUC (Institutional Animal Care and Use Committees) under the study number 1974-061 in Charles River Mattawan (MI, USA).<br><br>NHPs: German primate centre, Göttingen  (file reference: 33.19-42502-04-16/2264), Karolinska Institutet, Sweden (file reference: N 277/14) and Walter Brendel Institute, LMU, Germany (file reference: ROB-55.2--2532.Vet_02-14-184). |

Note that full information on the approval of the study protocol must also be provided in the manuscript.

# Flow Cytometry

## Plots

Confirm that:

☒ The axis labels state the marker and fluorochrome used (e.g. CD4-FITC).

☒ The axis scales are clearly visible. Include numbers along axes only for bottom left plot of group (a 'group' is an analysis of identical markers).

☒ All plots are contour plots with outliers or pseudocolor plots.

☒ A numerical value for number of cells or percentage (with statistics) is provided.

## Methodology

| | |
|---|---|
| Sample preparation | For baseline expression of ISl-1, cTNT and Tra-1-60, human ES cells were differentiated into ventricular progenitor cells, in vitro. On D6 and D25, cells were dissociated with accutase before staining for desired epitopes according to manufacturer's instructions. |

| Instrument | FACS Canto III, Gallios flow cytometer (Beckman Coulter, USA) |
|---|---|
| Software | FloJo software version 8 (BD Biosciences), Kaluza software version 1.2 (Beckman Coulter, USA) |
| Cell population abundance | Purity of sample was determined with FACS Canto III or Galios flow cytometer, and it is dependent on markers to markers. |
| Gating strategy | Gating was done in light of negative control (secondaries only or unstained control cells). |

☒ Tick this box to confirm that a figure exemplifying the gating strategy is provided in the Supplementary Information.

