## [Peer Review File · Nature Cell Biology]

Peer Review Information

Journal: Nature Cell Biology

Manuscript Title: Migratory and anti-fibrotic programs define the regenerative potential of human cardiac progenitors

Corresponding author name(s): Karl-Ludwig Laugwitz

Editorial Notes:

Redactions – transferred manuscripts (mention of the other journal) This manuscript has been previously reviewed at another journal. This document only contains reviewer comments, rebuttal and decision letters for versions considered at Nature Cell biology. Mentions of the other journal have been redacted.

Redactions – unpublished data Parts of this Peer Review File have been redacted as indicated to maintain the confidentiality of unpublished data.

Reviewer Comments & Decisions:

Decision Letter, initial version:
--

Dear Professor Laugwitz,

Thank you for allowing Nature Cell Biology the opportunity to consider your manuscript, "Developmental programs determine migratory and anti-fibrotic potential of human cardiac progenitors in heart regeneration", for publication.

[REDACTED] and Nature Cell Biology are editorially independent. However, when authors transfer a manuscript to Nature Cell Biology after it has been reviewed for [REDACTED], the upstream journal passes the referees' comments (and their identities) to us, with the author's permission.

During the consultation with [REDACTED], I have discussed the manuscript, previous referee comments, rebuttal and proposed experiments in detail with my colleagues, and we believe that your findings could be of significant interest to this journal's readership. Thank you for providing us with an updated cover letter and the previous rebuttal that outline how you intend to address the remaining referee concerns with specific focus on NCB's readership.

We would be happy to consider a manuscript revised along the lines you propose in your rebuttal. In particular (to reiterate our conclusion after the consultation), it would be important to strengthen the experimental evidence for the HVP's ability to remuscularise chronically injured myocardium, validate findings from the ex-vivo NHP model in porcine hearts in vivo, improve the characterisation of the cellular migration process, its dependence on CXCR4/CXCL12 signalling, and the crosstalk between cardiac progenitors and the mesenchyme. We are willing to consider a revised manuscript without the data on arrhythmogenicity.

As always, any decision to re-review (with the current or new referees) such a revised manuscript would depend on the strength of the revisions and the published literature at the time of resubmission as assessed editorially.

Please use the link below to submit a suitably revised manuscript and an updated point-by-point rebuttal to the previous referee reports.

In addition, please pay close attention to our guidelines on statistical and methodological reporting (listed below) as failure to do so may delay the reconsideration of the revised manuscript. In particular please provide:

- a Supplementary Figure including unprocessed images of all gels/blots in the form of a multi-page pdf file. Please ensure that blots/gels are labeled and the sections presented in the figures are clearly indicated.
- a Supplementary Table including all numerical source data in Excel format, with data for different figures provided as different sheets within a single Excel file. The file should include source data giving rise to graphical representations and statistical descriptions in the paper and for all instances where the figures present representative experiments of multiple independent repeats, the source data of all repeats should be provided.

On resubmission please also provide the completed Reporting Summary (found here <https://www.nature.com/documents/nr-reporting-summary.pdf>). This is essential for reconsideration of the manuscript and these documents will be available to editors and referees in the event of peer review. For more information see <http://www.nature.com/authors/policies/availability.html>.

When submitting the revised version of your manuscript, please pay close attention to our [href="https://www.nature.com/nature-research/editorial-policies/image-integrity">Digital Image Integrity Guidelines](https://www.nature.com/nature-research/editorial-policies/image-integrity). and to the following points below:

Resubmission link:
[REDACTED]

I should also let you know that, unfortunately, I will be unable to handle your revised manuscript, as I have now resigned from the journal and will leave the company this week. However, my colleagues are fully up-to-date about the status of your manuscript and await your resubmission. If you have any questions in the meantime, please do not hesitate to contact my colleague Dr Zhe Wang (zhe.wang@nature.com), who will be your point of contact after I have left.

If you are not interested in submitting a revised manuscript, please also let us know so we can close your file.

With best wishes,

Christine.

Christine Weber, PhD
Senior Editor
Nature Cell Biology
E-mail: christine.weber@nature.com
Phone: +44 (0)207 843 4924

Author Rebuttal to Initial comments

POINT-BY-POINT RESPONSE TO REVIEWERS' COMMENTS

We thank the reviewers for their interest in our work, constructive criticisms and instructive comments. We have addressed each of the major concerns raised by performing a set of new experiments, as well as *via* editorial and statistic revision. We believe these clarify key issues highlighted in their review. These changes have been incorporated into the revised version of the manuscript and Supplementary Appendix. The new manuscript text and Figures, as well as a point-by-point rebuttal are provided for the reviewers and the editors.

Summary of the major changes: at the suggestion of Reviewer #1 and #2 we have extended our *in vivo* analysis of the regenerative potential of HVPs in a clinically relevant porcine model of chronic heart failure with cardiac fibrosis and can demonstrate that HVPs lead to *de novo* formation of ventricular heart muscle, resulting in significant scar volume reduction and prevention of heart failure progression, as assessed by cardiac MRI at 3 months after cell transplantation. The new set of experiments also confirmed absence of teratoma formation in a 3-month follow-up period as well as consistent effects of HVPs derived from different human pluripotent stem cell lines. These analyses have resulted in a new Figure 7 and a new Extended Data Figure 8. Moreover, as recommended by Reviewer #2, a new set of experiments

addressed the behaviour of HVPs after RFA injury on de-cellularized NHP LV slices and confirmed the importance of host NHP cells as mediator for HVP migration (new panel h in Extended Data Figure 3). In addition, a further transcriptional analysis evaluated the progressive myofibril and electrophysiological / Ca²⁺-handling maturation of HVP-derived cardiomyocytes in the 3D co-culture, resulting in a new panel d in the Extended Data Figure 2. Furthermore, a new panel d of the Extended Data Figure 3 provides evidence on the purity level of the HVP and cardiomyocyte preparations used in the study. At the suggestion of Reviewer #3, a revised Figure 2d provides new GO analysis of signaling pathways enriched in HVPs at the early time of co-culture with NHP LV slices. Moreover, information on the percentage of proliferating HVPs and derived cardiomyocytes in the ex vivo NHP RFA injury model have been compiled in the new panels c and d of the revised Figure 3. Additionally, a new set of experiments using the specific CXCR4 antagonist AMD3100 in both ex vivo and in vivo models of RFA injury confirmed the relevance of the CXCR4/CXCL12 axis in the HVP migration process. This resulted in the new panel c of Figure 4, new panel e of the revised Figure 6, and the new panel d of the Extended Data Figure 4. Lastly, new analysis of cardiac fibroblasts during the repulsion process has addressed the molecular identity of ROBO1⁺ cells and resulted in the new panel f of the revised Figure 5. Finally, as requested by Reviewers #1 and #3, the new Figure 7 (panels b-e) and new Extended Data Figure 8 (panels b and c) provide classical immunohistochemistry as well as low and high magnification images for detection and cellular characterization of the human grafts.

Responses to Reviewer 1

Schneider and colleagues present data on the migratory and anti-fibrotic capacity of hPSC-derived ventricular progenitor cells (HPV). HPVs were isolated by TRA1-60 depletion after induction of mesoderm differentiation using a GiWi protocol. Migratory capacity of HPV was demonstrated after seeding on non-human primate heart slices with radiofrequency ablation (RFA) injury. Comprehensive molecular profiling identified non-active, sensing, migrating/counteracting states as well as signaling between host and graft cells. In vivo administration in an acute model of I/R injury attenuated scarring. The study is interesting, but needs further data to support the claim that muscle can be rebuilt by HPV implantation.

We would like to thank the reviewer for the insightful comments and appreciation of our work. In accordance with the suggestion of the reviewer, we have decided to focus the study to the novel mechanistic profiling of HVPs' ability to sense, migrate and counter-act cardiac injury and provide further experimental evidence on the HVP capability to re-muscularize damaged myocardium *in vivo* as final counteraction to injury.

The rationale for the use of NHP instead of human heart slices, previously reported by Dendorfer and colleagues, is not clear. It seems that the NHP slices, at least in some experiments, can be less well preserved compared to the human slices. I am not sure that this makes them a better model for chronic heart injury. In addition, RFA injury seems quite variable

with only a mild effect in the presented (Fig 2d) RFA+HVP group vs the RFA+CM and RFA only groups; in both, there is no deterioration in function with time.

Our study was inspired by the idea that, for *ultimate* cardiac regeneration and permanent benefit of cell-based therapeutics, it will be ideal to target not only de-novo muscularization, but also vascularization of the graft as well as pathologically activated fibrosis. Challenges we are facing in achieving these goals rely on the difficulties to dissect the cellular and functional dynamics of host and graft crosstalk *in vivo*. Thus, the aim of our study was **1)** to establish a standardized ex vivo model of chronic and acute cardiac injury, which could allow us to investigate molecular pathways and functional dynamic states of human cardiac progenitor cells during heart regeneration and repair at a single cell resolution; and **2)** validate the findings from the ex vivo model in the porcine heart in vivo with high fidelity for human translation.

We agree with the Reviewer that human heart slices can be preserved longer than NHP heart slices in *ex vivo* biomimetic cultures. However, the vast majority of human heart samples are obtained from patients with end-stage heart failure subjected to heart transplantation. Living heart tissue from healthy individuals is extremely rare. Patients with end-stage heart failure represent a very heterogeneous population, since they suffer from different types of cardiomyopathies and are at distinct stages of their heart muscle disease. Furthermore, these patients show different transcriptional profiles of diseased tissues with fibrosis and their medication accounts for additional confounding factors. Thus, human heart slices are less suitable for standardized studies. For this reason, we considered healthy NHP hearts as an appropriate surrogate. Our NHP slices were obtained after termination of control animals that were part of various studies from primate centers in Göttingen, at the Karolinska Institute or the LMU Munich.

We identified conditions that produce standardized RFA injury lesions in the *ex vivo* cultured NHP samples. With 20 Watt for 15 sec., we were able to generate non-transmural lesions with comparable sizes, leading to a decrease of contractile performance of ~50% in all conditions by day 14 (old Fig. 2d, now revised Fig. 3f). Only with HVPs however a gradual increase of contractile force was observed between day 14 and day 21; in this window of time, seeding of CMs failed to restore contraction force of the tissue and, similarly to the RFA-alone condition, functional deterioration further progressed (old Fig. 2d, now revised Fig. 3f). These results corroborate the fact that HVPs, once they reach the RFA area, differentiate into functional CMs and electromechanically couple with the NHP myocardium (old Fig. 2e, now revised Fig. 3g), leading to “scar-less” healing of the injury.

Of note, in the RFA model, in order to investigate the migratory capability of HVPs by live-cell imaging, we only used 5×10^5 cells. In the model of chronic heart injury, 2×10^6 HVPs were used to investigate the regenerative potential of the cells *ex vivo*.

The observation of damage triggered migration of HPV to the site of injury is interesting. The role of CXCR4 suggests similar mechanisms as for bone marrow derived cells. HPV should be compared to cells with migratory activity and not to non-migratory cardiomyocytes.

Cardiomyocytes would most likely seed in the defect area and reconstitute heart muscle if properly placed. No arrhythmia after HPV implantation is an advantage, but only if palpable muscle is formed.

We agree with the reviewer that it is indeed intriguing that HVPs follow the same migratory cues *via* CXCR4 as bone marrow-derived mesenchymal stem cells (MSCs) have been associated with. During the past two decades, bone marrow-derived MSCs, have been used in numerous trials for cell therapy of myocardial infarction. Initially, MSCs' cardioprotective effects were linked to their direct differentiation into CMs on site of the injury; however, more recently, it has been understood that this CM differentiation is diminutive and the main effect of MSCs is paracrine through their secretome and its effects on endogenous cardiac cells (White et al., Clin Ther 2020; A futile cycle in cell therapy, Nat Biotechnol. 2017). In contrast, HVPs, after migration to sites of injury, integrate and directly re-muscularize damaged myocardial tissue and are, thus, themselves direct actors for functional cardiac repair. These actions are more comparable to therapies based on differentiated CMs and these similarities convinced us to choose differentiated CMs as controls instead of MSCs. Moreover, our intention was to compare whether ESC-derived CMs and HVPs have similar migratory features to defined areas of cardiac injury, an aspect that has not been previously explored, and lead to similar muscle formation. We agree with the reviewer that it might also be of interest to compare the functional recovery of injured myocardium after application of a cell population that harbour migratory capacity but lack CM differentiation potential, but we feel that this goes beyond the focus of the current study.

The authors should consider that the animal model, acute I/R injury with concurrent HPV injection, does not provide data to support a remuscularization claim. GFP⁺ cells in SFig 8a show very little if any specific cTNT signal. In contrast, Fig 4g suggests massive remuscularization in the RFA injury model with no muscle present in the border zone, which is surprising giving the precisely demarcated damage inflicted by RFA. In addition, functional recovery (SFig. 7a) did not correlate with the apparent anti-fibrotic effect (Fig 5c).

We thank the reviewer for the comment on the results of the acute I/R injury model. These experiments were primarily driven by arrhythmic event endpoints. For this reason, our main focus in these experiments was to investigate arrhythmic episodes in both CM- and HVP-treated animals. As the evidence for the occurrence of sustained VTs predominantly occurred in the CM-treated animals, immunofluorescence was focused on the presence of electrically integrated cells in the area corresponding to the VT origin identified by our electrophysiological studies (old SFig. 8a). Of note, the tissue slices shown in the old SFig. 8a are from hearts injected with CMs and the focus was to ensure visibility of CX-43, not cTNT signal. Cardiac TNT in GFP⁺ cells derived from HVP transplants were illustrated in the old SFig .7b, indicating clear expression. Furthermore, the *in vivo* analysis of HVPs in the RFA injury model validates the capability of cardiac differentiation and maturation of HVPs *in vivo* (old Fig. 4g, now revised Fig. 6h).

We agree with the reviewer that the anti-fibrotic effect in the acute I/R injury model (old Fig. 5c) appears to be greater than the functional recovery (old SFig. 7a). We would like to point out that the balloon occlusion of the left descending coronary artery created small infarct sizes (around 11% of the LV) and resulted in small effects on global ejection fraction. For an ultimate correlation between functional recovery and scar size reduction, it would be necessary to modify the experimental conditions and perform larger infarctions followed by *in vivo* hemodynamic assessment of LV performance.

Being aware that our acute I/R injury model does not definitively prove remuscularization on a functional level but rather addresses more specifically arrhythmogenesis (a claim that has been dropped in the revised study), we have removed this model from the revised manuscript. Instead, we now provide data on the therapeutic impact of HVPs in a clinically relevant porcine model of chronic heart failure with cardiac fibrosis (new Fig. 7 of the revised manuscript). Our new data indicate that once transplanted into necrotic tissue with extensive fibrosis 21 days post myocardial infarction (new Fig. 7a), HVPs lead to *de novo* formation of ventricular heart muscle (new Fig. 7b, c), resulting in significant scar volume reduction (new Fig. 7 g) and prevention of heart failure progression (new Fig. 7h, i), as assessed by cardiac MRI at 3 months after cell transplantation. For the ease of accessibility, the new Fig. 7 can also be found below.

Figure 7. HVPs re-muscularize fibrotic scars and preserve cardiac function in a porcine model of chronic ischemia *in vivo*. **a**, Schematic of *in vivo* experimental design. Myocardial infarction was performed by balloon-occlusion of the left coronary artery (LAD) (ischemia) and reperfusion after 90 minutes. Triple- immunosuppressive regime (IMS) consisted of cyclosporine (D-6 to D84), methylprednisolone (D-1 to D84) and Abatacept (D-1 to D84). Analysis of baseline infarct volume was conducted by cardiac MRI (cMRI) on day -7 followed by epicardial cell injection (15 injection sites, total 1×10^9 HVPs) into the chronic ischemic area. After 12 weeks follow-up, cMRI scans were the clinical end-point and animals were sacrificed for histological analyses. **b**, Overview of infarct zone and human grafts with labelling of porcine myocardium (HuNu⁺ cTNT⁺), infarct zone and cardiac troponin T positive grafts (HuNu⁺ cTNT⁺). Scale bar 2 mm. **c-e**, Representative immunohistochemical images of human grafts after staining for the ventricular muscle marker MLC2v (**c**), the intercalated disc protein N-cadherin (**d**), and the endothelial marker CD31 (**e**) at 12 weeks. Scale bar 50 μ m. Lower panels show magnifications of boxed areas. Scale bar 15 μ m. **f**, Representative left ventricular (LV) cMRI images of diastole and systole used for calculation of infarct volume, left ventricular ejection fraction and global longitudinal strain. **g-i**, Statistical analysis of infarct volume (**g**), left ventricular ejection fraction (LVEF, **h**) and global longitudinal strain (GLS, **i**). Data are shown as min-to-max range with individual data points, n=10 pigs in the vehicle group and n=7 pigs in the HVP-treated group, *p<0.05, **p<0.005, ***p<0.001 (t-test).

Cx43 stains are nice to have, but not conclusive to support an electrical integration claim. Given the lack of cardiomyocytes in the border zone (Fig 4g) and the rather patchy appearance of Cx43 it is quite unlikely that the provided data is in support of electrical integration claim.

We agree with the reviewer, that electrical integration cannot be claimed by the sole presence of Cx-43 stains. To assess the potential of HVP-derived CMs to functionally integrate into the electromechanical syncytium after injury *ex vivo*, we performed real-time intracellular Ca²⁺ analysis comparing regions of interest (ROI) within the damaged and native myocardium (old Fig. 2e, now revised Fig. 3g). Fluo-4 fluorescence clearly propagated through the injury area when HVPs had been applied, with differentiated HVPs displaying [Ca²⁺] oscillations similar to and synchronized with those in adjacent native NHP CMs. This is a clear indication of electromechanical integration of the HVP-derived CMs in the NHP myocardial slices. *In vivo* electrical activation mappings demonstrated that areas of VT origin corresponded to the cell implantation foci (old SFig. 8). Therefore, we believe that the functional and structural correlation supports our claim of electrical integration of the cells. However, to avoid misinterpretation of our statement, the sentence describing Cx43 expression

has been revised and the part “*suggesting electromechanical coupling of the graft*” was removed (page 11, line 22 of the Results).

Please clarify that juvenile pigs were used and discuss differences, including potential endogenous regeneration, in young vs old pigs.

In this study, juvenile pigs of approximately 4.5 to 5 months of age were used. For the employed pig breed, this represents the age of sexual maturity. Unlike many other tissues, the adult heart in mammals including the pig consists of mostly post mitotic cardiomyocytes and thus possesses no meaningful regenerative capacity (Bergmann O et al **Cell**. 2015; 161:1566–1575. doi: 10.1016/j.cell.2015.05.026.). A transient regenerative window in the neonatal mouse heart is scientifically well established (Porrello et al., Science 2011) and recent findings suggest a similar situation in the neonatal pig heart (Ye et al., Circulation 2018; Zhu et al., Circulation 2018). While the exact time frame of this transient regenerative window in pigs has not been conclusively investigated yet, the longest putative duration appears to be located in the range of 14 days post birth. Since the pigs employed in this study are months beyond this age, we are not aware of any presently available study results that would suggest the possibility of endogenous regeneration disconnected from the HVPs taking place in these animals.

To improve clarity, HE and classical immunohistochemistry should be applied to for example identify grafts and CD68⁺ cells.

We thank the reviewer for this suggestion. Accordingly, the new Fig. 7 (panels b-e) and new Extended Data Fig. 8 (panels b and c) now provide classical immunohistochemistry as well as low and high magnification images for detection and cellular characterization of the human grafts.

The rationale for the use of the LEA29Y pigs with massive immune suppression needs to be explained.

For the *in vivo* transplantation study, we have used two pig cohorts: **1) LEA29Y pigs for the RFA model** and **2) wild-type pigs for the chronic I/R model**. In previous *in vivo* transplantation studies, we have used the commercially available pharmacologic protein LEA29Y (Belatacept) that yielded very little cellular rejection. For this reason, we sought to target the same axis by using transgenic LEA29Y pigs that were generated in Munich (Wolf-van Buerck et al., Scientific reports, 2017). All wild-type animals received a combination of cyclosporin A, methylprednisolone and Abatacept according to Romagnuolo et al. (Stem Cell Reports, 2019). In contrast, the LEA29Y pigs received only methylprednisolone. We have clarified this issue in the revised method section of the manuscript on page 24, lines 1-13, and pages 22 and 23, lines 32 and 1-2, respectively.

A 14-day follow-up period is insufficient to study functional consequences or unwanted side effects such as tumor formation. The administration of PSC-derivatives at a progenitor stage will very likely be associated with a higher risk for tumor growth. A follow-up of 3 months would be appropriate.

We agree with the reviewer that a longer follow-up period is needed in cell transplantation studies to ultimately prove functional recovery and unwanted side effects as tumor formation. We have now provided additional 3-month follow-up data in the porcine chronic heart failure model (7 out of 17 pigs received HVPs), which indicate prevention of heart failure progression by HVP application (new Fig. 7g-i) and substantiate our claim of no risk for tumor formation with these cells. No macroscopic or microscopic signs of teratoma were observed in any animals after 3 months HVP application nor presence of undifferentiated pluripotent stem cells was detected in the transplanted grafts (new Extended Data Fig. 8a, b). This has been stated in the revised Results on page 12, lines 11-13, and revised Methods on page 24, lines 25-30.

These findings are completely in agreement with our previous study (Foo et al., Molecular Therapy 2018) that showed no signs of any tumor formation in a cohort of 25 NSG mice after 2 months (n=21) and 8 months (n=4) intra-myocardial transplantation of 2×10^6 (MAC sorted; TRA-1-60+ < 3%) HVPs. We believe that when we negatively sort out potential TRA-1-60+ pluripotent stem cells to less than 3%, the likelihood of tumor formation is extremely low.

In addition to the eGFP detection, parallel NKX2.5 antibody staining of the HPV in vitro and in vivo must be provided to demonstrate specificity.

The ES03 NKX2.5eGFP and H9 NKX2.5eGFP cell lines used are well-studied lines that have been extensively validated in our previous work (Foo et al., Mol Ther 2018) and by other groups (Elliott et al., Nat Methods 2011; Anderson et al., Nat Commun 2018). Therefore, we did not perform new characterizations but provide below co-immunostainings of HVPs with antibodies for human nuclei (HuNu) and Nkx2.5 that were performed to validate GFP specificity *in vivo* below.

Statistical testing in less than $n=3$ data sets must be avoided (e.g., Figs 4e and f)

We have now increased the sample number to 3 in the statistical analysis of the relative reduction of scar volume with HVPs compared to CMs on D21 in our *ex vivo* NHP RFA-injury model (old Fig. 2c, now revised Fig. 3e). For statistical testing of scar volume in the *in vivo* porcine RFA-injury model the n is 3 at D5 after HVP transplantation and 2 at D14. We are aware that the statistical power at D14 is low, but the experiments in this model were primarily driven by migration endpoints at D5. We have now better emphasized this aspect in the revised text on page 10, lines 30-31. The capability of HVPs to reduce scar volume was further investigated in a clinically relevant porcine model of chronic heart failure (new Fig. 7 of the revised manuscript), as mentioned above. In this new set of experiments, 17 pigs underwent myocardial infarction and were treated with HVPs ($n=7$) or vehicle ($n=10$) 3 weeks later. HVPs led to *de novo* formation of ventricular heart muscle (new Fig. 7b, c), resulting in significant scar volume reduction (new Fig. 7g) and prevention of heart failure progression (new Fig. 7h, i), as assessed by cardiac MRI. These data are now described on a new paragraph “HVPs re-muscularize fibrotic scars and preserve cardiac function in a porcine model of chronic ischemia *in vivo*” (pages 11-13 of the revised Results).

Responses to Reviewer 2

This is an interesting and extensive study, from a well-established group in the field, which may be important for the emerging cardiovascular regenerative medicine discipline. Nevertheless, there are some important limitations to the models used as well as major issues that should be addressed as discussed below.

We thank the reviewer for the appreciation of our work and the efforts and critical input during the review of our manuscript. The specific responses to each of the points are noted below.

Major Comments:

(1) The authors claim that NHP slices offer an “ideal setting for investigating cell-based mechanisms of cardiac repair”. I am not sure that in-vitro cell loss may be a relevant or ideal model for tissue damage as occurs in relevant clinical settings such as different stages of myocardial infarction or acquired and inherited cardiomyopathies. I would be glad if the authors can show molecular and functional data that validate the relevance of this model showing that it can recapitulate pathophysiological events occurring in-vivo.

Our study was inspired by the idea that, for *ultimate* cardiac regeneration and permanent benefit of cell-based therapeutics, it will be ideal to target not only de-novo muscularization, but also vascularization of the graft as well as pathologically activated fibrosis. Challenges we are facing in achieving these goals rely on the difficulties to dissect the cellular and functional dynamics of host and graft crosstalk *in vivo*. Thus, the aim of our study was 1) to establish a standardized ex vivo model of chronic and acute cardiac injury, which could allow us to investigate molecular pathways and functional dynamic states of human cardiac progenitor cells during heart regeneration and repair at a single cell resolution; and 2) validate the findings from the ex vivo model in the porcine heart in vivo with high fidelity for human translation.

The biomimetic culture chambers enable the *ex vivo* culture of native human and non-human primate (NHP) heart slices. Previously, the group of Andreas Dendorfer (*Fischer et al. Nat. Communications, 2018*) has established this system and extensively characterized native human heart slices that were cultured herein for up to 4 months. The *ex vivo* culture system is missing physiological stimuli. But genes involved in E/C coupling, adrenoreceptor signaling, and markers of cardiac contractility, mitochondria and growth-factor responses showed stable expression over 35 days *ex vivo*. Furthermore, nutritive demands of the myocardium appeared to be fulfilled. However, up-regulation of matrix–integrin interaction and attenuation of inflammation corresponded to a described remodeling process during the *ex vivo* tissue culture.

Strikingly, the multi-cellular nature of the cultured *ex vivo* tissue seemed to support cellular interactions by the continuous release of functionally important endogenous mediators. In our analysis on NHP heart tissue, we have seen accumulation of cell death from day 14-21 onward (Fig. 1c and Extended Data Fig. 1b). With the evidence of continuous release of functionally important endogenous mediators, shown by *Fischer et al.*, our secretome analysis of RFA-injured slices proved that an active response to inflammatory and fibrotic cues recapitulate pathophysiologic events (Schruf et al., *Respir Res* 2019; Saadat et al. *Front in Cardiovasc Med*, 2021) in the *ex vivo* setting (Extended Data Fig. 5b). The secreted proteins involved in the inflammatory and fibrotic cues such as ATP5A1, FSTL1, COL6A1, LDHB and COL1A2 were also detected as transcripts in our scRNA-seq data and were expressed in both NHP-cardiac fibroblasts and NHP-macrophages; GOT2, MDH2, SPARC, DLD, LDHA, FN1 were found only in the NHP-macrophages (Supplementary Table 1).

(2) It is possible that the 3D ECM architecture of the LV slice on itself would be sufficient to guide similar engraftment processes of the HVPs without the presence of host cardiomyocyte tissue. The authors could therefore evaluate whether similar results can be obtained if the experiments were performed in decellurized (acellular) LV slices.

We thank the reviewer for this valuable comment. In fact, it has been shown that the cardiac extracellular matrix (cECM) can provide recruitment signals for bone-marrow derived stem cells, myofibroblasts and macrophages after myocardial infarction *in vivo* (Nelson & Bissell, *Annu Rev Cell Dev Biol*, 2006). Furthermore, cECM has demonstrated to provide a unique environment in which stem cells can attach, migrate and mature phenotypically (Hashimoto et al., *Nat. Rev. Cardiol.* 2018; Kc et al., *Regen. Biomater.* 2019).

Indeed, in other studies we have used acellular extracellular matrix (ECM) of native NHP heart slices and seeded on top ESC-derived HVPs or CMs. Both cell types can nicely integrate in the matrix; HVPs differentiate into CMs, which then elongate and further mature overtime under electromechanical conditioning. Particularly, we have observed an overall similar contractile performance of HVPs in co-culture with native NHP LV and HVPs in de-cellularized LV-ECM slices. However, under co-culture conditions, the speed of differentiation is delayed and HVPs appear to first proliferate and migrate before they start to differentiate in response to the dying native myocardium (Fig. 1c, revised Fig. 2d and Extended Data Fig. 1f, g). Subsequently, cellular maturation is significantly increased in the presence of native dying myocardium (as assessed by scRNAseq and myofibril/sarcomere structure, Fig. 1d, revised Fig. 2e, and Extended Data Fig. 2c, d). The figure below illustrates key differences between the co-culture and the de-cellularized LV-ECM systems and is presented here for the reviewer only.

HVPs integrate and functionally mature in de-cellularized cardiac ECM (cECM) slices. **a**, Statistical analysis of contraction force of cECM constructs re-seeded with CMs or HVPs on days 7, 14 and 21. **b**, Contractile performance of HVPs in coculture with native NHP heart slices (NHP+ HVPs) and HVPs in de-cellularized cECM slices (cECM + HVPs) on indicated days. **c**, Statistical analysis of

proliferating (EdU⁺) GFP⁺ HVPs on D3 and D7 in NHP co-culture or cECM slices and **d**, GFP⁺/ISL1⁺ HVPs in NHP co-culture or cECM slices. **e**,

Representative immunostainings for anti- GFP and cardiac Troponin T (cTNT) of HVPs in NHP coculture (top) and cECM slices (bottom) on day 21. Scale bar 25 μm. All data are indicated as mean ± SEM and individual data points. *p<0.05, **p<0.005, ***p<0.001 (Two-way ANOVA for a,b and multiple t-test for c,d).

Given that the focus of this manuscript was to define whether HVPs could effectively provide heart regeneration in the setting of myocardial damage by orchestrating and activating sequential programs of cardiac development, we would prefer not to show these data in the current manuscript and describe the behaviour of HVPs on decellularized LV slices in a separate work.

However, we have now included experimental evidence on HVP behavior after RFA injury on de-cellularized LV slices (revised Extended Data Figure 3h). Most interestingly, a directed migration towards the RFA injury was not recapitulated in the absence of native NHP cells, as stated on page 7, lines 31-32 of the revised manuscript. This further indicates that the presence of native NHP cells as mediator for HVP migration is key in the process of HVP- mediated cardiac regeneration after injury.

(3) Similarly, an RF ablation injury is probably also not the best model to evaluate for myocardial regeneration, signals for chemo-attraction, and anti-fibrotic processes in areas of

tissue damage; since it differs from clinically relevant myocardial injury scenarios such as acute/subacute/ or chronic ischemia.

We agree with the reviewer that acute or chronic ischemia restricted to a defined area of the myocardium would be more clinically relevant than RFA injury. However, such models are impossible to achieve in heart slices *ex vivo*, where vessels cannot be occluded and only global ischemia *via* deprivation of oxygen is feasible. RFA is used routinely in the clinic for ablation of rhythm disorders like atrial fibrillation and allows generating a very precise and standardized myocardial injury.

We could confirm that our findings obtained in the *ex vivo* model of RFA-mediated myocardial injury can be recapitulated in an equivalent porcine model *in vivo* (old Fig. 4, now revised Fig. 6). Moreover, new data confirm that directed migration of HVPs to the injured RFA area *in vivo* is similarly dependent from CXCR4 and can be pharmacologically blocked by application of the CXCR4 specific antagonist AMD3100 (revised Fig. 6e).

Additionally, in the revised manuscript, we now provide a new set of data obtained in a porcine model of chronic ischemia (new Fig. 7), which we whole-heartedly agree provides a more clinically relevant myocardial injury scenario (see also response to point 4 below).

(4) Consequentially, due to the limitation of the aforementioned *in vitro* models in mimicking clinical relevant scenarios, the authors are encouraged to assess similar chemo-attraction and anti-fibrosis signaling as well as cellular engraftment in more physiologically relevant models of myocardial damage. One suggestion may be to use a myocardial infarction model in rodents. The authors can evaluate these processes both in the acute ischemia and chronic scar stages by transplanting the cells either at the injury territory or away from this territory to assess for migration.

We thank the reviewer for this suggestion. Considering the numerous differences in heart physiology and structure between mice and humans (beating rate, compensatory mechanisms during disease development, collateral vessel architecture, etc.) we have opted for *in vivo* models based on large animals such as pigs, which provide critical advantage in translating findings for cardiac regenerative medicine. For example, the pig heart has a cardiac structure, sinus rate, contractile function, and weight-to-body ratio that closely resemble that of an adult human.

We now provide additional data on the therapeutic impact of HVPs in a clinically relevant porcine model of chronic heart failure with cardiac fibrosis (new Fig. 7 of the revised manuscript). Our new data indicate that once transplanted into necrotic tissue with extensive fibrosis 21 days post MI (new Fig. 7a), HVPs lead to *de novo* formation of ventricular heart muscle (new Fig. 7b, c), resulting in significant scar volume reduction (new Fig. 7 g) and prevention of heart failure progression (new Fig. 7h, i), as assessed by cardiac MRI at 3 months after cell transplantation. Moreover, we observed that human cTNT⁺ cells were broadly dispersed around the injection canal, corroborating the intrinsic migratory capacity of

HVPs also in the setting of chronic injury (new Extended data Fig. 8c and new Extended Data Movie 3). For the ease of accessibility, the new Fig. 7 can also be found below.

Figure 7. HVPs re-muscularize fibrotic scars and preserve cardiac function in a porcine model of chronic ischemia

in vivo. **a**, Schematic of *in vivo* experimental design. Myocardial infarction was performed by balloon-occlusion of the left coronary artery (LAD) (ischemia) and reperfusion after 90 minutes. Triple- immunosuppressive regime (IMS) consisted of cyclosporine (D-6 to D84), methylprednisolone (D-1 to D84) and Abatacept (D-1 to D84). Analysis of baseline infarct volume was conducted by cardiac MRI (cMRI) on day -7 followed by epicardial cell injection (15 injection sites, total 1×10^9 HVPs) into the chronic ischemic area. After 12 weeks follow-up, cMRI scans were the clinical end-point and animals were sacrificed for histological analyses. **b**, Overview of infarct zone and human grafts with labelling of porcine myocardium (HuNu⁺ cTNT⁺), infarct zone and cardiac troponin T positive grafts (HuNu⁺ cTNT⁺). Scale bar 2 mm. **c-e**, Representative immunohistochemical images of human grafts after staining for the ventricular muscle marker MLC2v (**c**), the intercalated disc protein N-cadherin (**d**), and the endothelial marker CD31 (**e**) at 12 weeks. Scale bar 50 μ m. Lower panels show magnifications of boxed areas. Scale bar 15 μ m. **f**, Representative left ventricular (LV) cMRI images of diastole and systole used for calculation of infarct volume, left ventricular ejection fraction and global longitudinal strain. **g-i**, Statistical analysis of infarct volume (**g**), left ventricular ejection fraction (LVEF, **h**) and global longitudinal strain (GLS, **i**). Data are shown as min-to-max range with individual data points, n=10 pigs in the vehicle group and n=7 pigs in the HVP-treated group, *p<0.05, **p<0.005, ***p<0.001 (t-test).

These data are now described on a new paragraph “HVPs re-muscularize fibrotic scars and preserve cardiac function in a porcine model of chronic ischemia *in vivo*” (pages 11-13 of the revised Results).

(5) The scRNAseq conducted from day 0-21 should be compared to a similar in-vitro maturation process without exposing the HVPs to the degenerating NHP slide. In addition, if one would want to discuss a hypothesized sensing mechanism, a control containing non-degenerating tissue should be used (e.g. cultured highly matured multicellular cardiac tissue derived from pluripotent stem cells). Otherwise, any discussion regarding a sensing mechanism is highly speculative.

We thank the reviewer for the critical feedback. We have now compared expression levels of key genes captured by scRNAseq during the differentiation process of HVPs to cardiomyocytes in the NHP LV slices with scRNAseq data from 2D *in vitro* differentiation that we previously published (Mononen et al., Stem Cells 2020) and were obtained with the same scRNAseq platform (SMARTseq2). As illustrated in the Figure below, HVPs after 3 days in the NHP co-culture are less differentiated than equivalent cells in 2D (Day 9), supporting the fact that under co-culture conditions the speed of differentiation is delayed and HVPs appear first to proliferate and migrate before they start to differentiate in response to the dying native myocardium (Fig. 1c, revised Fig. 2d and Extended Data Fig. 1f, g).

scRNAseq comparison between 2D in-vitro differentiation and 3D ex vivo co-culture of HVPs. Violin plots are shown for selected key genes during cardiac differentiation. Expression of cardiomyocyte genes *MYL3*, *MYL7*, *TNNT2*, *MEF2C*, and *NKX2-5* is delayed under co-culture condition on NHP LV slices and HVPs maintain higher levels of *ISL1*.

We agree with the reviewer that the scRNAseq data gathered in our “chronic injury model” of spontaneously degenerating NHP heart slices do not ultimately prove a HVP sensing mechanism but only suggest it, as clearly stated in the text on page 6, line 1. However, our RFA model of acute injury unambiguously catches the different stages of specific HVP migration towards the injury site (old Fig. 3a-c, now revised Fig. 4). This process was not recapitulated by matured CMs (old Fig. 2a, now revised Fig. 3a), or in the absence of an acute injury (Extended Data Figure 3c), thus highlighting the unique potential of HVPs to sense and react to tissue defects. By using D0 as an intrinsic control, we have seen dynamic changes in gene regulation that corresponded to HVP behavior in tissue repair, starting from a non-activating state, followed by a specific up-regulation of cell activation and ECM genes 24h after injury that moved on to an upregulation of cell motility, projection and secretion genes. Importantly, in our more simplistic 2D model of RFA injury that includes only HVPs and NHP cardiac fibroblasts (old Fig. 3d, now revised Fig. 5) we have been able to recapitulate HVP directed migration, fibroblast repulsion and ultimate cardiac differentiation after RFA injury without exposing HVPs to the degenerating NHP tissue. The latter results clearly indicate that the HVPs’ specific sensing-reacting response to the tissue environment is largely dictated by the activated cardiac fibroblasts, at least in the *ex vivo* setting.

(6) What is the relative difference in cardiac fibroblast content between the HVP cells and the

CMs evaluated in the experiments in figure 2? What will be the regenerative capacity of CMs and cardiac-fibroblast cells if both were studied? Is there a possibility that higher fibroblast content in the HVPs may explain the higher migratory capacity of those cells?

While we believe that the fibroblast content of HVPs and CMs, both being derived from pluripotent stem cells by the same protocol that specifically directs cells towards the myocyte lineage, is neglectable, we agree with the reviewer that a different content of cardiac fibroblasts between HVP and CM batches could influence the migratory capacity of the cells. As the reviewer is certainly aware, cardiac fibroblasts arising during differentiation of hiPSCs that goes through an ISL1⁺ progenitor state largely maintains ISL1 expression (Zhang et al., Nat Commun 2019). On the other hand, ISL1⁺ HVPs also express typical fibroblast markers as TCF21, PDGFR α , VIM, FN as illustrated in the Figure below, making it difficult to distinguish the two cell types. In order to analyze whether differences in cell purity exists between our HVPs and CMs batches we performed FACS analysis for ISL1, cTNT, and TRA-1-60. As illustrated in the new revised Extended Data Figure 3d, more than 87% of cells were ISL1⁺ in the HVP batches and over 81% were cTNT⁺ in the CM batches, supporting their HVP and CM identity, respectively. Almost 50% of cells expressed already cTNT in the HVP preparations, while only a very small proportion of ISL1⁺ cells (2.1%) was found in the CMs. Positivity for TRA1-1-60 accounted only 1.8% of cells in HVPs and 1.6% in CMs. Given a similar purity over 80%, we believe that HVPs and CMs preparations are comparable and contain a similar proportion of “contaminating” cells that might eventually affect their behaviour, e. g. the migration capability.

HVPs express typical fibroblast markers at D0. The selected fibroblast gene expression distribution is displayed on UMAP plots depicting cells on D0 (HVPs before seeding on NHP LV slices) and eGFP⁺ cells isolated on D3 and D21 of *ex-vivo* co-culture in NHP LV slices. Single cell RNAseq data from our previously published scRNAseq dataset from D-3 of *in-vitro* differentiation are also integrated. Each dot is a single cell, and cells are coloured based on the gene expression level

- (7) The large animal pig studies, including n=2 or n=3, are by no means sufficient to derive significant conclusions regarding scar remuscularization and arrhythmia vulnerability. Larger numbers are required to assess these outcomes. Moreover, the functional analysis is relatively limited to EF measurements by ventriculography. The authors are encouraged to

provide further data regarding volumes and wall motion abnormalities (echocardiography or MRI) and hemodynamic parameters (pressure recordings).

We agree with the reviewer. As mentioned above, we now provide additional data on the therapeutic impact of HVPs in a clinically relevant porcine model of chronic heart failure (new Fig. 7 of the revised manuscript). In this study 17 pigs underwent MI by occluding the left anterior descending (LAD) coronary artery using a percutaneous balloon catheter for 90 min, followed by reperfusion (see Methods and Fig. 7a). In 7 animals, we delivered $\sim 1 \times 10^9$ HVPs by surgically exposing the hearts 21 days after ischemia/reperfusion procedure and injecting the cells into the necrotic tissue. Cardiac MRI was performed 7 days before and 12 weeks after cell transplantation to assess LV function and infarct volume. HVPs led to de novo formation of ventricular heart muscle (new Fig. 7b, c), resulting in significant scar volume reduction (new Fig. 7g) and prevention of heart failure progression (new Fig. 7h, i). These results are now described on a new paragraph “HVPs re-muscularize fibrotic scars and preserve cardiac function in a porcine model of chronic ischemia *in vivo*” (pages 11-13 of the revised Results).

Being aware that our acute I/R injury model does not definitively prove remuscularization on a functional level (due to small infarct sizes - around 11% of the LV – which resulted in small effects on global ejection fraction) but rather was designed to address more specifically arrhythmogenesis (a claim that has been dropped in the revised study), we have now removed this model from the revised manuscript.

(8) Robo-1 knockdown and upregulation should be assessed in order to make sure that the repulsion mechanism is mediated via Robo-1.

We appreciate the reviewer’s suggestion. However, knockdown or overexpression of Robo-1 in our system poses some challenges. By applying Robo-1 antibody at early time points during cardiac fibroblast invasion of the RFA we could see that Robo-1 is also important for this process. Therefore, a constitutive Robo-1 knockdown in the cardiac fibroblast would impair their ability to populate the RFA area and in turn affect HVP chemo- attraction/migration. A conditional knockdown would most likely not allow a temporal precise reduction of Robo-1 in the short time window when the fibroblasts are repelled. On the other hand, Robo-1 overexpression would likely result to an uncontrolled movement of cardiac fibroblasts to the RFA. Application of Robo-1 antibody in the culture medium, instead, consents to achieve a time-controlled and reversible blockage of the protein and offers a clear advantage for such loss-of-function experiments.

We believe that our pharmacological treatments with anti-Robo-1 antibodies combined with gain-of-function experiments using recombinant Slit2 proteins at the stage of cardiac fibroblast repulsion strongly support our hypothesis that Slit2/Robo1 interaction is one of the key signaling pathways in HVP-guided fibroblast repulsion.

(9) The authors should compare the relative frequency of atrial, ventricular, and nodal like

differentiating cells between cell derivatives of the HVP used and the hPSC-CMs. Could this explain the difference between the two cell types in the different model and especially in the arrhythmic behavior? In addition, the authors should perform some type of *in vitro* electrophysiological comparisons between the two cellular types, again to try to define reasons for the differences in arrhythmogenicity? Finally, are their differences in cellular density in the infarct area between the groups, which can explain differences in arrhythmia propensity?

We thank the reviewer for the valid suggestions, that we will certainly consider in the future study that will address arrhythmogenicity in both acute and chronic ischemia/reperfusion pig models. As mentioned above, the current revised manuscript omits the arrhythmogenesis part of the work and rather focuses on the HVPs ability to re-muscularize injured myocardium.

Nevertheless, to address the reviewer's points, we would like to emphasize that HVPs and CMs are both derived from pluripotent stem cells by the same protocol. Analysis of atrial, ventricular, and nodal-like cells has been conducted by bulk RNAseq on days 0-7, D19 and D35 in a previous study (Foo et al., Mol. Therapy 2018) and has indicated minimal contamination of HVPs with nodal and atrial cells, even varying with preparations. HVPs appear to give rise mainly to ventricular cardiomyocytes, as also confirmed by our immunohistological and molecular analyses presented in Fig. 1d, new Fig. 7c, and revised Extended Data Fig. 2d.

Regarding the question on cellular density in the infarct area between the HVP and CM groups, an interesting observation arose from analyzing cell grafts morphology and distribution after *in vivo* injection of HVPs and CMs in our model of acute ischemia. Contrarily to HVPs, which were broadly dispersed around the injection canal, CMs remained highly concentrated at the application site and formed more condensed cell clusters than those seen in the HVP group (old SFig. 7c). This might suggest that ectopic automaticity may depend on local cell density within transplants, which is reduced for HVPs due to their migratory capacity. This warrants further examination.

(10) The authors state that: "taken together, these EP data suggest that arrhythmia arise from foci of abnormal impulse generation acting as ectopic ventricular pacemakers". Was a micro-reentry mechanism really ruled out?

Electro-anatomical mapping studies identified spontaneous, incessant VTs originating from the graft in the CM-transplanted animals. Neither overdrive pacing nor cardioversion could extinguish ventricular arrhythmias. Furthermore, ventricular entrainments were negative, indicating an ectopic impulse generation rather than a re-entry mechanism as their underlying cause.

(11) Regarding the histological analysis (especially in the pig model study) the author provide mostly high magnification images, where it is difficult to appreciate the relative degree of cellular re-population of the infarcted area. The authors should provide low-magnification

immunofluorescence (or even better immunohistochemical) images showing the entire scar, in which one could appreciate the extent of the cell graft and fibrotic scar.

We thank the reviewer for this suggestion. Accordingly, the new Fig. 7 (panels b-e) and new Extended Data Fig. 8 (panels b and c) now provide classical immunohistochemistry as well as low and high magnification images for detection and cellular characterization of the human grafts.

(12) The authors used only one hPSC line, the NKX2.5eGFP/wt hESC line (in which only one NKX2.5 allele is active) to derive the HVPs or CMs. The authors should repeat some of the key experiments also using another hPSC line (preferably not transgenic) to reproduce their findings, in order to determine whether the signaling pathways identified and the migration and engraftment process behavior evaluated represent a more general phenomenon.

We mainly used 3 hPSC lines: the ES03 NKX2.5eGFP and H9 NKX2.5eGFP were used in the *ex vivo* experiments and in the RFA-injury porcine model *in vivo*; the WA09 wild-type line was used in the chronic ischemia porcine model *in vivo*. We specified this in the revised manuscript in the Methods (page 16, lines 3 and 28) and the Results (pages 4 and 12, lines 14-15 and 5, respectively) sections.

(13) In many of the immunofluorescence images provided, the eGFP appearance of the NKX2.5eGFP/wt hESC derived HVPs or CMs seems cytoplasmic. Shouldn't it be nuclear in this transgenic line?

While it is true that NKX2.5 is a transcription factor, and mainly located within the nucleus, the eGFP protein contains no nuclear tag in this line, thus it is cytosolic.

Minor Comments:

1. Throughout the text and abstract, fibrosis has been over-emphasized as a barrier to myocardial regeneration while actually in most cases fibrosis is only a marker appearing follow ischemia, cardiomyocyte abnormalities or due to endothelial dysfunction. I would suggest to temper down its role to a more realistic one.

Nearly all aetiologies of heart disease involve pathological myocardial remodeling characterized by excessive deposition of extracellular matrix proteins by cardiac fibroblasts, leading to cardiac fibrosis, which reduces tissue compliance and accelerates the progression to heart failure (Joshua et al., *Circ Res* 2016). Myocardial ischemia, inherited heart diseases and microvascular obstructions all lead to cell death and fibroblast activation, which result in cardiac fibrosis. To our understanding, myocardial regeneration can only be achieved substantially, by targeting fibrosis.

2. The final sentence of the abstract “As such, they may represent an ideal bio-therapeutic for functional heart rejuvenation” may be appropriate for use if this strategy was proven in a large randomized clinical trial.

We thank the reviewer for this cautionary remark. Given the addition of new data on the therapeutic impact of HVPs in a clinically relevant porcine model of chronic heart failure (new Fig. 7 of the revised manuscript), indicating *de novo* formation of ventricular heart muscle (new Fig. 7b, c), significant scar volume reduction (new Fig. 7 g) and prevention of heart failure progression (new Fig. 7h, i), we feel justified to speculate that “HVPs may represent an ideal bio-therapeutic for functional heart rejuvenation”. Our findings in a translational large animal model of chronic heart failure that closely resemble the situation seen in heart failure patients pave the way for embarking on first-in-men clinical trials.

3. The authors show some evidence for HVP maturation on the NHP slice. It will strengthen the manuscript may the authors show the levels of TNNI3/TNNI1 proteins and the electrophysiological signature of the cells prior and following the maturation process.

Following the reviewer’s suggestion, we have now included new qRT-PCR expression data of HVPs on day 0 (before seeding on NHP LV slices) and eGFP⁺ cells isolated on day 14 and day 21 of *ex-vivo* co-culture with the NHP myocardium, which further highlight the gradual maturation process of HVP-derived cardiomyocytes in the 3D co-culture (new panel d in the revised Extended Data Figure 2). Progressive myofibril maturation is indicated by sarcomeric isoform switching (including *TNNI1/TNNI3*, *MYH6/MYH7*, and *MYL7/MYL2*) and is paralleled by a maturation of electrophysiological and Ca²⁺ handling signatures, as measured by increased expression of cardiac ion channels important for action potential generation (as *CAV1.2*, *KCNJ2*, *SCN5A*) and Ca²⁺ handling genes (as *RYR2*, *PLN*, *ATP1A2*, *ATP2A2*, *HRC*, *SLC8A1*, etc.). We now described these results in the revised manuscript text on page 6, lines 8-10.

4. Line 187-189: "We established 20W for 15sec as efficient RFA conditions to destroy, in a standardized manner, a defined area of the cellular compartment within the NHP heartslices, leaving the extracellular matrix (ECM) as scaffold intact (Supplementary Fig. 3a)." RF injury results in ECV damage due to the increase temperature that also caused protein denaturation, so I'm not sure that the ECM really remains intact.

We thank the reviewer for this comment. We agree that without proving lack of denaturation of ECM by RFA, it is not appropriate to state that the ECM remains intact. Therefore we have modified the sentence “..leaving the extracellular matrix (ECM) as scaffold intact..” to “...leaving the extracellular matrix (ECM) **structure** as scaffold intact” (page 6, line 22).

Responses to Reviewer 3

Remarks to the Author:

This is a very elegant and insightful study that build on previous murine findings showing the cardioprotective effects of HVPs after cardiac injury. Using an ex vivo non-human primate 3D model the authors are able to explain at least parts of the cardioprotective mechanisms. While the ex vivo data are supportive of the conclusions, it would have been good to validate some of these findings in the porcine in vivo models.

We thank the reviewer for the appreciation of our work and constructive criticism and insights that helped us to improve the manuscript. In accordance with the suggestion of the reviewer, we have extended our characterization of the novel mechanistic profiling of HVPs' ability to sense, migrate and counter-act cardiac injury, validated the CXCR4-dependency for their migration in vivo, and provide further experimental evidence on the HVP capability to re-muscularize chronically damaged myocardium as final counter-action to injury. The specific responses to each of the points are noted below.

Some specific comments:

Related to figure 1 the authors show that co-culturing of HVPs in the non-human tissue 3D model basically completely becomes Nkx2.5 expressing cardiac progenitor cells over the course of 50 days. What signal triggers these cells to divide and replace the lost tissue?

We appreciate the comment regarding the HVP's potential to repopulate the dying tissue slice. After seeding of HVPs onto native NHP LV slices, an increase in proliferation rate on day 3 was detected by FACS analysis (Fig. 1c). To further investigate the signals that may trigger HVPs to divide, we have now extended the analysis of our scRNAseq data (gathered at day 0, 3 and 21 after HVP seeding on NHP myocardial slices) and performed GO analysis of signaling pathways enriched between day 0 and day 3. In addition to Wnt signaling, which we previously showed to promote ISL1⁺ cardiac progenitor expansion *ex vivo* (Qyang et al., Cell Stem Cell 2007), we detected a significant upregulation of ERK1/2 cascade and activation of TOR signaling, while BMP pathway was repressed indicating that most HVPs has already acquired a myocytic fate on day 3 (Prall et al., Cell 2007). ERK1/2 and TOR pathways are involved in embryonic cardiac development and important for proliferation and cardiac growth as well as ventricular trabeculation (Rose et al. Physiol Rev 2010; Sciarretta et al., Cardiovasc Research 2021). Interestingly, ERK1/2 signaling plays an important role during valvulogenesis for integration of signals from the extracellular matrix to regulate cardiac cushion proliferation and EMT before differentiation (Rose et al. Physiol Rev 2010). As ECM remodeling and metallopeptidase activity are likewise processes that become activated on day 3, ERK1/2 cascade seems an intriguing developmental pathway by which HVPs may sense and react to the tissue environment prior to differentiation in the setting of the morphological remodeling observed in the NHP slices (Rose et al. Physiol Rev 2010). These new results are now

presented in the revised Fig. 2d and described on page 5, lines 25-32 of the revised manuscript text.

On Pg 5 the authors state that the heart slices regain contractility in the presence of HVPs. However, this needs some rewording as it rather reflects maintenance of function in time.

We appreciate the reviewer's comment and agree that, by looking at Fig. 1b, depicting the average contraction force at day 7, 14, 21, and 50, force appears "maintained" over time in presence of HVPs. However, continuous recording of the contractile performance of NHP LV slices persistently indicated a drop of contraction force from day 14 onwards and only with HVPs application the slices could regain function after this initial drop. This is illustrated in the Extended Data Fig. 1e. For clarification, we have now revised our statement on page 4, lines 25-27, which now reads as *"..heart slices gradually regained contractile force in the third week of co-culture (Extended Data Fig. 1e), reaching 2mN force generation that was further maintained up to D50 (Fig. 1b and Extended Data Fig. 1e)."*

Why do the scRNA seq data only detect CM on D21 and not endothelial cells as expected based on data shown in figure 1E?

The scRNAseq data on D21 were obtained from cells that were positively sorted for GFP expression. While it is true that by immunostaining ~3,8% of GFP⁺ cells showed co-expression of CD31⁺ (most likely due to prolonged GFP expression after silencing of the NKX2.5 promoter in the endothelial lineage) these cells are most likely underrepresented in the FACS-sorted GFP⁺ population at D21. Furthermore, the single cell sequencing technology we applied may have further hampered their detection. scRNAseq was performed on a SMARTseq2 platform, whose advantage is to provide a good coverage of the transcriptome with rarer transcripts being detectable. However, because of the manual nature of the protocol, processing of cell number is limited to the hundreds. Since only 313 cells were recovered on D21, endothelial cells were unfortunately not captured.

The authors should include the GO analysis on d3 cells in Figure 1H, as these are likely the most interesting population to look at functionally.

We thank the reviewer for this idea. Accordingly, as mentioned also above, we have now included GO analysis of cells from D3. The results are depicted in the revised Fig. 2d (old Fig. 1h) and described the manuscript text on page 5, lines 25-32.

In Figure 2A the authors show that at D4 after RFA injury of the 3D NHP model GFP positive cells are recruited to the site of injury. What does this look like at D15 or 21, as Figure 2B appears to indicate to progress in time?

Live cell tracking experiments indicated that already on D4 the vast majority of HVPs has departed from the seeding site and reached the RFA area (old Fig. 2a, now revised Fig. 3a), suggesting that HVP recruitment is accomplished within the first few days after injury. We agree with the reviewer that a gradual increase of GFP⁺ cells in the RFA area is visible until D21. New analyses can now indicate that local proliferation of GFP⁺ cells within the injury takes place and is higher in the first 2 weeks, with around 27% and 11% of the cells being positive for PH3 at D7 and D15, respectively. Later, on D21, when over 90% of GFP⁺ cells are cTNT⁺ cardiomyocytes, proliferation rate drops to 5%, in concordance with myocytic maturation. These new results are presented in the revised Fig. 3, panels c and d, and described in the text on page 7, lines 1-2.

Which percentage of the GFP⁺ cells are cTNT positive at D15 and why does this not correspond to a functional improvement?

Following the reviewer' suggestion, we have now analyzed the percentage of GFP⁺ cells that are expressing cTNT on D15 and D21. As now indicated in the revised Fig. 3c, around 63% of cTNT⁺ cells are detected on D15, and their number increase to 91% on D21. However, as visible in Fig. 3b, cTNT⁺ cells on D15 are still immature cardiomyocytes with poorly organized myofibril and in the process of integrating and forming cell-cell contacts, before functional improvement is seen on Day 21.

To explore the mechanism for HVP migration the authors performed ligand-receptor pairing analysis on scRNA seq data from HVP cells at 24 and 48 hours after injury and resident host cells. Why was only 1 timepoint of the host cells included here? And why do these cells only include macrophages and fibroblasts as indicated in Supplementary Fig 4a?

The timepoint to study cellular interaction between host cells and seeded HVPs was chosen 24h after injury application, in order to capture the initial state of migration and monitor key signals, which trigger the migratory process. Heart slices were subjected to enzymatic dissociation with papain (see Methods, page 18, lines 28-30); FACS sorting separated GFP⁺ from GFP⁻ cells. Our mild dissociation conditions, which allowed high viability of HVPs, were however not sufficient to dissociate native host CMs and vascular cells (which require more aggressive procedures based on repetitive digestions with enzymatic cocktails including collagenase II). This was taken into consideration at the design of the experiments, as the host cells of most interest - based on the preliminary histological results - were fibroblasts.

On the basis of the data obtained, we believe that there is a specific interaction between HVPs and fibroblasts, which is a key signal in the injury repair mechanism demonstrated. However, we agree with the reviewer, that by excluding all cell types other than macrophages and fibroblasts from our analysis, we have a limited view on the biological processes. There are likely other signaling pathways involved between distinct cell types and HVPs, which are not

represented in our analysis.

The authors should confirm the relevance of these ligand receptor pairings in the NHP model and in the porcine model.

The constructive feedback on the *in vitro* receptor-ligand analysis is much appreciated. We have now extended our *ex vivo* and *in vivo* analysis of the HVP migratory capacity towards RFA injured myocardium by performing a new set of experiments using the specific CXCR4 antagonist AMD3100. Treatment of HVPs with AMD3100 demonstrated a dose-dependent inhibition of CXCL12-mediated attraction in the trans-well migration assay, as illustrated in the revised Fig. 4c. Moreover, pharmacological CXCR4 blockade could dramatically reduce HVP migration both in the NHP and the porcine RFA-injury models *ex vivo* and *in vivo*, strengthening the relevance of the CXCR4/CXCL12 axis in this process. These new results are presented in the revised Extended Data Fig. 4d and the revised Fig. 6e and described in the text on page 8, lines 24-30 and page 11, lines 10-11 of the revised manuscript.

The authors should extend the timepoints for single cell analysis to define the mechanism for chemotaxis as the recruitment mainly takes places after 48 hours.

We agree with the reviewer that extending the single cell analysis to further time points later than 48h might uncover additional mechanisms for chemotaxis different from the CXCR4/CXCL12 axis. However, our analysis at 24h and 48h already captured the key signals that initiate the migratory process and we can now demonstrate that HVP chemotaxis can be modulated by the specific CXCR4 blocker AMD3100, both *ex vivo* and *in vivo*, as detailed in the point above. Thus, we believe that CXCR4/CXCL12 plays a pivotal role in the migratory process of the HVPs towards the injured myocardium.

How does the mass spec data related to secreted proteins presented in Supplementary Fig 5b correspond to the ligands found by the scRNA seq analysis?

We thank the reviewer for this question. Actually, we could not detect the CXCL12 ligand in our mass spec data of secreted proteins. We attribute this to the technical processing of the samples for mass spec, which included a filtration step to eliminate proteins with molecular weight smaller than 10 kDa (corresponding to the "medium-only" background). We have specified this in the Methods on page 21, line 13. All isoforms of CXCL12 have a molecular mass smaller than 10 kDa at the mass spectrometry and were very likely lost.

In order to verify the validity of our mass spec results, we have now evaluated the expression of the secreted proteins detected by mass spec in our scRNAseq data set. Most of the secreted proteins (GOT2, ATP5A1, FSTL1, MDH2, DLD, SPARC, LDHA, COL6A1, FN1, LDHB, COL1A2) are detectable as transcripts in the clusters 3, 1 and 4, corresponding to migrating HVPs, NHP-cardiac fibroblasts and NHP-macrophages, respectively. Below, we provide for the reviewer the comparison between secretome at 48h and scRNA-seq.

Secretome	scRNA-Seq						
	0. Early HVPs	1. NHP_cardiac Fibroblasts	2. Activated HVPs	3. Migrating HVPs	4. NHP_Macrophages	5. Proliferating HVPs	6. Early vCMs
DEPs at 48h after RFA							
GOT2				GOT2	GOT2		
ATP5A1	ATP5A1	ATP5A1		ATP5A1	ATP5A1		
FSTL1		FSTL1	FSTL1	FSTL1			FSTL1
MDH2	MDH2			MDH2	MDH2		MDH2
DLG					DLG		
SPARC	SPARC			SPARC	SPARC		
HSPG2							
LDHA	LDHA			LDHA	LDHA		
COL6A1		COL6A1		COL6A1	COL6A1		
PRDX1	PRDX1	PRDX1			PRDX1		PRDX1
FN1	FN1			FN1	FN1		
LDHB		LDHB	LDHB	LDHB	LDHB	LDHB	LDHB
YWHAZ							
SPTAN1		SPTAN1			SPTAN1		
THBS4							
COL1A2		COL1A2		COL1A2	COL1A2	COL1A2	

The authors state that SLIT/ROBO signaling is responsible for the retraction of fibroblasts from the injured area. This is a very intriguing suggestion that deserves further investigation. What happens to the identity of the fibroblasts?

To better dissect the identity of ROBO1⁺ fibroblasts that undergo repulsion, we examined the expression of periostin (as marker of a specialized reparative subpopulation of cardiac fibroblasts required for healing and scar formation after injury) in dsRed⁺ cells on D8 using flow cytometry. Interestingly, the vast majority of ROBO1⁺ fibroblasts expressed periostin, while only few ROBO1⁻ fibroblasts were positive for this marker. These new results are now presented in panel f of the revised Fig. 5 and described in the text on page 10, lines 26-28.

The data obtained in the injured porcine heart are promising and support a cardioprotective effect. However, the authors should confirm some of mechanisms defined in the ex vivo NHP model also in the intact mammalian heart to further validate the relevance of the findings. Is the migration observed in Figure 4D also dependent on CXCR4 and CXCL12? Do the fibroblasts retract and is SLIT/ROBO signaling involved?

We thank the reviewer for the valid suggestions. As mentioned above, we have now extended our *in vivo* analysis of the HVP migratory capacity towards RFA injured porcine myocardium by applying the specific CXCR4 antagonist AMD3100 in concomitance with HVPs injection. Pharmacological CXCR4 blockade could dramatically reduce HVP migration *in vivo*, strengthening the relevance of the CXCR4/CXCL12 axis in this process. The new results are presented in the revised Fig. 6e and described in the text on page 11, lines 10-11 of the revised manuscript.

Moreover, we now provide new data on the cardioprotective effects of HVPs in an additional porcine model of chronic heart failure with cardiac fibrosis, which we believe has high clinical relevance (new Fig. 7 of the revised manuscript). Our new data indicate that once transplanted into necrotic tissue with extensive fibrosis 21 days post myocardial infarction (new Fig. 7a), HVPs lead to de novo formation of ventricular heart muscle (new

Fig. 7b, c), resulting in significant scar volume reduction (new Fig. 7 g) and prevention of heart failure progression (new Fig. 7h,i), as assessed by cardiac MRI at 3 months after cell transplantation. These results are described in a new paragraph “HVPs re-muscularize fibrotic scars and preserve cardiac function in a porcine model of chronic ischemia *in vivo*” on pages 11-13 of the revised manuscript. For the ease of accessibility, the new Fig. 7 can also be found below.

Figure 7. HVPs re-muscularize fibrotic scars and preserve cardiac function in a porcine model of chronic ischemia *in vivo*. **a**, Schematic of *in vivo* experimental design. Myocardial infarction was performed by balloon-occlusion of the left coronary artery (LAD) (ischemia) and reperfusion after 90 minutes. Triple-immunosuppressive regime (IMS) consisted of

cyclosporine (D-6 to D84), methylprednisolone (D-1 to D84) and Abatacept (D-1 to D84). Analysis of baseline infarct volume was conducted by cardiac MRI (cMRI) on day -7 followed by epicardial cell injection (15 injection sites, total 1×10^9 HVPs) into the chronic ischemic area. After 12 weeks follow-up, cMRI scans were the clinical end-point and animals were

sacrificed for histological analyses. **b**, Overview of infarct zone and human grafts with labelling of porcine myocardium (HuNu⁻ cTNT⁺), infarct zone and cardiac troponin T positive grafts (HuNu⁺ cTNT⁺). Scale bar 2 mm. **c-e**, Representative immunohistochemical images of human grafts after staining for the ventricular muscle marker MLC2v (**c**), the intercalated disc protein N-cadherin (**d**), and the endothelial marker CD31 (**e**) at 12 weeks. Scale bar 50 μ m. Lower panels show magnifications of boxed areas. Scale bar 15 μ m. **f**, Representative left ventricular (LV) cMRI images of diastole and systole used for calculation of infarct volume, left ventricular ejection fraction and global longitudinal strain. **g-i**, Statistical analysis of infarct volume (**g**), left ventricular ejection fraction (LVEF, **h**) and global longitudinal strain (GLS, **i**). Data are shown as min-to-max range with individual data points, n=10 pigs in the vehicle group and n=7 pigs in the HVP-treated group, *p<0.05, **p<0.005, ***p<0.001 (t-test).

Decision Letter, first revision:

Dear Professor Laugwitz,

Your manuscript, "Developmental programs determine migratory and anti-fibrotic potential of human cardiac progenitors in heart regeneration", has now been seen by two of the original referees. As you will see from their comments (attached below), referee 1 finds this work of interest, but still raises some important points. Although we are also very interested in this study, we believe that these concerns should be addressed before we can consider publication in Nature Cell Biology.

Nature Cell Biology editors discuss the referee reports in detail within the editorial team, including the chief editor, to identify key referee points that should be addressed with priority, and requests that are overruled as being beyond the scope of the current study.

In this case, we find that it is important to comply with all the referee's suggestions concerning the interpretation of particular parts and we request that you "tone down" accordingly, but also provide the necessary clarifications.

On the contrary, although we agree with the referee that the following additions would provide valuable insight, we do not find it necessarily within the scope of your study to:

1. use human heart muscle from patients to perform further experimental analysis
2. provide EKG data, although we recommend that you do discuss the latter in your manuscript, as suggested by the referee
3. address the question whether the HPV approach achieves higher remuscularization compared to direct cardiomyocyte implantation at a lower risk of arrhythmia

Therefore, addressing the points 1-3 listed right above will not be necessary for reconsideration of the manuscript at this journal. We are committed to providing a fair and constructive peer-review process, so please feel free to contact me if you would like to discuss any of the referee comments further.

Finally please pay close attention to our guidelines on statistical and methodological reporting (listed below) as failure to do so may delay the reconsideration of the revised manuscript. In particular, if you have not done so already, please provide:

We therefore invite you to take these points into account when revising the manuscript. In addition, when preparing the revision please:

- ensure that it conforms to our format instructions and publication policies (see below and www.nature.com/nature/authors/).

- provide a point-by-point rebuttal to the full referee reports verbatim, as provided at the end of this letter.

- provide the completed Editorial Policy Checklist (found here <https://www.nature.com/authors/policies/Policy.pdf>), and Reporting Summary (found here <https://www.nature.com/authors/policies/ReportingSummary.pdf>). This is essential for reconsideration of the manuscript and these documents will be available to editors and referees in the event of peer review. For more information see <http://www.nature.com/authors/policies/availability.html> or contact me.

Nature Cell Biology is committed to improving transparency in authorship. As part of our efforts in this direction, we are now requesting that all authors identified as 'corresponding author' on published papers create and link their Open Researcher and Contributor Identifier (ORCID) with their account on the Manuscript Tracking System (MTS), prior to acceptance. ORCID helps the scientific community achieve unambiguous attribution of all scholarly contributions. You can create and link your ORCID from

the home page of the MTS by clicking on 'Modify my Springer Nature account'. For more information please visit www.springernature.com/orcid.

[REDACTED]

We normally request to receive the revision within four weeks. Given the upcoming holidays, however, I would suggest that you send us back the revised manuscript within six weeks. If submitted within this time period, reconsideration of the revised manuscript will not be affected by related studies published elsewhere, or accepted for publication in Nature Cell Biology in the meantime. We would be happy to consider a revision even after this timeframe, but in that case we will consider the published literature at the time of resubmission when assessing the file. Please, let me know if there are any additional issues and you need more time.

We hope that you will find our referees' comments, and editorial guidance helpful. Please do not hesitate to contact me if there is anything you would like to discuss.

Best wishes,
Stelios

Stylianos Lefkopoulos, PhD
He/him/his
Associate Editor, Nature Cell Biology
Springer Nature
Heidelberger Platz 3, 14197 Berlin, Germany

E-mail: stylianos.lefkopoulos@springernature.com
Twitter: @s_lefkopoulos

Reviewers' Comments:

Reviewer #1:

Remarks to the Author:

This is a revised manuscript with a comprehensive set of new data. Most importantly, remuscularization by HVP implantation was confirmed in a chronic model of heart disease in pigs. The study comprises a large set of in vitro experiments with a main focus on HVP properties after seeding on non-human primate heart slices. The authors have a valid point by arguing that the NHP model, in contrast to a human heart model, allows for studies in healthy tissue. However, since the goal is to use HVP in heart failure, it may be more adequate to use human heart muscle from patients. The fibroblast repulsive effects of HPVs are nicely shown in vitro. Whether such effects would be present and relevant in vivo cannot be concluded from the pig studies. Finally, the advantage over the application of PSC-derived cardiomyocytes remains unclear and the extent of remuscularization rather small, despite injection of a huge amount of HVP (1×10^9). From a developmental biology perspective, it is of cause interesting to observe that the HPVs are capable of further differentiating along their predetermined lineage after injection into the adult heart. Whether this can establish meaningful amounts of new muscle cannot be concluded from the presented data. A key advantage may be the lack of arrhythmia after implantation of developmentally less mature HPVs compared to electrically active terminally differentiated cardiomyocytes.

Specific comments:

The key concern in remuscularization therapies is arrhythmia induction. I am surprised that no EKG data are presented. The authors need to either present EKG data or if not collected in the pig model clearly state that the lack of EKG recording is a major limitation of the study.

The claim that there would be a therapeutically relevant migration of HPVs to the site of injury in vivo is not well supported by the presented data. In the chronic pig study HPVs were implanted into border zone and scar tissue (page 24, lines 19-20).

Page 11 line 8 states that methylprednisolone and tacrolimus were administered in the LEA29Y model. This is in contrast to the statement in the response letter.

There are several misleading/overstatements, which should be avoided, e.g., "ideal biotherapeutic" (page 2, line 23), "scar-less healing" (page 3, line 24), HPV seeded on a tissue slice as 3D tissue culture (page 4, line 6), "ideal setting for investigating cell-base mechanisms" (page 4, lines 13-14), "allogeneic" (page 10, line 32).

Do the authors interpret the smaller decrease (Fig. 7h) in heart function in the HPV treated group compared to the controls as key evidence for efficacy of HPV treatment?

vehicle $39.4 \pm 1.3\%$ to $29.4 \pm 3.9\%$ EF- indicated as deterioration
HVP $37.3 \pm 2.8\%$ to $31.9 \pm 3.0\%$ EF - indicated as preservation
With no differences in delta LVEF this would be a rather bold assumption.

Overall, this is an interesting study with interesting biology by an excellent group of investigators. It advances the therapeutic strategy of the previously completed ESCORT clinical trial, from which no evidence for cardiomyocyte differentiation has been reported, so far. Key questions, which remain unanswered are whether the proposed HPV implantation approach can achieve higher remuscularization compared to direct cardiomyocyte implantation (data from Murry and LaFlamme groups) at a lower risk for arrhythmia.

Reviewer #3:

Remarks to the Author:

This is an important and insightful study and most of the issues that were previously raised by reviewers have now been dealt with. There are no further comments.

GUIDELINES FOR SUBMISSION OF NATURE CELL BIOLOGY ARTICLES

ARTICLE FORMAT

ABSTRACT – should not exceed 150 words and should be unreferenced. This paragraph is the most visible part of the paper and should briefly outline the background and rationale for the work, and accurately summarize the main results and conclusions. Key genes, proteins and organisms should be specified to ensure discoverability of the paper in online searches.

TEXT – the main text consists of the Introduction, Results, and Discussion sections and must not exceed 3500 words including the abstract. The Introduction should expand on the background relating to the work. The Results should be divided in subsections with subheadings, and should provide a concise and accurate description of the experimental findings. The Discussion should expand on the findings and their implications. All relevant primary literature should be cited, in particular when discussing the background and specific findings.

REFERENCES – are limited to a total of 70 in the main text and Methods combined,. They must be numbered sequentially as they appear in the main text, tables and figure legends and Methods and must follow the precise style of Nature Cell Biology references. References only cited in the Methods

should be numbered consecutively following the last reference cited in the main text. References only associated with Supplementary Information (e.g. in supplementary legends) do not count toward the total reference limit and do not need to be cited in numerical continuity with references in the main text. Only published papers can be cited, and each publication cited should be included in the numbered reference list, which should include the manuscript titles. Footnotes are not permitted.

Methods should be written concisely, but should contain all elements necessary to allow interpretation and replication of the results. As a guideline, Methods sections typically do not exceed 3,000 words. The Methods should be divided into subsections listing reagents and techniques. When citing previous methods, accurate references should be provided and any alterations should be noted. Information must be provided about: antibody dilutions, company names, catalogue numbers and clone numbers for monoclonal antibodies; sequences of RNAi and cDNA probes/primers or company names and catalogue numbers if reagents are commercial; cell line names, sources and information on cell line identity and authentication. Animal studies and experiments involving human subjects must be reported in detail, identifying the committees approving the protocols. For studies involving human subjects/samples, a statement must be included confirming that informed consent was obtained. Statistical analyses and information on the reproducibility of experimental results should be provided in a section titled “Statistics and Reproducibility”.

All Nature Cell Biology manuscripts submitted on or after March 21 2016, must include a Data availability statement as a separate section after Methods but before references, under the heading “Data Availability”. For Springer Nature policies on data availability see <http://www.nature.com/authors/policies/availability.html>; for more information on this particular policy see <http://www.nature.com/authors/policies/data/data-availability-statements-data-citations.pdf>. The Data availability statement should include:

- Accession codes for primary datasets (generated during the study under consideration and designated as “primary accessions”) and secondary datasets (published datasets reanalysed during the study under consideration, designated as “referenced accessions”). For primary accessions data should be made public to coincide with publication of the manuscript. A list of data types for which submission to community-endorsed public repositories is mandated (including sequence, structure, microarray, deep sequencing data) can be found here <http://www.nature.com/authors/policies/availability.html#data>.

- Unique identifiers (accession codes, DOIs or other unique persistent identifier) and hyperlinks for datasets deposited in an approved repository, but for which data deposition is not mandated (see here for details <http://www.nature.com/sdata/data-policies/repositories>).
- At a minimum, please include a statement confirming that all relevant data are available from the authors, and/or are included with the manuscript (e.g. as source data or supplementary information), listing which data are included (e.g. by figure panels and data types) and mentioning any restrictions on availability.
- If a dataset has a Digital Object Identifier (DOI) as its unique identifier, we strongly encourage including this in the Reference list and citing the dataset in the Methods.

We recommend that you upload the step-by-step protocols used in this manuscript to the Protocol Exchange. More details can found at www.nature.com/protocolexchange/about.

DISPLAY ITEMS – main display items are limited to 6-8 main figures and/or main tables. For Supplementary Information see below.

FIGURES – Colour figure publication costs \$395 per colour figure. All panels of a multi-panel figure must be logically connected and arranged as they would appear in the final version. Unnecessary figures and figure panels should be avoided (e.g. data presented in small tables could be stated briefly in the text instead).

All imaging data should be accompanied by scale bars, which should be defined in the legend. Cropped images of gels/blots are acceptable, but need to be accompanied by size markers, and to retain visible background signal within the linear range (i.e. should not be saturated). The boundaries of panels with low background have to be demarked with black lines. Splicing of panels should only be considered if unavoidable, and must be clearly marked on the figure, and noted in the legend with a statement on whether the samples were obtained and processed simultaneously. Quantitative comparisons between samples on different gels/blots are discouraged; if this is unavoidable, it has to be performed for samples derived from the same experiment with gels/blots were processed in parallel, which needs to be stated in the legend.

Figures should be provided at approximately the size that they are to be printed at (single column is 86 mm, double column is 170 mm) and should not exceed an A4 page (8.5 x 11"). Reduction to the scale that will be used on the page is not necessary, but multi-panel figures should be sized so that the whole figure can be reduced by the same amount at the smallest size at which essential details in each panel are visible. In the interest of our colour-blind readers we ask that you avoid using red and green for

contrast in figures. Replacing red with magenta and green with turquoise are two possible colour-safe alternatives. Lines with widths of less than 1 point should be avoided. Sans serif typefaces, such as Helvetica (preferred) or Arial should be used. All text that forms part of a figure should be rewritable and removable.

Regardless of format, all figures must be vector graphic compatible files, not supplied in a flattened raster/bitmap graphics format, but should be fully editable, allowing us to highlight/copy/paste all text and move individual parts of the figures (i.e. arrows, lines, x and y axes, graphs, tick marks, scale bars etc). The only parts of the figure that should be in pixel raster/bitmap format are photographic images or 3D rendered graphics/complex technical illustrations.

Unprocessed scans of all key data generated through electrophoretic separation techniques need to be presented in a supplementary figure that should be labeled and numbered as the final supplementary figure, and should be mentioned in every relevant figure legend. This figure does not count towards the total number of figures and is the only figure that can be displayed over multiple pages, but should be provided as a single file, in PDF or TIFF format. Data in this figure can be displayed in a relatively informal style, but size markers and the figures panels corresponding to the presented data must be indicated.

The total number of Supplementary Figures (not including the “unprocessed scans” Supplementary Figure) should not exceed the number of main display items (figures and/or tables (see our Guide to Authors and March 2012 editorial <http://www.nature.com/ncb/authors/submit/index.html#suppinfo>; <http://www.nature.com/ncb/journal/v14/n3/index.html#ed>). No restrictions apply to Supplementary Tables or Videos, but we advise authors to be selective in including supplemental data.

Each Supplementary Figure should be provided as a single page and as an individual file in one of our accepted figure formats and should be presented according to our figure guidelines (see above). Supplementary Tables should be provided as individual Excel files. Supplementary Videos should be

provided as .avi or .mov files up to 50 MB in size. Supplementary Figures, Tables and Videos must be accompanied by a separate Word document including titles and legends.

GUIDELINES FOR EXPERIMENTAL AND STATISTICAL REPORTING

REPORTING REQUIREMENTS – To improve the quality of methods and statistics reporting in our papers we have recently revised the reporting checklist we introduced in 2013. We are now asking all life sciences authors to complete two items: an Editorial Policy Checklist (found here <https://www.nature.com/authors/policies/Policy.pdf>) that verifies compliance with all required editorial policies and a Reporting Summary (found here <https://www.nature.com/authors/policies/ReportingSummary.pdf>) that collects information on experimental design and reagents. These documents are available to referees to aid the evaluation of the manuscript. Please note that these forms are dynamic ‘smart pdfs’ and must therefore be downloaded and completed in Adobe Reader. We will then flatten them for ease of use by the reviewers. If you would like to reference the guidance text as you complete the template, please access these flattened versions at <http://www.nature.com/authors/policies/availability.html>.

We strongly recommend the presentation of source data for graphical and statistical analyses as a separate Supplementary Table, and request that source data for all independent repeats are provided when representative experiments of multiple independent repeats, or averages of two independent experiments are presented. This supplementary table should be in Excel format, with data for different

figures provided as different sheets within a single Excel file. It should be labelled and numbered as one of the supplementary tables, titled "Statistics Source Data", and mentioned in all relevant figure legends.

Author Rebuttal, first revision:

POINT-BY-POINT RESPONSE TO REVIEWERS' COMMENTS

We are pleased and thank the Reviewers and Editors for their constructive criticism and for appreciating the substantial improvements made during the revision. We have now implemented all requested editorial changes in the updated manuscript. Our revised manuscript including figures, as well as point-by-point rebuttal are provided for the reviewers and editors.

Responses to Reviewer 1

This is a revised manuscript with a comprehensive set of new data. Most importantly, remuscularization by HVP implantation was confirmed in a chronic model of heart disease in pigs. The study comprises a large set of in vitro experiments with a main focus on HVP properties after seeding on non-human primate heart slices. The authors have a valid point by arguing that the NHP model, in contrast to a human heart model, allows for studies in healthy tissue. However, since the goal is to use HVP in heart failure, it may be more adequate to use human heart muscle from patients.

We would like to thank the reviewer for the appreciation of our work and the additional sets of data included during the revision. We are also grateful for the further insightful comments and suggestions.

As pointed out before, patients with end-stage heart failure represent a very heterogeneous population, since they suffer from different types of cardiomyopathies and are at distinct stages of their heart muscle disease. Furthermore, these patients show different transcriptional profiles of diseased tissues with fibrosis and their medication accounts for additional confounding factors. Thus, human heart slices are less suitable for standardized investigational studies.

For this reason, we considered healthy NHP hearts as an appropriate surrogate to establish a standardized *ex vivo* model of chronic and acute cardiac injury, which could allow us to investigate molecular pathways and functional dynamic states of human cardiac progenitor cells during heart regeneration and repair at a single cell resolution. Certainly, it would be of importance to recapitulate our findings *ex-vivo* in human heart

muscle slices from patients with chronic heart failure (e.g. ischemic heart disease, genetic cardiomyopathies, and post-myocarditis) and assess whether HVP therapy could be beneficial to reduce pre-existing fibrosis and improve cardiac function. In our opinion, these studies will provide further valuable insights, but be beyond the scope of the current study. We have now elaborated on this in the revised Discussion on **page 12, lines 12-14**, as follow: *“Future studies should investigate whether ex-vivo human heart slices could predict outcome of cell-based regeneration in patients with different aetiologies of heart failure (e.g. ischemic, genetic, and inflammatory).”*

The fibroblast repulsive effects of HPVs are nicely shown in vitro. Whether such effects would be present and relevant in vivo cannot be concluded from the pig studies.

We completely agree with the reviewer. Therefore, we have now discussed this point accordingly in the revised Discussion on **page 12, lines 21-24**, as following: *“Our scRNAseq unravelled that the SLIT2/ROBO1 axis mediates the ability of HVPs to repel CFs, thus reducing scarring ex-vivo. It will be of particular interest to evaluate whether such signalling pathway plays a similar role in-vivo and its pharmacological manipulation could circumvent cell application”.*

Finally, the advantage over the application of PSC-derived cardiomyocytes remains unclear and the extent of remuscularization rather small, despite injection of a huge amount of HVP (1×10^9). From a developmental biology perspective, it is of cause interesting to observe that the HPVs are capable of further differentiating along their predetermined lineage after injection into the adult heart. Whether this can establish meaningful amounts of new muscle cannot be concluded from the presented data. A key advantage may be the lack of arrhythmia after implantation of developmentally less mature HPVs compared to electrically active terminally differentiated cardiomyocytes.

Our *in-vivo* data on the porcine model of chronic ischemic injury clearly indicate that HVP treatment results in *de novo* formation of heart muscle that functionally suffices to prevent progression of heart failure. As such, we believe that the amount of new muscle is indeed “meaningful”. Moreover, we would like to highlight that no other large animal studies, to the best of our knowledge, have evaluated graft-size after 3 months cell transplantation. Two studies using comparable numbers of hPSC-derived CMs to ours ($0.75\text{-}1 \times 10^9$ cells per heart), one in small macaques (*Liu et al., Nat Biotech 2018*) and the other in juvenile pigs (*Romagnuolo et al., Stem Cell Reports 2019*), described average human graft-size ranging between **10-15% of the infarct area after 4 weeks** transplantation. We have measured, under a similar immunosuppressive regime, a graft size ranging **3.0-9.4% (average 4.2%) of the infarct area after 12 weeks**. However, it should be noted that infarct size in our pig cohort ranged between 30 to 40% of the LV myocardium, while it was only half (15- 20%) in the other two studies.

We are aware that direct comparisons with other studies is challenging due to differences in experimental design and methodology assessment. Nevertheless, in our opinion, the extent of remuscularization that we observed in our study is far from being “small” and does not differ significantly from what has been reported in previously published works using large animal models.

On the other hand, we agree with the reviewer that, without a direct head-to-head comparison between HVP and CM application in the same large animal model, the therapeutic advantage of HVPs *versus* CMs remains elusive. This is the subject of ongoing studies and is beyond the scope of the current manuscript. Certainly, our data clearly shows that HVPs, in contrast to CMs, have the capability to sense and migrate to sites of myocardial injury *ex-vivo* and in a RFA-injury model *in-vivo*. HVPs repulse activated fibroblasts *ex-vivo* and reduce scar formation *in-vivo* in the setting of both acute and chronic injury. CMs do not embark on these cellular properties during heart regeneration.

Specific comments:

The key concern in remuscularization therapies is arrhythmia induction. I am surprised that no EKG data are presented. The authors need to either present EKG data or if not collected in the pig model clearly state that the lack of EKG recording is a major limitation of the study.

[REDACTED]

The claim that there would be a therapeutically relevant migration of HPVs to the site of injury *in vivo* is not well supported by the presented data. In the chronic pig study HPVs were implanted into border zone and scar tissue (page 24, lines 19-20).

Though we agree with the reviewer that our data on the chronic ischemia pig model do not support a therapeutically relevant migration of HVPs, our data on the acute RFA-injury pig model support this claim. Here, HVPs were injected ~1cm far away from the RFA-injured site and a directed, CXCR4-guided migration of eGFP⁺-HVPs towards the RFA-injured area was observed, ultimately resulting to a significant reduction of scar volume, as shown in **Fig. 6d-f**.

In the chronic ischemia pig study, where HVPs were implanted into border zone and scar tissue, we observed that human HVP-derived CMs were broadly dispersed around the injection canal, suggesting an intrinsic migratory capacity of HVPs also in the setting of chronic injury (**Extended Data Movie 3**). Given that in this model we have not applied pharmacological CXCR4 blockage (AMD3100), we admit that our interpretation of the

data might have been overstated. Thus, we have now removed this statement from the revised Results section on page 10.

Page 11 line 8 states that methylprednisolone and tacrolimus were administered in the LEA29Y model. This is in contrast to the statement in the response letter.

We apologize for the incongruence and thank the reviewer for indicating this. We have now corrected this point on page 9, lines 28-29, which reads as follow: *“Animals were treated daily with methylprednisolone and euthanized...”*.

There are several misleading/overstatements, which should be avoided, e.g., “ideal biotherapeutic” (page 2, line 23), “scar-less healing” (page 3, line 24), HPV seeded on a tissue slice as 3D tissue culture (page 4, line 6), “ideal setting for investigating cell-base mechanisms” (page 4, lines 13-14), “allogeneic” (page 10, line 32).

Following the reviewer’s suggestion, all indicated statements have been removed from the revised manuscript text and clarifications have been provided.

Do the authors interpret the smaller decrease (Fig. 7h) in heart function in the HPV treated group compared to the controls as key evidence for efficacy of HPV treatment?
vehicle $39.4 \pm 1.3\%$ to $29.4 \pm 3.9\%$ EF- indicated as deterioration
HVP $37.3 \pm 2.8\%$ to $31.9 \pm 3.0\%$ EF - indicated as preservation
With no differences in delta LVEF this would be a rather bold assumption.

We acknowledge the reviewer’s concern regarding the interpretation of our results. However, in clinical medicine, treatments are generally considered to be effective not only if they improve the functional status but also if they prevent further worsening of a severe clinical condition over time. All therapeutic interventions for chronic heart failure patients to date, e.g. drugs (β -blockers, mineralocorticoids, SGLT-2 inhibitors, and ARB/nepriylsin inhibitors) and medical devices (cardiac resynchronisation therapy), aim primarily at preventing the adverse cardiac remodelling of a failing heart in patients and do not generally improve global LV ejection fraction.

In a clinically relevant porcine model of chronic heart failure with large fibrotic scar tissue after MI (infarct volumes averaging 33% of the LV myocardium), we show that HVPs lead to *de novo* formation of ventricular heart muscle, resulting in significant scar volume reduction and prevention of adverse cardiac remodeling progression at 3 months after cell transplantation. The global longitudinal strain, a sensitive cMRI measure of LV function, significantly worsened in the vehicle-treated group (-3.1 ± 1.0) compared to the HVP-group (-0.2 ± 0.6), demonstrating that HVP-treatment attenuated the progressive decline of cardiac function. Thus, in our opinion, HVP therapy is

considered effective.

As correctly indicated by the reviewer, in this large animal model, we did not observe a statistically significant difference in Delta-LVEF between the HVP-treated ($-5.4\% \pm 8.8\%$) and control ($-10.0\% \pm 12.3\%$) groups. For clarity, we have now emphasized this in the revised Results on **page 11, lines 4-6**, as follow: “Over 12 weeks, LVEF further deteriorated significantly by $\sim 10\%$ in controls ($29.4 \pm 3.9\%$) and only by half ($\sim 5\%$) in HVP-treated animals ($31.9\% \pm 3.0\%$), though differences between the groups did not reach statistical significance (Fig.7h)”.

Overall, this is an interesting study with interesting biology by an excellent group of investigators. It advances the therapeutic strategy of the previously completed ESCORT clinical trial, from which no evidence for cardiomyocyte differentiation has been reported, so far. Key questions, which remain unanswered are whether the proposed HPV implantation approach can achieve higher remuscularization compared to direct cardiomyocyte implantation (data from Murry and LaFlamme groups) at a lower risk for arrhythmia.

We thank the reviewer for the overall positive assessment of the importance and novelty of our work. Our study should be considered primarily as a mechanistic work but clearly points to the potential value of considering progenitors as a new heart regenerative therapeutic, with properties distinct from cardiomyocytes. Whether they prove to be clinically valuable, with fewer arrhythmias requires further investigations. As mentioned above, we have now acknowledged this in the revised Discussion on **pages 12 and 13, lines 31-32 and 1**, respectively. The new sentence reads “Moreover, before HVP-transplantation can be translated to humans future investigations should determine whether HVP-based therapies could achieve higher remuscularization compared to direct CM implantation with a lower risk of ventricular arrhythmia

Responses to Reviewer 3

This is an important and insightful study and most of the issues that were previously raised by reviewers have now been dealt with. There are no further comments.

We would like to thank again the reviewer for the appreciation of our work and his/her efforts and critical input during the review of our manuscript, which helped us to improve our study considerably.

Decision Letter, second revision:

22nd January 2022

Dear Dr. Laugwitz,

Thank you for submitting your revised manuscript "Migratory and anti-fibrotic programs define the regenerative potential of human cardiac progenitors" (NCB-L46317B). We find that in this version you have addressed the remaining referee requests and therefore we'll be happy in principle to publish it in Nature Cell Biology, pending minor revisions to comply with our editorial and formatting guidelines.

Thank you again for your interest in Nature Cell Biology Please do not hesitate to contact me if you have any questions.

Best regards,
Stelios

Stylianos Lefkopoulos, PhD
He/him/his
Associate Editor, Nature Cell Biology
Springer Nature
Heidelberger Platz 3, 14197 Berlin, Germany

E-mail: stylianos.lefkopoulos@springernature.com
Twitter: @s_lefkopoulos

4th February 2022

Dear Dr. Laugwitz,

Thank you for your patience as we've prepared the guidelines for final submission of your Nature Cell Biology manuscript, "Migratory and anti-fibrotic programs define the regenerative potential of human cardiac progenitors" (NCB-L46317B). Please carefully follow the step-by-step instructions provided in the attached file, and add a response in each row of the table to indicate the changes that you have made. Please also check and comment on any additional marked-up edits we have proposed within the text. Ensuring that each point is addressed will help to ensure that your revised manuscript can be swiftly handed over to our production team.

We would like to start working on your revised paper, with all of the requested files and forms, as soon as possible (preferably within one week). Please get in contact with us if you anticipate delays.

In recognition of the time and expertise our reviewers provide to Nature Cell Biology's editorial process, we would like to formally acknowledge their contribution to the external peer review of your manuscript entitled "Migratory and anti-fibrotic programs define the regenerative potential of human cardiac progenitors". For those reviewers who give their assent, we will be publishing their names alongside the published article.

Nature Cell Biology offers a Transparent Peer Review option for new original research manuscripts submitted after December 1st, 2019. As part of this initiative, we encourage our authors to support increased transparency into the peer review process by agreeing to have the reviewer comments, author rebuttal letters, and editorial decision letters published as a Supplementary item. When you submit your final files please clearly state in your cover letter whether or not you would like to participate in this initiative. Please note that failure to state your preference will result in delays in accepting your manuscript for publication.

Cover suggestions

As you prepare your final files we encourage you to consider whether you have any images or illustrations that may be appropriate for use on the cover of Nature Cell Biology.

Nature Cell Biology has now transitioned to a unified Rights Collection system which will allow our Author Services team to quickly and easily collect the rights and permissions required to publish your work. Approximately 10 days after your paper is formally accepted, you will receive an email in providing you with a link to complete the grant of rights. If your paper is eligible for Open Access, our Author Services team will also be in touch regarding any additional information that may be required to arrange payment for your article.

Please note that Nature Cell Biology is a Transformative Journal (TJ). Authors may publish their research with us through the traditional subscription access route or make their paper immediately open access through payment of an article-processing charge (APC). Authors will not be required to make a final decision about access to their article until it has been accepted. Find out more about Transformative Journals

Authors may need to take specific actions to achieve compliance with funder and institutional open access mandates. For submissions from January 2021, if your research is supported by a funder that requires immediate open access (e.g. according to Plan S principles) then you should select the gold OA route, and we will direct you to the compliant route where possible. For authors selecting the subscription publication route our standard licensing terms will need to be accepted, including our self-archiving policies. Those standard licensing terms will supersede any other terms that the author or any third party may assert apply to any version of the manuscript.

For information regarding our different publishing models please see our Transformative Journals page. If you have any questions about costs, Open Access requirements, or our legal forms, please contact ASJournals@springernature.com.

Please use the following link for uploading these materials:
[REDACTED]

Best regards,

Nyx Hills
Staff
Nature Cell Biology

On behalf of

Stylios Lefkopoulos, PhD
He/him/his
Associate Editor
Nature Cell Biology
Springer Nature
Heidelberger Platz 3, 14197 Berlin, Germany

E-mail: stylios.lefkopoulos@springernature.com
Twitter: @s_lefkopoulos

Final Decision Letter:

Dear Karl,

Thank you for your patience while we have been evaluating your revisions. I am pleased to inform you that your manuscript, "Migratory and anti-fibrotic programs define the regenerative potential of human

cardiac progenitors", has now been accepted for publication in Nature Cell Biology. Congratulations to you and all the coauthors for this nice work and for making it to publication!

Please note that Nature Cell Biology is a Transformative Journal (TJ). Authors may publish their research with us through the traditional subscription access route or make their paper immediately open access through payment of an article-processing charge (APC). Authors will not be required to make a final

decision about access to their article until it has been accepted. Find out more about Transformative Journals

Authors may need to take specific actions to achieve compliance with funder and institutional open access mandates. If your research is supported by a funder that requires immediate open access (e.g. according to Plan S principles) then you should select the gold OA route, and we will direct you to the compliant route where possible. For authors selecting the subscription publication route, the journal's standard licensing terms will need to be accepted, including self-archiving policies. Those licensing terms will supersede any other terms that the author or any third party may assert apply to any version of the manuscript.

If your paper includes color figures, please be aware that in order to help cover some of the additional cost of four-color reproduction, Nature Research charges our authors a fee for the printing of their color figures. Please contact our offices for exact pricing and details.

If you have not already done so, we strongly recommend that you upload the step-by-step protocols used in this manuscript to the Protocol Exchange (www.nature.com/protocolexchange), an open online resource established by Nature Protocols that allows researchers to share their detailed experimental know-how. All uploaded protocols are made freely available, assigned DOIs for ease of citation and are fully searchable through nature.com. Protocols and the Nature and Nature research journal papers in which they are used can be linked to one another, and this link is clearly and prominently visible in the online versions of both papers. Authors who performed the specific experiments can act as primary authors for the Protocol as they will be best placed to share the methodology details, but the Corresponding Author of the present research paper should be included as one of the authors. By uploading your Protocols to Protocol Exchange, you are enabling researchers to more readily reproduce or adapt the methodology you use, as well as increasing the visibility of your protocols and papers. You can also establish a dedicated page to collect your lab Protocols. Further information can be found at www.nature.com/protocolexchange/about

You can use a single sign-on for all your accounts, view the status of all your manuscript submissions and reviews, access usage statistics for your published articles and download a record of your refereeing activity for the Nature journals.

Please feel free to contact us if you have any questions and have a great weekend!

With kind regards,
Stelios

Stylios Lefkopoulos, PhD
He/him/his
Associate Editor
Nature Cell Biology
Springer Nature
Heidelberger Platz 3, 14197 Berlin, Germany

E-mail: stylios.lefkopoulos@springernature.com
Twitter: @s_lefkopoulos
